# Co-LoRA: Collaborative Model Personalization on Heterogeneous Multi-Modal Clients

**Minhyuk Seo**[1,2,*] **Taeheon Kim**[2,*] **Hankook Lee**[3] **Jonghyun Choi**[2,†] **Tinne Tuytelaars**[1,†]

[1] KU Leuven  [2] Seoul National University  [3] Sungkyunkwan University
{minhyuk.seo, tinne.tuytelaars}@kuleuven.be
hankook.lee@skku.edu, {thkim0305, jonghyunchoi}@snu.ac.kr

## Abstract

As AI becomes more personal, *e.g.*, Agentic AI, there is an increasing need for personalizing models for various use cases. Personalized federated learning (PFL) enables each client to *collaboratively* leverage other clients' knowledge for better adaptation to the task of interest, without privacy risks. Despite its potential, existing PFL methods remain confined to rather simplified scenarios where data and models are the same across clients. To move towards realistic scenarios, we move beyond these restrictive assumptions by addressing both data and model heterogeneity. We propose a task-relevance-aware model aggregation strategy to reduce parameter interference under heterogeneous data. Moreover, we introduce **Co-LoRA**, a dimension-invariant module that enables knowledge sharing across heterogeneous architectures. To mimic the real-world task diversity, we propose a multi-modal PFL benchmark spanning 40 distinct tasks with distribution shifts over time. Extensive experiments shows that our proposed method significantly outperforms the state-of-the-art PFL methods under heterogeneous scenarios. Code is available at https://github.com/snumprlab/fedmosaic.

## 1 Introduction

Multimodal Large Language Models (MLLMs) with billions of parameters often employ centralized training on massive, heterogeneous datasets using high-performance computing resources (Hurst et al., 2024; Yang et al., 2024a; Team et al., 2024). Such centralized training raises significant concerns about data privacy and high transmission costs (Alvi et al., 2022; Yang et al., 2023). Moreover, as the demand for personalization grows, further fine-tuning is essential to adapt these centrally trained general-purpose models to individual user preferences (Lau et al., 2024; Zhang, 2024).

To address both privacy and personalization, personalized federated learning (PFL) (Smith et al., 2017) has emerged as a decentralized alternative. Recent PFL methods demonstrate that collaboratively leveraging knowledge from other clients significantly improves personalization to the task of interest (Scott et al., 2024; Xie et al., 2024). Unfortunately, despite its potential, most PFL studies overlook the heterogeneity inherent in real-world clients (Zhang et al., 2024b; Bujotzek et al., 2025), where clients typically have distinct models depending on their computational resources (*i.e.*, *model heterogeneity*) and deal with highly personalized data (*i.e.*, *data heterogeneity*), as in agentic AI.

Although some recent work addresses client heterogeneity, it mostly considers either model heterogeneity (Fang et al., 2023; Wu et al., 2024a; Yi et al., 2023) or data heterogeneity (Chen et al., 2024a; Xie et al., 2024; Tamirisa et al., 2024). Some tackle both, but in less realistic, simplified setups, *e.g.*, assigning each client a model with different LoRA (Hu et al., 2022) adapter ranks while keeping the base architecture identical (Cho et al., 2024; Bai et al., 2024), or lacking data heterogeneity by simply splitting a single dataset into non-i.i.d. partitions (Li & Wang, 2019; Alam et al., 2022).

As a realistic setup tackling both challenges, illustrated in Fig. 1, we consider (i) *data heterogeneity*, where clients tackle highly personalized tasks, and (ii) *model heterogeneity*, where clients employ models of different families (*e.g.*, Llama-based (Grattafiori et al., 2024)- vs. Qwen-based (Yang et al., 2024a) MLLMs) and scales (*e.g.*, 1B vs. 3B). To mimic the real-world data heterogeneity, we first

---

*indicates equal contribution. † indicates corresponding authors.  JC is with ECE, IPAI, ASRI at SNU.

Figure 1: **Overview of the heterogeneous personalized federated learning scenarios.** $L_i$ refers to the local model for the $i_{th}$ client. Clients focus on different tasks (*i.e.*, data heterogeneity) where new data are encountered continuously. In addition to data heterogeneity, model architectures may differ across clients (*i.e.*, model heterogeneity) due to differences in hardware constraints.

introduce DRAKE, a comprehensive benchmark for multi-modal PFL with 40 diverse tasks. Unlike prior works that simulate heterogeneity via non-i.i.d. splits of a single dataset (Xie et al., 2024; Long et al., 2024; Morafah et al., 2024), our benchmark assigns each client a distinct multi-modal task (*e.g.*, visual question answering or visual reasoning), while also incorporating temporal distribution shifts inherent in the real world. To the best of our knowledge, this is the first benchmark for multi-modal FL that considers data heterogeneity, as well as distribution shifts.

To address the real-world challenge, we propose **FedMosaic** that jointly addresses both data and model heterogeneity in the realistic scenario. Under data heterogeneity, naive model averaging (McMahan et al., 2017) often degrades performance (Wu et al., 2024c; Yadav et al., 2023) due to interference between models trained on unrelated tasks. To mitigate the interference, inspired by the fact that models trained on similar tasks have less conflict (Gurulingan et al., 2022), we propose **REL**evance-guided **A**ggregation (**RELA**), which constructs a customized global model for each client based on task-relatedness. This enables related clients to share knowledge more effectively.

Under model heterogeneity, aggregating model weights is infeasible due to architectural mismatch (Fan et al., 2024). Although federated distillation (FD) has been proposed to aggregate heterogeneous models' knowledge using logits from public data (Li et al., 2024e), domain discrepancies between public and client data hinder effective knowledge transfer (Wang et al., 2023), and logit extraction is computationally expensive (Malladi et al., 2023), especially for large models. Instead, we propose **Co**llaborative-**LoRA** (**Co-LoRA**), which enables cross-architecture collaboration via dimension-invariant modules $P \in \mathbb{R}^{r \times r}$ and $Q \in \mathbb{R}^r$. Since their sizes depend only on low-rank size $r$, independent of hidden dimension, they are directly shareable across heterogeneous models.

We summarize our contributions as follows:

- Proposing DRAKE, a comprehensive multi-modal federated learning benchmark.
- Proposing RELA, a model aggregation strategy that promotes selective knowledge sharing among models learning relevant tasks only, addressing data heterogeneity.
- Proposing Co-LoRA, shareable across heterogeneous models, addressing model heterogeneity.

## 2   RELATED WORK

**Personalized Federated Learning.** Federated learning aims to train a strong global model in a distributed manner while preserving privacy by sharing model weights instead of raw data (Yurdem et al., 2024; Hu et al., 2024). With the growing importance of model personalization, which allows large foundation models to adapt to individual user preferences, contexts, and needs (Zhang et al., 2024c), personalized federated learning (PFL) (Smith et al., 2017) has emerged. PFL aims to train a personalized model on each client's local data by leveraging shared knowledge from other clients to enhance both personalization and generalization while preserving privacy (Xie et al., 2024).

Recent PFL methods, such as DITTO (Li et al., 2021), FedSim (Pillutla et al., 2022), FedDAT (Chen et al., 2024a), and FedDPA (Long et al., 2024), adopt dual-adapter designs, maintaining separate local and global adapters. The local adapter captures client-specific knowledge to address data heterogeneity, while the global adapter preserves client-agnostic knowledge and mitigates overfitting. Despite their effectiveness, these methods build the global adapter by averaging local adapter parameters, which restricts collaboration to homogeneous models due to dimension and depth mismatches. In contrast, FedMosaic enables knowledge sharing even across heterogeneous architectures by proposing a dimension-invariant shareable module. Moreover, unlike prior work that simulates data heterogeneity

through non-i.i.d. splits of a single dataset, we explore a more realistic setting where clients focus on different tasks, reflecting the diversity of real-world deployments.

**Data Heterogeneity.** Prior work simulates data heterogeneity by partitioning a single image classification dataset (*e.g.*, MNIST (Deng, 2012)) into non-i.i.d. subsets per client. However, such label-skew fails to capture real-world data heterogeneity (Borazjani et al., 2025), where clients tackle different tasks across vision and language domains (Madni et al., 2024). Recent work (Chen et al., 2024a) moves beyond label skew by assigning different VQA datasets to clients, but remains confined to a single task type and single-image inputs. In contrast, our proposed benchmark, DRAKE, spans a broader range of multi-modal tasks, including VQA, visual reasoning, and visual relation, covering single- and multi-image inputs, while also modeling temporal distribution shifts within each client, reflecting the evolving and non-stationary nature of real-world data (Garg et al., 2024).

**Model Heterogeneity.** Federated distillation (FD) transfers knowledge across heterogeneous models by sharing logits on public data. FedMD (Li & Wang, 2019), and PerAda (Xie et al., 2024) average logits from local models, while FedMKT (Fan et al., 2025) uses those with the lowest loss. However, public-client domain gaps limit transferability (Wang et al., 2023), and logit sharing introduces both privacy risks (Lyu et al., 2022) and high computational cost for large models (Malladi et al., 2023).

Recent works address the limitations of FD through direct model aggregation for heterogeneous models under LoRA-based fine-tuning. HETLoRA (Cho et al., 2024) and FLEXLORA (Bai et al., 2024) handle varying LoRA ranks through zero-padding/truncation and SVD-based redistribution, respectively. Although they address the varying rank sizes, both assume (i) identical hidden dimensions and (ii) uniform depths across clients, limiting their applicability to heterogeneous architectures that differ in both dimensions and depths (Yao, 2024). In contrast, our proposed Co-LoRA accommodates both dimensional and depth heterogeneity, enabling more general heterogeneous setups.

## 3 PRELIMINARIES

**Low-Rank Adaptation (LoRA).** LoRA (Hu et al., 2022) assumes that fine-tuning updates lie in a low-rank space. Building on this assumption, LoRA constrains the weight update $\Delta W$ for a pre-trained weight matrix $W_p \in \mathbb{R}^{d_O \times d_I}$ through low rank decomposition using matrices $A \in \mathbb{R}^{r \times d_I}$ and $B \in \mathbb{R}^{d_O \times r}$, where the rank $r \ll \min(d_O, d_I)$. During training, only $A$ and $B$ are updated, while $W_p$ remains frozen. With LoRA, the original output $h_O = W_p h_I \in \mathbb{R}^{d_O}$ for an input hidden state $h_I \in \mathbb{R}^{d_I}$ is modified by incorporating the low-rank update as $h_O = (W_p + \Delta W)h_I = (W_p + BA)h_I$.

**Problem Statement of Personalized Federated Learning.** We consider a PFL setup with $N$ clients, where each client $i \in [N]$ has a local dataset $\mathcal{D}_i$. Reflecting real-world scenarios where data arrives incrementally (Seo et al., 2024), each client receives a continuous stream of samples $(x_1^{(i)}, y_1^{(i)}), (x_2^{(i)}, y_2^{(i)}), \cdots$. Given a set of model architectures $W = \{W_1, \ldots, W_K\}$, each client $i$ selects its local pre-trained model $W_p^{(i)} = V(i)$, based on its hardware constraints, where $V$ is a mapping function $V : \{1, \ldots, N\} \to W$. For efficiency, each client freezes $W_p^{(i)}$ while training only its local LoRA adapter $L_i$. At each round, the server computes the global adapter $G$ by aggregating the local adapters $\{L_1, \ldots, L_N\}$ and sends $G$ back to the clients. Let $f(x; W_p^{(i)}, L_i, G)$ be the model output, and let $\ell$ be the loss function. The empirical loss for client $i$ over its local dataset $\mathcal{D}_i$ is $\mathcal{J}_i(L_i, G) = \frac{1}{|\mathcal{D}_i|} \sum_{(x,y) \in \mathcal{D}_i} \ell(f(x; W_p^{(i)}, L_i, G), y)$. We then formulate the PFL as $\min_{\{L_1, \ldots, L_N\}} \frac{1}{N} \sum_{i=1}^{N} \mathcal{J}_i(L_i, G)$, following Zhang et al. (2021); Tamirisa et al. (2024).

## 4 PROPOSED METHOD

To address the real-world heterogeneities, *i.e.*, data and model heterogeneity, that hinder client collaborations in PFL, we propose **FedMosaic**, illustrated in Fig. 2, comprising of: **RELA** (**REL**evance-guided **A**ggregation) and **Co-LoRA** (**Co**llaborative-**LoRA**). RELA mitigates data heterogeneity by restricting knowledge sharing to local models trained on *related* tasks, thus reducing interference during aggregation (*i.e.*, merging by parameter averaging). Under model heterogeneity, model aggregation becomes infeasible. To enable cross-architecture collaboration, we introduce Co-LoRA, which incorporates shareable modules $P \in \mathbb{R}^{r \times r}, Q \in \mathbb{R}^r$ in LoRA, whose sizes depend only on low-rank size $r$, not on the hidden dimension. We provide a pseudocode in Sec. A.35.

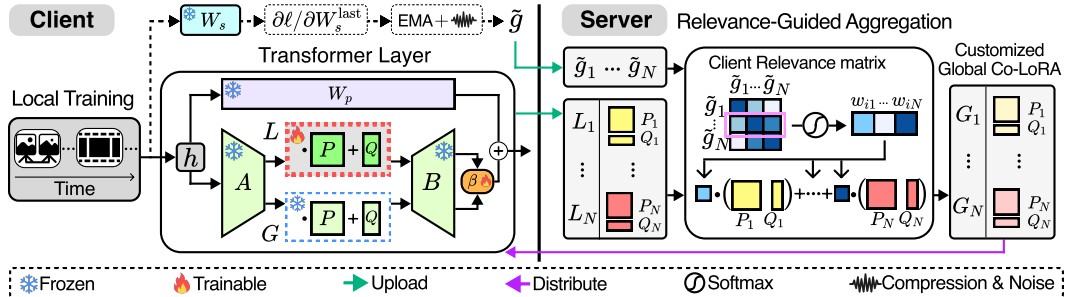

Figure 2: **Overview of proposed FedMosaic.** On every round, the local Co-LoRA $L_i$ fine-tuned during local training and the sanitized last layer gradient $\tilde{g}_i$ are uploaded from the $i_{\text{th}}$ client to server. The last layer gradient is extracted on every $m$ iterations from the small pre-trained model $W_s$, which is then EMA updated and sanitized to $\tilde{g}$. In server, the sanitized gradients $\tilde{g}_i$ are used to measure client task relevance and to build customized global Co-LoRA $G_i$, which is distributed and kept frozen. $h$ and $W_p$ denote the hidden state input and the pre-trained weight, respectively. $\beta$ is a learnable gating parameter that balances the output from the global and local models.

## 4.1 RELEVANCE-GUIDED AGGREGATION

Model aggregation builds a single model that excels in multiple tasks without accessing raw data (Wei et al., 2025). It is thus widely used to construct a shared global model in federated learning. However, naively averaging models trained on different tasks often causes parameter interference (Yadav et al., 2023; Yang et al., 2024b). Recent work shows that models solving similar tasks share more transferable knowledge with fewer conflicts (Gurulingan et al., 2022). Motivated by this, we replace the uniform averaging with a relevance-guided strategy that assigns a higher aggregation weight to clients with closer task relations, providing each client with a customized global model.

**Client-Wise Gradient $g_i$.** To measure task relevance between clients, we calculate the similarity of client-wise gradients. Formally, we calculate $g_i$, the gradient for $i_{\text{th}}$ client as follows:

$$g_i = \mathbb{E}_{z \subset \mathcal{D}_i} \left[ \nabla_{W_s} \ell(z) \right], \tag{1}$$

where $\mathcal{D}_i$ is $i_{\text{th}}$ client's data stream, $\ell$ refers to the loss function, $z \subset \mathcal{D}_i$ is a mini-batch, and $W_s$ is a small-scale frozen pre-trained model $W_s$. For efficiency, we (i) use gradients from a small-scale frozen pre-trained model $W_s$ that provides sufficient representativeness with reduced overhead (Lee et al., 2024), and (ii) compute only the last-layer gradient, as preceding layers' gradients are proportional to it based on the chain rule (Seo et al., 2025). Moreover, we compute the gradient $g_i$ every $m$ batch iterations rather than every batch, thus incurring negligible additional cost, as shown in Sec. A.8. Note that we measure gradients from a frozen pre-trained model, not the actual training model. This is because in heterogeneous PFL, clients train on diverse tasks, and gradient similarities from models trained on heterogeneous data may not capture task similarity (Tang et al., 2020; Evans et al., 2024).

**Decayed Client-Wise Gradient $\hat{g}_i$.** However, $g_i$ may not reflect learned task relevance under shifting data distributions, as it is an expectation over the entire data stream $D_i$ (Eq. 1), *i.e.*, unweighted average of gradients across all time points, ignoring forgetting of the model over time. Consequently, client 1 learning $A \rightarrow B$ and client 2 learning $B \rightarrow A$ yield the same $g_i$, despite retaining different knowledge due to catastrophic forgetting (McCloskey & Cohen, 1989; Ratcliff, 1990). To reflect the model knowledge shifts under distribution shifts, we introduce decayed gradient $\hat{g}_i$, computed using the exponential moving average (EMA) of past gradients, inspired by the exponential decay of knowledge in forgetting (Mahto et al., 2021; Chien et al., 2021; Seo et al., 2025). Formally, for the $i_{\text{th}}$ client, $\hat{g}_i(t)$ with EMA ratio $\alpha$ at timestep $t$ is defined recursively as:

$$\hat{g}_i(t) = (1 - \alpha) \cdot \hat{g}_i(t-1) + \alpha \cdot g_i(t), \tag{2}$$

where $g_i(t) = \nabla_{W_s} \ell(z_t)$ is the gradient vector for the given batch $z_t \subset \mathcal{D}_i$ at timestep $t$.

**Sanitized Client-Wise Gradient $\tilde{g}_i$.** Transmitting EMA-aggregated gradients (*i.e.*, $\hat{g}_i$) rather than per-sample gradients mitigates gradient-based privacy attacks, as gradient mixing (*i.e.*, aggregating) increases resistance to gradient inversion (Mo et al., 2021). We further prevent privacy risks by transmitting sanitized gradient $\tilde{g}_i$ through: (i) adding Gaussian noise $\epsilon$ to $\hat{g}_i$ and (ii) applying gradient compression (*i.e.*, randomly selecting only $N_s\%$ of the gradient vector dimensions from $\hat{g}_i \in \mathbb{R}^d$), as

randomly sampled dimensions can approximate full-gradient distributions (Li et al., 2023b) while making gradient inversion substantially more difficult than using full gradients (Zhu et al., 2019) and simultaneously reducing transmission costs (detailed in Sec. A.9). Formally, $\tilde{g}_i$ is defined as:

$$\tilde{g}_i(t) = \mathbf{M} \odot (\hat{g}_i(t) + \mu\epsilon), \quad \epsilon \sim \mathcal{N}(0, \boldsymbol{I}_d) \tag{3}$$

where $\mathbf{M} \in \{0,1\}^d$ is the binary mask for random subsampling and $\mu$ denotes the noise scale.

Using the sanitized client-wise gradients $\{\tilde{g}_1, \ldots, \tilde{g}_N\}$ from $N$ clients, we construct a client-relevence matrix $S \in \mathbb{R}^{N \times N}$, where $S_{ij} = \cos(\tilde{g}_i, \tilde{g}_j)$. The customized global module for the $i_{\text{th}}$ client, $G_i$, is then constructed by weighted aggregation of local modules $\mathcal{L} = \{L_1, \ldots, L_N\}$ as follows:

$$G_i = \sum_{j=1}^{N} w_{ij} \, L_j, \quad w_{ij} = \frac{e^{\cos(\tilde{g}_i, \tilde{g}_j)/\tau}}{\sum_{n=1}^{N} e^{\cos(\tilde{g}_i, \tilde{g}_n)/\tau}}, \tag{4}$$

where $\tau$ denotes the softmax temperature and $\cos(\cdot, \cdot)$ denotes cosine similarity.

## 4.2 CO-LORA

LoRA matrices $A \in \mathbb{R}^{r \times d_I}$ and $B \in \mathbb{R}^{d_O \times r}$ depend on model-specific hidden dimensions $d_I$ and $d_O$, preventing direct aggregation across different models. To enable knowledge sharing among heterogeneous architectures, we introduce Co-LoRA, which inserts dimension-invariant modules $P \in \mathbb{R}^{r \times r}$ and $Q \in \mathbb{R}^r$ between $A$ and $B$. Their dimensions depend only on the low-rank $r$, making them shareable across heterogeneous models. We illustrate a comparison of Co-LoRA with conventional LoRA in Fig. 3. Formally, Co-LoRA outputs $h_O$ as:

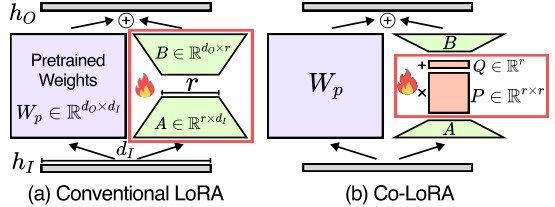

(a) Conventional LoRA    (b) Co-LoRA

Figure 3: **Illustration of (a) Conventional LoRA and (b) Co-LoRA.** While $A$ and $B$ are trainable in conventional LoRA, Co-LoRA freezes both, updating only the dimension-invariant modules $P \in \mathbb{R}^{r \times r}$ and $Q \in \mathbb{R}^r$ during training.

$$h_O = W_p h_I + B(PAh_I + Q), \tag{5}$$

for input hidden states $h_I$ and pre-trained weight $W_p$. After local training on each client, the $P$ and $Q$ modules are aggregated and shared between heterogeneous clients for knowledge sharing.

Although $P$ and $Q$ are shareable across heterogeneous architectures having different dimensions, two main challenges hinder their aggregation: (i) depth heterogeneity - heterogeneous architectures often differ in depth, making it non-trivial to determine which layers across models should be aggregated; and (ii) weight misalignment - interpolating weights with different optimization trajectories can degrade performance (Jordan et al., 2023; Stoica et al., 2025). We address these challenges with two strategies: (i) block-wise aggregation, which aligns layers at the same relative depth, and (ii) weight alignment, ensuring heterogeneous models share the same initialization. For simplicity, we consider aggregating two heterogeneous models, though it naturally extends to multiple heterogeneous models.

### 4.2.1 BLOCK-WISE AGGREGATION

To decide which layers of heterogeneous models should share knowledge, we measure layer-wise representation alignment between two MLLMs, $W_i$ and $W_j$, of different depths using CKA (Kornblith et al., 2019). We observe high similarity between layers at similar relative depths, *i.e.*, approximately linear alignment (Fig. 4). See Sec. A.10 for broader analysis beyond Llama-1B/3B. Accordingly, we divide each model into $N_B$ blocks linearly and attach a Co-LoRA to each block's final layer, enabling cross-model sharing at relevant depths. The attachment layer index $I_k$ of the $k$-th Co-LoRA in a $|W|$-layer model is defined as:

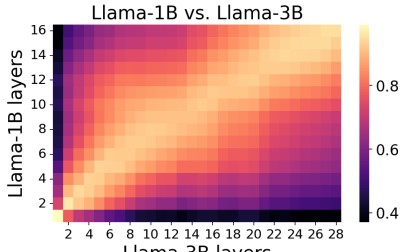

Figure 4: **Layer-wise similarity between Llama-1B/3B measured with CKA.** The diagonal brightest band shows the strongest alignment between layers at similar relative depths (*e.g.*, Llama-1B layer 8 - Llama-3B layer 14).

$$I_k = \begin{cases} k \cdot \left\lfloor \frac{|W|}{N_B} \right\rfloor & \text{for} \quad k = 1, 2, \ldots, N_B - 1 \\ |W| & \text{for} \quad k = N_B. \end{cases} \tag{6}$$

### 4.2.2 Weight Alignment in Co-LoRA

We then align the $N_B$ number of Co-LoRA modules at layer indices $\mathcal{I}_i = \{I_1^i, \ldots, I_{N_B}^i\}$ in model $M_i$ and $\mathcal{I}_j = \{I_1^j, \ldots, I_{N_B}^j\}$ in model $M_j$. For simplicity, we describe the alignment for a single pair $X = (\{P_i, Q_i, A_i, B_i\}, \{P_j, Q_j, A_j, B_j\})$, which is generalized across all $N_B$ pairs.

Motivated by findings that models fine-tuned from the same initialization share optimization paths and can be merged without interference (Wortsman et al., 2022a;b; Yadav et al., 2023), we set the pair $X$ to share a common initialization. Dimension-invariant modules, $P_i, P_j \in \mathbb{R}^{r \times r}$ and $Q_i, Q_j \in \mathbb{R}^r$, can share the same weight directly, but dimension-dependent modules, $A_i \in \mathbb{R}^{r \times d_I^{(i)}}$, $A_j \in \mathbb{R}^{r \times d_I^{(j)}}$ and $B_i \in \mathbb{R}^{d_O^{(i)} \times r}$, $B_j \in \mathbb{R}^{d_O^{(j)} \times r}$, cannot due to dimension mismatches. Although only $P$ and $Q$ are shared between heterogeneous models $M_i$ and $M_j$, $A$ and $B$ should also be aligned, as they affect the output $h_O$ (Eq. 5) and optimization trajectories. Therefore, we align $A_i, A_j$ and $B_i, B_j$, ensuring aligned representations for the same input, even in parameter spaces with different dimensions.

**Aligning $A$ Matrices.** We first align $A_i \in \mathbb{R}^{r \times d_I^{(i)}}$ and $A_j \in \mathbb{R}^{r \times d_I^{(j)}}$ using L2 loss in the shared $r$-dimensional space with publicly available data $\mathcal{D}_p$. For $\mathcal{D}_p$, we use a subset of the MLLMs' pre-training data, as detailed in Sec. A.17. In particular, we use the smaller model $A_i$ (where $d_I^{(i)} < d_I^{(j)}$) as the pivot model by freezing it and updating the larger model $A_j$ to minimize the L2 loss on $r$-dimensional representations obtained from $\mathcal{D}_p$, as $\min_{A_j} \frac{1}{|\mathcal{D}_p|} \sum_{(x,y) \in \mathcal{D}_p} \|(A_i(x) - A_j(x)\|_2^2$, where $A(x)$ denotes $r$-dimensional feature extracted by $A$ for input $x$. Please see Sec. A.29 for a detailed justification of our choice of a smaller model as the pivot model.

**Aligning $B$ Matrices.** We then align $B_i$ and $B_j$. Since their output dimensions differ ($d_O^{(i)} \neq d_O^{(j)}$), L2 loss is inapplicable. Instead, we employ canonical correlation analysis (CCA) (De Bie et al., 2005), which finds projection matrices maximizing the correlation between two feature sets. Using $m$ public data $\{(x_k, y_k)\}_{k=1}^m \subset D_p$, we extract output features $H_i \in \mathbb{R}^{m \times d_O^{(i)}}$ and $H_j \in \mathbb{R}^{m \times d_O^{(j)}}$ from $B_i$ and $B_j$, then find projection matrices $\Pi_i \in \mathbb{R}^{d_O^{(i)} \times r}$ and $\Pi_j \in \mathbb{R}^{d_O^{(j)} \times r}$, which project them to the maximally correlated space of dimension $r$ (i.e., $H_i \Pi_i \approx H_j \Pi_j$). We project $B_i$ to the shared maximally correlated space, i.e., $(\Pi_i)^T \cdot B_i$, and subsequently bring it to $B_j$'s space by inverse projection, as $B_j = (\Pi_j^\dagger)^T \cdot (\Pi_i)^T \cdot B_i$, where $\Pi_j^\dagger$ denotes the pseudo-inverse of $\Pi_j$.

During the alignment of $A$ and $B$, we enforce orthogonality in $A$ and $B$ to maximize the expressive capacity (i.e., span) of Co-LoRA weight updates, following Theorem 1. We provide details of enforcing orthogonality in Sec. A.5 and the proof of Theorem 1 in Sec. A.2.

**Theorem 1.** *If the column vectors of matrix $B \in \mathbb{R}^{d_O \times r}$ are orthogonal and the row vectors of matrix $A \in \mathbb{R}^{r \times d_I}$ are orthogonal, then the span of the weight update space of Co-LoRA, $\text{span}\{\Delta W\}$, has $r^2$ dimension, which is the maximum possible dimension under frozen $B$ and $A$.*

In addition, our method does not incur much computational cost since (i) Co-LoRA aligning is performed *once before federated training* to establish a shared initialization, (ii) it is required only for *heterogeneous model-type pairs* (since some clients may share the same architecture). After aligning $A$ and $B$, we update shareable modules $P$ and $Q$, while freezing $A$ and $B$ during local training to preserve alignment. We provide theoretical justification for this freezing approach in Sec. A.1. This freezing design also reduces the communication cost of Co-LoRA relative to conventional LoRA, as it requires communication of only $P, Q$ modules. See Sec. A.9 for details of communication costs.

After receiving the aggregated Co-LoRA (i.e., global Co-LoRA) by RELA at each communication round, clients freeze it during training to preserve global knowledge and update only the local model for personalization. Specifically, at the $l_\text{th}$ layer, given an input hidden state $h_I$, we combine the output of the local LoRA (i.e., $h_L$), the frozen global LoRA (i.e., $h_G$), and the pre-trained weights (i.e., $W_p h_I$), by adaptively balancing them using a learnable gating parameter $\beta$. With sigmoid-normalized balancing parameter $\tilde{\beta} = \sigma(\beta)$, the output hidden state $h_O$ is computed as follows:

$$h_O = W_p h_I + (1 - \tilde{\beta}) h_L + \tilde{\beta} h_G. \tag{7}$$

| Dataset | Multi-Data Sources | Distribution Shifts | Multi-Image Support | Multi-Modalities | Unseen Evaluation |
|---|---|---|---|---|---|
| Split-CIFAR (ICML 2021) | ✗ | ✓ | ✗ | ✗ | ✗ |
| NonIID-50 (ICML 2021) | ✓ | ✓ | ✗ | ✗ | ✗ |
| LEAF-FCL (ICLR 2023) | ✓ | ✓ | ✗ | ✗ | ✗ |
| MNIST-Shuffle (ICLR 2024) | ✗ | ✓ | ✗ | ✗ | ✗ |
| HC-FMTL (CVPR 2024) | ✓ | ✗ | ✗ | ✗ | ✗ |
| Fed-SNI (NeurIPSW 2023) | ✓ | ✗ | - | ✗ | ✗ |
| FEDLEGAL (ACL 2023) | ✓ | ✗ | - | ✗ | ✗ |
| Fed-Aya (NeurIPS 2024) | ✗ | ✗ | - | ✗ | ✗ |
| Fed-FLAN (NeurIPS 2024) | ✓ | ✗ | - | ✗ | ✗ |
| HFLB (AAAI 2024) | ✓ | ✗ | ✗ | ✓ | ✗ |
| DRAKE (**Ours**) | ✓ | ✓ | ✓ | ✓ | ✓ |

Table 1: **Comparison of FL benchmarks across key dimensions:** Multi-Data Sources (using diverse datasets vs. non-i.i.d. splits of a single dataset), Distribution Shifts (evolving client data distributions), Multi-Image Support (handling multiple images per input), and Unseen Evaluation (testing on tasks unseen during training). See Sec. A.28 for the detailed comparisons.

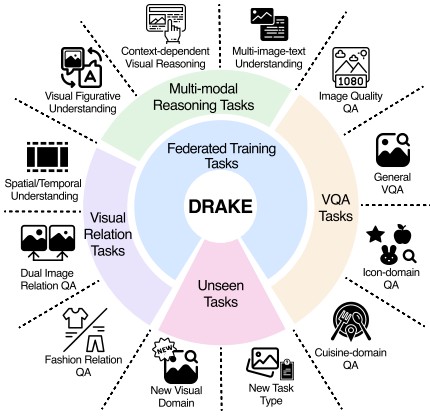

Figure 5: Overview of DRAKE.

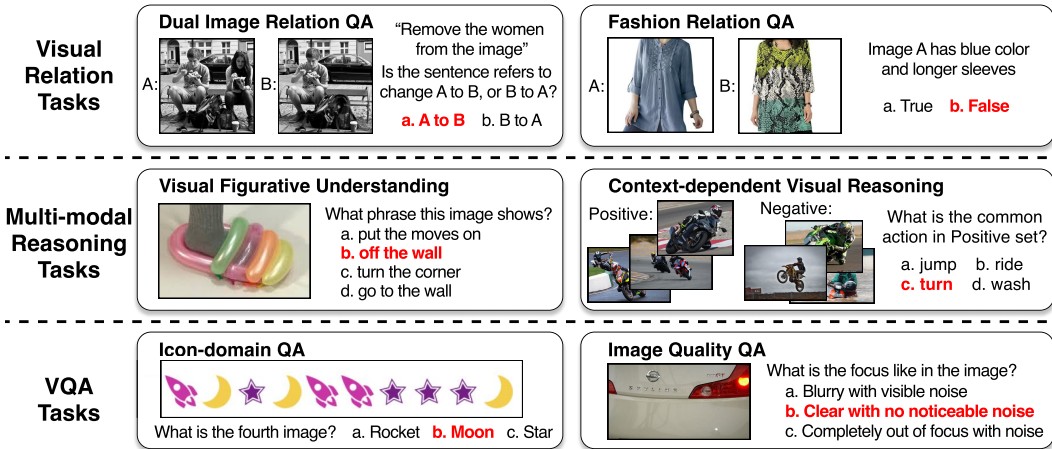

Figure 6: **Data samples of DRAKE.**

# 5 DRAKE BENCHMARK

We propose a novel multi-modal FL benchmark, called DRAKE, with three key merits: (i) **Task heterogeneity**: Each client handles distinct multi-modal tasks (*e.g.*, visual reasoning, VQA), while existing benchmarks merely assign non-i.i.d. subsets from a single dataset. (ii) **Dynamic distribution**: Client datasets contain progressive tasks (*e.g.*, encountering new visual concepts), simulating real-world temporal distribution shifts. To the best of our knowledge, DRAKE is the first benchmark supporting multi-modal federated learning under distribution shifts. (iii) **Generalizability Evaluation**: DRAKE incorporates unseen task data to evaluate the generalizability of FL models. We summarize the comparison with existing FL benchmarks in Tab. 1. Specifically, we curate DRAKE using large-scale multi-modal benchmarks that differ in distribution or properties from LLaVA's pre-training data and provide clear metadata or instructions, enabling task-wise splits for emulating distribution shifts. DRAKE consists of three training task subgroups, *i.e.*, VQA, visual relation, and multi-modal reasoning, and two unseen task subgroups, comprising 40 heterogeneous tasks sourced from 19 different multi-modal datasets, totaling 375k images and 274k questions, illustrated in Fig. 5. We provide data samples of each sub-group in Fig. 6. Please refer to Sec. A.27 for more details.

**Visual Relation Tasks.** Visual relation tasks focus on capturing relationships among visual objects. They comprise 12 distinct tasks grouped into three splits: *fashion-relation*, *dual-image relation*, and *spatial/temporal relation*. The fashion-relation split targets fine-grained attribute understanding in clothing images, the dual-image relation split evaluates comparative reasoning across separate images, and the spatial/temporal split assesses understanding of positional and temporal dependencies.

**Multi-modal Reasoning Tasks.** Multi-modal reasoning tasks require integrating visual information with language and common-sense knowledge. We group 12 tasks into three splits: *visual figurative understanding*, which inspects interpretation of non-literal, figurative, and idiomatic descriptions of images, *context-dependent visual reasoning*, which tests discrimination among highly similar yet

contrastive visual sets, and *multi-image-text understanding*, which evaluates reasoning over multiple images or image-text pairs provided as contextual evidence to understand.

**VQA Tasks.** The *VQA* sub-group contains 16 tasks that differ from LLaVA's pre-training data and is organized into four splits: *image quality*, *icon-domain*, *cuisine-domain*, and *general VQA*. The image quality split evaluates photographic properties such as focus or brightness. The icon-domain split requires interpreting iconic images. The cuisine-domain split involves fine-grained food recognition combined with geographic or cultural knowledge. The general VQA split covers standard settings, including chart and diagram understanding, visual instance identification, and counting.

**Unseen Tasks.** To assess generalization capability, we add 7 additional tasks that are disjoint from the clients' tasks in DRAKE. These benchmarks are intentionally challenging and require substantial training, spanning novel task types and novel visual domains with familiar task formats.

## 6 EXPERIMENTS

### 6.1 SETUPS

**Models.** To simulate model heterogeneity in federated MLLM training, we employ LLaVA-1.5 (Liu et al., 2023b) variants. For LLM of LLaVA, we employ various sizes of Llama-3 (Grattafiori et al., 2024) (Llama-3.2-1B, Llama-3.2-3B, Llama-3.1-8B) and Qwen-2.5 (Yang et al., 2024a) (Qwen2.5-0.5B, Qwen2.5-1.5B, Qwen2.5-3B). We use the Llama-3 series for text-only benchmarks.

**Metrics.** We report $A_{last}$, the accuracy at the end of training, and $A_{\text{AUC}}$ (Koh et al., 2022), which computes the area under the accuracy curve by measuring accuracy at each evaluation period to capture intermediate performance. All experiments use five rounds of evaluation intervals and are averaged over three different random seeds, with standard deviations reported.

**Benchmarks.** We evaluate FedMosaic on our proposed DRAKE and the benchmark used in Chen et al. (2024a), which we refer to as the Heterogeneous Federated Learning Benchmark (HFLB). Our evaluation covers PFL-Static (*i.e.*, i.i.d. intra-client data distributions) and the more realistic PFL-Dynamic setup (*i.e.*, intra-client distribution shifts with four tasks). We also evaluate on text-only PFL benchmarks, *i.e.*, Fed-Scope (Kuang et al., 2024), Fed-Aya (Singh et al., 2024), and Fed-LLM-Large, which combines Fed-LLM (Ye et al., 2024) and Fed-FLAN (Long et al., 2024).

**Baselines.** We compare FedMosaic with SOTA PFL methods. We also compare with supervised fine-tuning (SFT), where clients train independently without knowledge sharing and often outperforming PFL methods under data heterogeneity (Ghari & Shen, 2024).

See Sec. A.6 for the details of benchmarks, baselines and implementation details.

### 6.2 QUANTITATIVE ANALYSIS

In all experiments, we evaluate each client's model on its own task (*i.e.*, 'Self') and on other clinets' tasks (*i.e.*, 'Others'), reporting average performance across clients. 'Self' performance shows personalization, while 'Others' indicates generalizability. Although personalization is PFL's main goal, generalization ability is also crucial for continual personalization, as it enables rapid adaptation to new tasks in the future during training (Finn et al., 2017; Rao et al., 2023).

**Heterogeneous Multi-Modal Clients.** We evaluate FedMosaic in multi-modal heterogeneous PFL-Dynamic and -Static setups, where clients employ different architectures (LLaVA-Llama3-1B or 3B). As shown in Tab. 2 and Tab. 3, FedMosaic consistently outperforms baselines on both clients' own tasks ('Self') and others' tasks ('Others') under static and dynamic distributions. HFLB's single VQA task with single-image inputs converges faster than DRAKE, yielding smaller gaps among baselines.

Note that Tab. 2 reports the average performance across all clients. Local client training (SFT) can suffice for simpler tasks (Mosbach et al., 2021; Woźniak et al., 2024), *e.g.*, single-image VQA (clients 2, 3 in Tab. 5), making average improvements appear small. In contrast, FedMosaic significantly enhances personalization for complex multi-image tasks (clients 5, 7, 9 in Tab. 5) through effective knowledge sharing. See Sec.A.16 for additional per-client results and Sec. A.27 for task descriptions.

As shown in Tab.5, not only do clients using smaller models (*i.e.*, LLaVA-Llama3.2-1B) benefit from knowledge sharing through PFL, but larger models (*i.e.*, LLaVA-Llama3.2-3B) also see significant

| | DRAKE-Dynamic | | | | HFLB-Dynamic | | | |
| --- | --- | --- | --- | --- | --- | --- | --- | --- |
| | Self | | Others | | Self | | Others | |
| Method | $A_{last}$ ↑ | $A_{\text{AUC}}$ ↑ | $A_{last}$ ↑ | $A_{\text{AUC}}$ ↑ | $A_{last}$ ↑ | $A_{\text{AUC}}$ ↑ | $A_{last}$ ↑ | $A_{\text{AUC}}$ ↑ |
| SFT | 65.79±0.20 | 57.62±0.26 | 47.66±0.13 | 46.67±0.11 | 79.99±0.66 | 77.14±0.35 | 61.66±0.56 | 61.43±0.14 |
| DITTO (ICML 2021) | 59.91±0.18 | 54.16±0.06 | 47.45±0.36 | 46.70±0.10 | 79.10±0.11 | 76.05±0.08 | 61.86±0.82 | 61.43±0.08 |
| FedSim (ICML 2022) | 63.98±1.19 | 56.42±0.70 | 47.04±0.16 | 46.15±0.17 | 79.90±0.09 | 76.47±0.01 | 59.92±0.54 | 59.56±0.13 |
| FedIT (ICASSP 2024) | 66.11±0.27 | 57.86±0.27 | 47.63±0.16 | 46.62±0.04 | 79.87±0.50 | 77.04±0.38 | 61.73±0.54 | 61.45±0.18 |
| TAKFL (NeurIPS 2024) | 64.54±0.85 | 56.19±0.98 | 47.38±0.06 | 46.47±0.14 | 79.76±0.06 | 76.86±0.04 | 61.42±0.44 | 61.22±0.10 |
| FedDPA (NeurIPS 2024) | 63.34±0.23 | 55.74±0.38 | 47.61±0.21 | 46.64±0.20 | 70.06±0.24 | 77.10±0.36 | 61.58±0.27 | 61.19±0.09 |
| FedDAT (AAAI 2024) | 58.47±1.10 | 54.38±0.28 | 48.91±0.42 | 47.78±0.18 | 79.61±0.42 | 77.43±1.02 | 64.15±0.08 | 64.51±0.13 |
| PerAda (CVPR 2024) | 59.75±1.06 | 54.48±0.13 | 47.30±0.45 | 46.72±0.03 | 79.03±0.01 | 75.92±0.01 | 61.76±0.80 | 61.39±0.06 |
| FedMKT (COLING 2025) | 61.38±0.18 | 55.45±0.31 | 47.50±0.07 | 46.68±0.08 | 79.48±0.01 | 76.57±0.06 | 61.54±0.61 | 61.41±0.19 |
| FedMosaic (**Ours**) | **67.86±0.51** | **59.83±0.16** | **51.16±0.04** | **49.36±0.08** | **80.80±0.26** | **78.43±0.14** | **67.07±0.25** | **66.02±0.16** |

Table 2: **Quantitative comparison in heterogeneous PFL.** 'Self' denotes evaluation on a client's own data, while 'Others' denotes evaluation on data from other clients. In DRAKE, 4 clients use LLaVA-Llama3.2-1B and 6 clients use LLaVA-Llama3.2-3B, while in HFLB, 3 clients use LLaVA-Llama3.2-1B and 6 clients use LLaVA-Llama3.2-3B. SFT refers to supervised fine-tuning on each client's data without cross-client knowledge sharing.

| | Self | | Others | |
| --- | --- | --- | --- | --- |
| Method | $A_{last}$ ↑ | $A_{\text{AUC}}$ ↑ | $A_{last}$ ↑ | $A_{\text{AUC}}$ ↑ |
| SFT | 68.50±0.42 | 63.84±0.27 | 47.91±0.39 | 47.56±0.23 |
| DITTO | 63.67±1.50 | 59.17±1.25 | 48.22±0.09 | 47.65±0.03 |
| FedSim | 66.75±0.56 | 61.70±0.60 | 46.93±0.25 | 46.76±0.02 |
| FedIT | 68.71±0.04 | 63.89±0.55 | 47.91±0.22 | 47.60±0.17 |
| TAKFL | 67.28±0.01 | 62.42±0.36 | 47.55±0.23 | 47.51±0.07 |
| FedDPA | 66.09±1.51 | 61.36±0.14 | 47.93±0.30 | 47.69±0.11 |
| FedDAT | 61.28±0.07 | 57.92±0.02 | 49.37±0.02 | 48.78±0.04 |
| PerAda | 63.67±1.01 | 59.03±1.19 | 48.13±0.03 | 47.60±0.05 |
| FedMKT | 65.81±1.09 | 60.37±1.16 | 47.19±0.86 | 46.44±1.49 |
| FedMosaic | **70.10±0.53** | **64.64±0.40** | **51.57±0.24** | **50.42±0.12** |

| | Self | | Others | |
| --- | --- | --- | --- | --- |
| Method | $A_{last}$ ↑ | $A_{\text{AUC}}$ ↑ | $A_{last}$ ↑ | $A_{\text{AUC}}$ ↑ |
| SFT | 68.60±0.57 | 61.33±1.03 | 48.34±0.18 | 47.53±0.22 |
| DITTO | 66.77±0.96 | 60.67±0.77 | 49.04±0.63 | 48.33±0.53 |
| FedSim | 66.65±0.26 | 59.49±0.17 | 46.77±0.29 | 46.42±0.10 |
| FedIT | 68.72±0.97 | 60.88±0.81 | 48.22±0.17 | 47.25±0.22 |
| TAKFL | 67.77±0.46 | 60.13±0.32 | 48.18±0.07 | 47.51±0.14 |
| FedDPA | 67.38±0.58 | 60.20±0.40 | 48.40±0.22 | 47.32±0.33 |
| FedDAT | 66.08±0.95 | 60.00±0.22 | 50.05±0.02 | 49.10±0.11 |
| PerAda | 64.86±0.51 | 58.73±0.48 | 47.89±0.70 | 47.45±0.45 |
| FedMKT | 65.44±0.91 | 59.18±0.62 | 48.09±0.41 | 47.51±0.18 |
| FedMosaic | **70.67±0.61** | **63.51±0.40** | **52.31±0.15** | **50.60±0.26** |

Table 3: **Quantitative comparison in PFL on DRAKE-static.** 4 clients use LLaVA-Llama3.2-1B, 6 clients use LLaVA-Llama3.2-3B.

Table 4: **Quantitative comparison in cross-family PFL on DRAKE-dynamic.** 3 clients use LLaVA-Llama3.2-3B, 4 use LLaVA-Qwen2.5-1.5B, 3 use LLaVA-Qwen2.5-3B.

| | Self $A_{last}$ / $A_{\text{AUC}}$ | | | | | | | | | |
| --- | --- | --- | --- | --- | --- | --- | --- | --- | --- | --- |
| | LLaVA-Llama3.2-**1B** | | | | LLaVA-Llama3.2-**3B** | | | | | |
| Method | Client 1 | Client 2 | Client 3 | Client 4 | Client 5 | Client 6 | Client 7 | Client 8 | Client 9 | Client 10 |
| SFT | 76.45 / 59.87 | 58.10 / 54.35 | 66.63 / 59.57 | 62.85 / 53.16 | 62.00 / 61.51 | 76.56 / 66.88 | 62.59 / 54.63 | 68.54 / 59.65 | 65.92 / 51.93 | 58.30 / 54.61 |
| FedMosaic | **77.42 / 60.45** | **58.50 / 55.26** | **66.71 / 59.60** | **63.13 / 55.84** | **68.87 / 63.60** | **77.61 / 69.80** | **69.19 / 56.66** | **70.54 / 58.93** | **70.68 / 54.09** | **59.21 / 55.32** |
| *Gain* | +0.97 / +0.57 | +0.40 / +0.91 | +0.08 / +0.04 | +0.27 / +2.67 | +6.86 / +2.09 | +1.05 / +2.93 | +6.61 / +2.03 | +2.00 / -0.72 | +4.76 / +2.16 | +0.91 / +0.71 |

Table 5: **Per-client 'Self' performance of FedMosaic vs. SFT in a heterogeneous PFL-Dynamic setup on DRAKE.** The *Gain* row shows positive gain of FedMosaic in blue, negative in red against SFT. Clients 1-4 use LLaVA-Llama3.2-1B, while Clients 5-10 use LLaVA-Llama3.2-3B.

gains. We attribute this to (i) RELA accurately measuring task relevance under distribution shifts, reducing interference during aggregation, and (ii) Co-LoRA enabling effective knowledge transfer between heterogeneous architectures, allowing smaller and larger models to mutually benefit.

We also emphasize that FedMosaic significantly outperforms baselines in generalization ('Others'), which is crucial under distribution shifts where new, unseen tasks continuously emerge. This enhanced generalizability enables faster adaptation and accelerates future personalization, as shown in Fig. 7.

**Cross-family Heterogeneity.** Beyond varying model sizes within the same family (*e.g.*, LLaVA-Llama-1B/3B), we further simulate cross-family heterogeneity (*e.g.*, Qwen- *vs.* Llama-based MLLMs). Tab. 4 shows FedMosaic consistently outperforms baselines under cross-family heterogeneity, highlighting its generality. See Sec. A.14 for additional cross-family experiments.

**Fast Adaptation Evaluation.** In real-world scenarios, unseen tasks continuously emerge, making rapid adaptation crucial for future personalization. We evaluate adaptation speed by initializing models with each PFL method's aggregated model and fine-tuning on unseen tasks for 200 iterations. Fig. 7 shows that while sufficient training eventually yields similarly high performance regardless of initialization, models initialized with FedMosaic achieve high performance within few steps, highlighting enhanced generalizability through effective knowledge sharing in heterogeneous PFL.

**Large-Scale Evaluation of Heterogeneous PFL in LLMs**. Beyond multi-modal PFL, we evaluate FedMosaic in heterogeneous PFL for LLM training (*i.e.*, text-only NLP domain). Tab. 6 shows FedMosaic significantly outperforms baselines in both 'Self' and 'Others', consistent with our

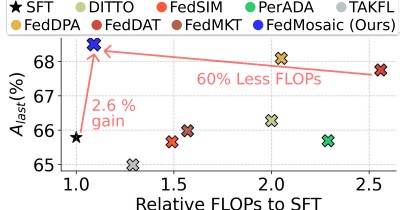

Figure 7: **Comparison of adaptation speed.** We use DRAKE's unseen tasks as downstream tasks. Random init starts from randomly initialized models, while other baselines are initialized from aggregated local models trained on DRAKE using each respective FL baseline.

| | Self | | Others | |
|---|---|---|---|---|
| Method | $A_{last}$ ↑ | $A_{\text{AUC}}$ ↑ | $A_{last}$ ↑ | $A_{\text{AUC}}$ ↑ |
| SFT | 18.03±0.52 | 17.11±0.56 | 14.18±0.69 | 13.70±0.35 |
| FedSim (ICML 2022) | 16.29±0.46 | 16.57±0.26 | 12.62±0.08 | 12.60±0.10 |
| FedIT (ICASSP 2024) | 17.10±0.20 | 16.83±0.37 | 14.03±0.49 | 13.68±0.32 |
| TAKFL (NeurIPS 2024) | 14.04±0.21 | 16.37±0.82 | 11.36±0.13 | 12.57±0.06 |
| FedDPA (NeurIPS 2024) | 17.60±0.25 | 16.66±0.42 | 13.98±0.05 | 13.66±0.07 |
| FedDAT (AAAI 2024) | 17.06±0.13 | 15.83±0.03 | 13.72±0.04 | 13.02±0.01 |
| PerAda (CVPR 2024) | 18.83±0.80 | 17.32±0.85 | 14.66±0.05 | 13.79±0.05 |
| FedMKT (COLING 2025) | 17.34±0.14 | 17.03±0.08 | 14.38±0.01 | 13.89±0.02 |
| FedMosaic (**Ours**) | **20.87±0.13** | **19.07±0.30** | **15.71±0.14** | **14.77±0.11** |

Table 6: **Large-Scale Experiments with 52 heterogeneous LLMs (Llama-1B/3B) on Fed-LLM-Large.**

Figure 8: **Accuracy and relative FLOPs in DRAKE-Dynamic.**

| | DRAKE | | | | Fed-Scope | | | |
|---|---|---|---|---|---|---|---|---|
| | Self | | Others | | Self | | Others | |
| Method | $A_{last}$ ↑ | $A_{\text{AUC}}$ ↑ | $A_{last}$ ↑ | $A_{\text{AUC}}$ ↑ | $A_{last}$ ↑ | $A_{\text{AUC}}$ ↑ | $A_{last}$ ↑ | $A_{\text{AUC}}$ ↑ |
| Vanilla | 66.10±0.27 | 57.87±0.27 | 47.63±0.16 | 46.62±0.04 | 25.06±1.36 | 28.89±0.66 | 25.86±0.94 | 27.94±0.44 |
| (+) Co-LoRA | 66.99±0.77 | 58.39±0.38 | **51.33±0.06** | 48.78±0.03 | 28.47±0.52 | 31.64±0.21 | 30.85±1.37 | 32.27±0.82 |
| (+) Co-LoRA &RELA (**Ours**) | **67.86±0.51** | **59.83±0.16** | 51.16±0.04 | **49.36±0.08** | **30.58±0.84** | **32.68±0.64** | **31.01±1.46** | **32.50±0.62** |

Table 7: **Ablations for proposed components of FedMosaic in heterogeneous PFL.** 'Vanilla' aggregates homogeneous local models within each model type by averaging them with equal weights.

**MLLM-FL benchmark results.** This experiment involves 52 clients, highlighting the scalability of FedMosaic to large client populations and its applicability in real-world deployments.

**Computation and Communication Cost Analysis.** We compare accuracy and relative FLOPs in Fig. 8. FedMosaic demonstrates higher personalization performance with computation comparable to SFT. See Sec. A.8 and Sec. A.9 for detailed analyses of computation and communication costs.

**Additional Experiments.** Additional results include: heterogeneous PFL with LLaVA-Llama3.2-1B/3B/8B (Sec.A.11), LLaVA-Qwen2.5 variants (Sec.A.13), text-only benchmarks (Sec. A.15), homogeneous PFL setup (Sec. A.12), extended fast adaptation evaluations (Sec. A.23), comparison of RELA with similarity-aware aggregation methods (Sec. A.22), and client model selection details (Sec. A.3) and configurations (Sec. A.7). We further provide hyperparameter analysis Sec. A.18), effect of weight alignment in Co-LoRA (Sec. A.21), effect of client model assignment (Sec. A.4), effect of decayed client-wise gradient $\hat{g}$ in RELA (Sec. A.19), and limitations (Sec. A.33).

**Ablation Study.** We ablate FedMosaic to investigate each proposed component's benefit, and summarize the results in Tab. 7. Our observations indicate that each component significantly enhances both the personalization ('Self') and the generalization ('Others') performance. Specifically, Co-LoRA enhances generalization by transferring knowledge across heterogeneous architectures, while RELA improves personalization by promoting knowledge sharing among relevant-task models only.

# 7 CONCLUSION

We address both data heterogeneity and model heterogeneity in PFL while prior works tackle only one challenge or both under simplified assumptions. For that, we propose FedMosaic, which handles both forms of heterogeneity through two components: RELA and Co-LoRA. Moreover, to better reflect real-world data heterogeneity in PFL scenarios, we introduce DRAKE, a comprehensive multi-modal FL benchmark capturing task heterogeneity and distribution shifts. Extensive evaluation shows FedMosaic achieves superior personalization compared to local training, while improving generalizability and few-shot adaptation capabilities essential for future personalization.

## ETHICS STATEMENT

We propose a better learning scheme for heterogeneous federated learning for realistic federated learning scenarios. While the authors do not explicitly aim for this, the increasing adoption of deep learning models in real-world contexts with streaming data could potentially raise concerns, such as inadvertently introducing biases or discrimination. We note that we are committed to implementing all feasible precautions to avert such consequences, as they are unequivocally contrary to our intentions.

## REPRODUCIBILITY STATEMENT

To further facilitate the reproduction, we provide open-source implementations of our proposed method (Sec. 4), benchmark (Sec. 5), along with data splits and baseline models used in our experiments (Sec. 6), available at https://github.com/snumprlab/fedmosaic.

## THE USE OF LARGE LANGUAGE MODELS

We use large language models (LLMs) to support labor-intensive and mistake-prone work. Specifically, we use LLMs (*e.g.*, GPT-4) to categorize samples from a single dataset into multiple tasks based on keywords or topics, to assess generation quality in specific benchmarks using LLMs, and to detect grammatical errors during writing.

## ACKNOWLEDGMENT

This work was supported by the European Research Council (ERC) under the European Union's Horizon 2020 research and innovation programme (Grant Agreement No. 101021347), the IITP grants (RS-2022-II220077, RS-2022-II220113, RS-2022-II220959, RS-2022-II220871, RS-2021-II211343 (SNU AI), RS-2025-25442338 (AI Star Fellowship-SNU), RS-2025-25442569, RS-2024-00437633) funded by the Korea government (MSIT), grants (RS-2025-25462891 (US-KOR BARI), RS-2025-25453780) funded by MOTIR, a grant of Korean ARPA-H Project through the Korea Health Industry Development Institute (KHIDI), funded by the Ministry of Health & Welfare, Republic of Korea (RS-2025-25424639), and the BK21 FOUR program, SNU in 2025.

We also acknowledge the EuroHPC Joint Undertaking for awarding this project access to the EuroHPC supercomputers MareNostrum5 at BSC, Spain; LEONARDO at CINECA, Italy; VEGA at IZUM, Slovenia; Karolina at IT4Innovations, Czech Republic; MeluXina at LuxProvide, Luxembourg; Discoverer at Sofia Tech Park, Bulgaria; and Deucalion at Minho Advanced Computing Centre, Portugal, under project IDs EHPC-DEV-2025D04-134, EHPC-BEN-2025B06-045, EHPC-DEV-2025D09-041, and EHPC-DEV-2025D09-049, through EuroHPC Development and Benchmark Access calls.

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

## A TECHNICAL APPENDICES AND SUPPLEMENTARY MATERIAL

This document provides a comprehensive overview of the technical appendices

**Theoretical Foundations**

- **Section A.1:** Theoretical justification for freezing matrices $A$ and $B$ in Co-LoRA, proving that this approach achieves zero aggregation error in federated learning.
- **Section A.2:** Proof of Theorem 1 showing that orthogonal initialization of $A$ and $B$ maximizes the representational capacity of Co-LoRA.

**Implementation Details**

- **Section A.3:** Details on client model selection methodology in heterogeneous PFL scenarios.
- **Section A.4:** Consistency of FedMosaic over diverse random heterogeneous PFL scenarios.
- **Section A.5:** In-depth explanation of Co-LoRA, including initialization with orthogonal sets, weight alignment, and orthogonality-enforcing post-processing.
- **Section A.6:** Comprehensive description of experimental configurations, including model architecture details, benchmark specifications, evaluation metrics, and hyperparameter settings.
- **Section A.7:** Summary of client model configuration of all heterogeneous PFL experiments.
- **Section A.8:** Comparative analysis of memory and computational costs across various baselines.
- **Section A.9:** Comparative analysis of communication (transmission) cost of FedMosaic.
- **Section A.10:** Empirical analysis supporting block-wise aggregation through CKA similarity measurements across heterogeneous models.
- **Section A.29:** Justification and empirical analysis for the choice of pivot model during Co-LoRA alignment.
- **Section A.35:** Detailed algorithms for the FedMosaic framework, including initialization, alignment, and training procedures.

**Extended Experimental Results**

- **Section A.11:** Results with more diverse heterogeneous PFL scenarios, including experiments with three different model architectures.
- **Section A.12:** Experimental results in homogeneous PFL setups.
- **Section A.13:** Experimental results in heterogeneous PFL scenarios using Qwen-based LLaVA models.
- **Section A.14:** Additional experiments on various cross-family heterogeneous PFL setups.
- **Section A.15:** Results on the text-only benchmark, complementing our Fed-LLM-Large experiments.
- **Section A.16:** Detailed client-wise accuracy analysis for both homogeneous and heterogeneous PFL setups.
- **Section A.23:** Extended evaluation of fast adaptation capabilities on additional unseen tasks.
- **Section A.24:** Experimental results in LoRA rank heterogeneous PFL scenarios.
- **Section A.25**: Experimental results in task-wise non-i.i.d. heterogeneous PFL scenarios.
- **Section A.26:** Experimental results in heterogeneous PFL setups using small T5 models.
- **Section A.32:** Experimental results in heterogeneous PFL-Dynamic setups with asynchronous data distribution-shift among clients.

**Ablation Studies**

- **Section A.17:** Analysis of the choice (*i.e.*, type and size) of public data $D_P$ for Co-LoRA alignment.
- **Section A.18:** Analysis of how the change of various hyperparameters introduced in FedMosaic affects performance.
- **Section A.19:** Ablation study on RELA, comparing different strategies for measuring task similarity.
- **Section A.20:** Analysis on the size of $W_s$ for RELA in terms of computational cost and performance in a PFL setup.
- **Section A.21:** Investigation of the effects of weight alignment in Co-LoRA.
- **Section A.22:** Discussion of RELA and existing similarity-based model aggregation in federated learning and multi-task learning.
- **Section A.30:** Privacy analysis of the sanitized gradient $\tilde{g}$ of RELA under gradient inversion attack.
- **Section A.31:** Visualization of task relevance matrix of DRAKE.

**Limitations/Future Work and Benchmark Details**

- **Section A.27:** Comprehensive description of our proposed DRAKE benchmark, including task categories, dataset sources, client configuration, and comparison with existing FL benchmarks.
- **Section A.28**: Comparison between DRAKE and existing FL benchmarks.
- **Section A.33:** Discussion of limitations and potential directions for future work.
- **Section A.34:** Impact statement addressing the broader implications of our research.

## A.1 THEORETICAL JUSTIFICATION FOR FREEZING $A$ AND $B$ IN CO-LORA

**Recap of Co-LoRA.**   Co-LoRA consists of LoRA matrices $A \in \mathbb{R}^{r \times d_I}$, $B \in \mathbb{R}^{d_O \times r}$, along with shareable modules $P \in \mathbb{R}^{r \times r}$, $Q \in \mathbb{R}^r$. Co-LoRA outputs $h_O$ for the given input $h_I$ as:

$$h_O = W_p h_I + B(PAh_I + Q), \tag{8}$$

where $W_p$ refers to the pre-trained weight.

For simplicity, we assume that $Q = \mathbf{0} \in \mathbb{R}^r$, *i.e.*, $h_O = BPAh_I$, and prove under the homogeneous FL setup, but can easily extend to $Q \neq \mathbf{0}$ and the heterogeneous FL setup.

**Definition (Aggregation Error $\delta$).**   We first define the aggregation error $\delta$. In an FL setup with $N$ clients, the server aggregates client updates $\Delta W_i = B_i P_i A_i$ for $i \in [N]$. With FedAvg aggregation, the 'ideal' aggregation $\Delta W^*$ (Sun et al., 2024; Guo et al., 2025) should be:

$$\Delta W^* = \frac{1}{N}(B_1 P_1 A_1 + B_2 P_2 A_2 + \cdots + B_N P_N A_N). \tag{9}$$

However, we cannot perform 'ideal' aggregation in FL, since $W^*$ cannot be decomposed into trainable parameters, *i.e.*, $B$, $P$, and $A$, on client sides (Guo et al., 2025). In other words, although $W^*$ can be computed on the server, it cannot be redistributed as trainable parameters on the client side. As a result, instead of directly averaging $\Delta W_i$ for $i \in [N]$, we average the trainable parameters separately and obtain the practical update $\Delta \tilde{W}^*$ as:

$$\Delta \tilde{W}^* = \left(\frac{1}{N}\sum_{i=1}^{N} B_i\right)\left(\frac{1}{N}\sum_{i=1}^{N} P_i\right)\left(\frac{1}{N}\sum_{i=1}^{N} A_i\right). \tag{10}$$

To this end, we define the aggregation error $\delta = |\Delta W^* - \Delta \tilde{W}^*|$. During local training, we update shareable modules $P$ and $Q$, while freezing $A$ and $B$ to preserve alignment. This way, we minimize the aggregation error $\delta$, following Theorem 2.

**Theorem 2.** *If we freeze $A \in \mathbb{R}^{r \times d_I}$ and $B \in \mathbb{R}^{d_O \times r}$, aggregation error $\delta = 0$.*

**Proof of Theorem 2.**

*Proof.* If $A$ is frozen after alignment, then $A = A_1 = A_2 = \cdots = A_N$. Similarly, freezing $B$ implies $B = B_1 = B_2 = \cdots = B_N$. Under this condition, both the ideal update $\Delta W^*$ (Eq. 9) and the practical update $\Delta \tilde{W}^*$ (Eq. 10) can be simplified to:

$$\Delta W^* = \Delta \tilde{W}^* = B \left( \frac{1}{N} \sum_{i=1}^{N} P_i \right) A. \tag{11}$$

Thus, $\delta = 0$. $\qquad\square$

### A.2 PROOF OF THEOREM 1

*Proof.* For simplicity, we assume that $Q = \mathbf{0} \in \mathbb{R}^r$.

The weight update for Co-LoRA is given by:

$$\Delta W = BPA, \tag{12}$$

where $B \in \mathbb{R}^{d_O \times r}, P \in \mathbb{R}^{r \times r}, A \in \mathbb{R}^{r \times d_I}$.

We can decompose the matrix $P$ using scalar values and basis vectors as follows:

$$P = \sum_{i=1}^{r} \sum_{j=1}^{r} P_{ij} \cdot e_i e_j^T, \tag{13}$$

where $P_{ij}$ refers to the scalar value of $(i, j)$ position of matrix P, and $e_i$ and $e_j$ refers to the standard basis vector in $\mathbb{R}^r$.

Substituting Eq.13 into Eq.12, we get:

$$\Delta W = BPA = B \left( \sum_{i,j} P_{ij} \cdot e_i e_j^T \right) A = \sum_{i,j} P_{ij} \cdot (Be_i) \cdot \left( e_j^T A \right). \tag{14}$$

Since $Be_i = b_i \in \mathbb{R}^{d_O}$ ($i_{\text{th}}$ column of $B$), $e_j^T A = a_j^T \in \mathbb{R}^{d_I}$ ($j_{\text{th}}$ row of $A$), we can simplify Eq. 14 as:

$$\Delta W = \sum_{i,j} P_{ij} \cdot b_i a_j^T. \tag{15}$$

Therefore, representation space of $\Delta W$, denoted as $S_{PQ}$, is defined as:

$$S_{PQ} = \text{span}\{b_i a_j^T \mid i, j = 1, \dots, r\}. \tag{16}$$

Since both $\{b_1, b_2, \dots, b_r\} \subset \mathbb{R}^{d_O}$ and $\{a_1^T, a_2^T, \dots, a_r^T\} \subset \mathbb{R}^{d_I}$ form orthogonal sets, their outer products $\{b_i a_j^T\}_{i,j=1}^{r}$ are linearly independent.

Therefore, the representation space $S_{PQ}$ has the maximum possible dimension:

$$\dim(S_{PQ}) = r^2. \tag{17}$$

$\qquad\square$

To satisfy Eq.17, it is sufficient for the column vectors of $B$ and the row vectors of $A$ to form linearly independent sets, even if they are not strictly orthogonal. However, compared to initialization with linearly independent vectors, orthogonal initialization offers additional advantages, such as faster (Hu et al., 2020) and stable (Nowak et al., 2024) convergence. Therefore, motivated by both the practical advantages and the theoretical justification provided in Theorem 1, we initialize $A$ and $B$ with orthogonal sets.

A.3  DETAILS OF CLIENT MODEL SELECTION IN THE HETEROGENEOUS PFL SCENARIO

In the heterogeneous PFL scenarios, we assign each client's model based on supervised fine-tuning (SFT) performance on their respective tasks. Specifically, clients are assigned larger models when these models exhibit significantly better SFT performance than smaller ones. Conversely, when the performance difference between model sizes is negligible, we allocate smaller models to maximize computational efficiency, as larger models require substantially more computational resources. We summarize the SFT performance of each LLaVA-Llama3 variant model and the resulting client-wise model allocation in Tab. 8.

| | DRAKE | | | | | | | | | |
|---|---|---|---|---|---|---|---|---|---|---|
| Model | Client 1 | Client 2 | Client 3 | Client 4 | Client 5 | Client 6 | Client 7 | Client 8 | Client 9 | Client 10 |
| LLaVA-Llama3.2-1B | 78.43 | 57.98 | 65.05 | 62.53 | 63.15 | 63.35 | 47.56 | 51.68 | 53.07 | 50.95 |
| LLaVA-Llama3.2-3B | **79.33** | **59.01** | **66.75** | **63.33** | **66.00** | **76.15** | **63.40** | **68.21** | **63.08** | **57.32** |
| Allocated Model | LLaVA-Llama3.2-**1B** | | | | LLaVA-Llama3.2-**3B** | | | | | |

| | HFLB | | | | | | | | |
|---|---|---|---|---|---|---|---|---|---|
| Model | Client 1 | Client 2 | Client 3 | Client 4 | Client 5 | Client 6 | Client 7 | Client 8 | Client 9 |
| LLaVA-Llama3.2-1B | 79.78 | **98.18** | 51,23 | 80.73 | 71.58 | 80.48 | 71.64 | 76.45 | 89.45 |
| LLaVA-Llama3.2-3B | **80.33** | 97.32 | **52.85** | **81.85** | **73.84** | **82.75** | **74.47** | **79.50** | **91.50** |
| Allocated Model | LLaVA-Llama3.2-**1B** | | | LLaVA-Llama3.2-**3B** | | | | | |

Table 8: **Per-client SFT performance of different LLaVA-Llama3 model variants.** We assign the larger model (*i.e.*, LLaVA-Llama-3.2-3B) to clients whose SFT performance shows a substantial gain, while the smaller model (*i.e.*, LLaVA-Llama-3.2-1B) is assigned otherwise.

A.4  EFFECT OF CLIENT MODEL ASSIGNMENT

To validate whether FedMosaic consistently achieves strong performance regardless of the specific model assigned to each client, we conduct experiments with varying ratios of LLaVA-1B and LLaVA-3B models, where model assignments are randomly determined in each run. In Table 9, we summarize the PFL results of supervised fine-tuning (SFT) and FedMosaic across diverse heterogeneous PFL scenarios, each corresponding to a different configuration of client model selections in the DRAKE-dynamic benchmark. We observe that FedMosaic consistently outperforms SFT with minimal performance variance across heterogeneous setups, demonstrating its robustness to the composition of client models.

| Scenario | Method | Self | | Others | |
|---|---|---|---|---|---|
| | | $A_{last}$ ↑ | $A_{\text{AUC}}$ ↑ | $A_{last}$ ↑ | $A_{\text{AUC}}$ ↑ |
| 3×1B Clients / 7×3B Clients | SFT | 65.57±0.83 | 57.46±0.44 | 47.80±0.17 | 46.80±0.10 |
| | FedMosaic | 67.02±0.34 | 58.85±0.15 | 51.55±0.31 | 49.62±0.04 |
| | Δ (FedMosaic- SFT) | +1.45 | +1.39 | +3.75 | +2.82 |
| 4×1B Clients / 6×3B Clients | SFT | 64.08±0.77 | 56.61±0.42 | 46.82±0.44 | 46.24±0.23 |
| | FedMosaic | 66.28±0.23 | 58.47±0.17 | 51.20±0.40 | 49.26±0.29 |
| | Δ (FedMosaic- SFT) | +2.20 | +1.86 | +4.38 | +3.02 |
| 5×1B Clients / 5×3B Clients | SFT | 63.72±0.77 | 56.61±0.16 | 46.70±0.07 | 45.99±0.06 |
| | FedMosaic | 65.34±0.21 | 58.25±0.17 | 51.08±0.23 | 49.07±0.25 |
| | Δ (FedMosaic- SFT) | +1.62 | +1.64 | +4.38 | +3.08 |
| 6×1B Clients / 4×3B Clients | SFT | 63.49±0.93 | 56.02±0.28 | 46.63±0.07 | 45.63±0.02 |
| | FedMosaic | 65.75±0.54 | 57.96±0.18 | 50.57±0.23 | 48.65±0.18 |
| | Δ (FedMosaic- SFT) | +2.26 | +1.94 | +3.94 | +3.02 |
| 7×1B Clients / 3×3B Clients | SFT | 62.72±0.19 | 55.54±0.02 | 45.54±0.22 | 44.83±0.07 |
| | FedMosaic | 65.40±0.53 | 57.56±0.07 | 49.41±0.18 | 47.54±0.08 |
| | Δ (FedMosaic- SFT) | +2.68 | +2.02 | +3.87 | +2.71 |

Table 9: **Effect of heterogeneous PFL scenarios on DRAKE-Dynamic.** We change the number of LLaVA-Llama3.2-1B and LLaVA-Llama3.2-3B models and randomly assign the model size of each client. The '4×1B Clients / 6×3B Clients' scenario here is different with the client model selection in Tab. 8.

## A.5 More Details of Co-LoRA

As discussed in Sec.4.2.2, we enforce orthogonality on the $A$ and $B$ matrices to maximize the capacity of Co-LoRA, following Theorem1. This is achieved through (i) initializing $A$ and $B$ with orthogonal sets, (ii) weight alignment, and (iii) orthogonality-enforcing post-processing.

**Initialization with Orthogonal Set.** We initialize row vectors of $A \in \mathbb{R}^{r \times d_I}$ and column vectors of $B \in \mathbb{R}^{d_O \times r}$ with an orthogonal set, as follows:

$$AA^\top = I_r, \quad B^\top B = I_r, \tag{18}$$

where $I_r$ refers to the $r \times r$ identity matrix.

**Weight Alignment.** For $B_i$ and $B_j$, we align them using canonical correlation analysis (CCA), as follows:

$$B_j = (\Pi_i{}^{-1})^T \cdot (\Pi_i)^T \cdot B_i, \tag{19}$$

where $\Pi_i$ and $\Pi_j$ are projection matrices that project $B_i$ and $B_j$ into to the maximally correlated space, respectively. During alignment, $B_j$ remains orthogonal because $B_i$ is initialized with orthogonal vectors, and the projection in Eq. 19 preserves this orthogonality.

However, during the alignment of $A_i$ and $A_j$ in Sec. 4.2.2, orthogonality can be disrupted due to L2 loss training. To prevent significant deviation from orthogonality, we add a regularization term to the objective function of $A$ alignment as follows:

$$\min_{A_j} \frac{1}{|\mathcal{D}_p|} \sum_{(x,y) \in \mathcal{D}_p} \|(A_i(x) - A_j(x)\|_2^2 + \lambda \left\| A_j^\top A_j - I \right\|_F^2, \tag{20}$$

where $\| \cdot \|_F$ denotes the Frobenius norm. Despite the regularization, $A_j$ may not strictly satisfy orthogonality. Therefore, to ensure exact orthogonality, we perform an additional post-processing step.

**Orthogonality-Enforcing Post-processing.** To enforce orthogonality in $A_j$, we apply orthogonal projection. Specifically, we aim to find the closest orthogonal matrix $A_j^*$ from $A_j$ by minimizing the Frobenius norm, as follows:

$$\min_{A_j^{*T} A_j^* = I} \|A_j - A_j^*\|_F. \tag{21}$$

This optimization can be efficiently solved using the Singular Value Decomposition (SVD). We compute the SVD of $A$:

$$A_j = U\Sigma V^T, \tag{22}$$

where $U \in \mathbb{R}^{r \times r}$, $\Sigma \in \mathbb{R}^{r \times d_I}$, and $V \in \mathbb{R}^{d_I \times d_I}$.

The closest orthogonal matrix $A_j^*$ is then given by:

$$A_j^* = UV^T. \tag{23}$$

As a result, we get $A_i$ and $B_i$, initialized with orthogonal sets and kept frozen during alignment, and $A_j^*$ and $B_j$, which are aligned with $A_i$ and $B_i$ while maintaining orthogonality.

## A.6 Details of Experimental Setups

**Models.** To employ LLaVA-Llama3 variant models (*i.e.*, LLaVA-Llama3.2-1B, LLaVA-Llama3.2-3B, LLaVA-Llama3.1-8B) and LLaVA-Qwen2.5 variant models (*i.e.*, LLaVA-Qwen2.5-0.5B, LLaVA-Qwen2.5-1.5B, and LLaVA-Qwen2.5-3B) in heterogeneous PFL scenarios, we instruction-tune them on LLaVA-Instruct-158K (Liu et al., 2023b), following the original LLaVA training setup (Liu et al., 2023b). We summarize the performance of the instruction-tuned variants in Tab. 10 and Tab. 11. We use Llama-3 base models for text-only benchmarks.

| Model | VQAv2 (Goyal et al., 2017) | GQA (Hudson & Manning, 2019) | VizWiz (Gurari et al., 2018) | SciQA (Lu et al., 2022) | TextVQA (Singh et al., 2019) |
|---|---|---|---|---|---|
| LLaVA-1.5-7B | 78.5 | 62.0 | 50.0 | 66.8 | 58.2 |
| LLaVA-Llama3.2-1B | 75.3 | 59.7 | 45.3 | 67.3 | 49.0 |
| LLaVA-Llama3.2-3B | 78.7 | 62.6 | 45.3 | 75.7 | 55.8 |
| LLaVA-Llama3.1-8B | 79.8 | 64.2 | 47.6 | 76.4 | 57.9 |
| LLaVA-Qwen2.5-0.5B | 71.8 | 56.5 | 37.3 | 58.1 | 43.4 |
| LLaVA-Qwen2.5-1.5B | 75.3 | 59.3 | 40.9 | 69.5 | 51.8 |
| LLaVA-Qwen2.5-3B | 77.2 | 61.4 | 50.9 | 74.1 | 54.7 |

Table 10: **Zero-Shot performance of instruction-tuned LLaVA-1.5 variants on academic-task-oriented benchmarks.** We use the CLIP ViT-L/336px model as the vision encoder for all variants.

| Model | POPE (Li et al., 2023c) | MME (Fu et al., 2023a) | MMBench (Liu et al., 2024) | SEED-Bench (Li et al., 2023a) | LLaVA-Wild (Liu et al., 2023b) | MM-Vet (Yu et al., 2023) |
|---|---|---|---|---|---|---|
| LLaVA-1.5-7B | 85.9 | 1510.7 | 64.3 | 58.6 | 65.4 | 31.1 |
| LLaVA-Llama3.2-1B | 85.0 | 1338.1 | 61.3 | 57.8 | 59.6 | 28.7 |
| LLaVA-Llama3.2-3B | 86.1 | 1446.5 | 70.9 | 62.5 | 67.0 | 34.5 |
| LLaVA-Llama3.1-8B | 86.3 | 1486.1 | 72.5 | 64.1 | 73.7 | 32.7 |
| LLaVA-Qwen2.5-0.5B | 86.2 | 1251.9 | 53.6 | 53.1 | 55.0 | 23.1 |
| LLaVA-Qwen2.5-1.5B | 86.4 | 1376.4 | 66.8 | 60.6 | 60.0 | 27.9 |
| LLaVA-Qwen2.5-3B | 87.2 | 1447.6 | 71.4 | 63.2 | 66.7 | 33.1 |

Table 11: **Zero-Shot performance of instruction-tuned LLaVA-1.5 variants on instruction-following benchmarks.** We use the CLIP ViT-L/336px model as the vision encoder for all variants.

**Baselines.** We compare FedMosaic with SOTA PFL methods: DITTO (Li et al., 2021), FedSim (Pillutla et al., 2022), FedIT (Zhang et al., 2024a), TAKFL (Morafah et al., 2024), FedDPA (Long et al., 2024), FedDAT (Chen et al., 2024a), PerAda (Xie et al., 2024), and FedMKT (Fan et al., 2025). Federated distillation baselines (*i.e.*, TAKFL, PerAda, and FedMKT) share knowledge between heterogeneous models via logits, while FedMosaic uses Co-LoRA. For the other baselines, we combine them with Fed-ET (Cho et al., 2022), which aggregates models among homogeneous clients. We also compare with supervised fine-tuning (SFT), where clients train independently without knowledge sharing and often outperforming PFL methods under data heterogeneity (Ghari & Shen, 2024).

**Benchmarks.** We evaluate FedMosaic on Multi-modal FL benchmarks, such as our proposed DRAKE and HFLB, and LLM-FL benchmarks, such as Fed-Scope and Fed-Aya, and Fed-LLM-Large. We summarize the details of benchmarks in Tab. 12. As shown in the table, unlike HFLB, which focuses solely on VQA tasks, DRAKE covers diverse task types and involves more images per sample due to its multi-image inputs. In DRAKE-Dynamic, each client learns four tasks, where all data samples are randomly mixed in PFL-Static, while the samples from each task are introduced incrementally in PFL-Dynamic. We provide more details of DRAKE in Sec. A.27.

HFLB (Heterogeneous Federated Learning Benchmark) is a multi-modal PFL benchmark used in FedDAT (Chen et al., 2024a) that includes 7 VQA datasets for heterogeneous tasks across clients, where we assign them into 9 clients by partitioning GQA (Hudson & Manning, 2019) into three clients based on the QA types (*e.g.*, Yes/No, 2 Multi-choice, and 4 Multi-choice). We further split each assigned dataset into four tasks to allow experiments on the PFL-Dynamic setup. We use keywords in questions or GPT to categorize each sample, where the details are provided in Tab. 13.

To simulate PFL in NLP, we modify existing sets (*i.e.*, Fed-Aya and Fed-Scope) and compose larger mixtures (*i.e.*, Fed-LLM-Large) to increase task variety and heterogeneity across clients. Specifically, Fed-Aya is composed of 12 different languages equally sampled from different language families. We further separate the tasks based on the topic of the QA using GPT-4. Fed-Scope (Kuang et al., 2024) includes coding, mathematics, and general capability datasets, where each client learns either one of the three datasets. Fed-LLM-Large is a large-scale text-only benchmark designed for personalized federated learning. To diversify the personal tasks among clients, we combined tasks from three sources, Fed-ChatbotIT and Fed-aya from Fed-LLM (Ye et al., 2024), and Fed-FLAN (Long et al., 2024). This results in 52 clients and 2 tasks per client, where each client fine-tunes on either instruction following tasks from different sources or in different languages (note that all other benchmarks have 4 tasks per client).

| Benchmark | # of Images | # of Questions | # of clients | # of rounds $R$ | # of local steps | Task types |
|---|---|---|---|---|---|---|
| DRAKE | 357K | 274K | 10 | 20 | 100 | Visual Relation, Multi-modal Reasoning, VQA |
| HFLB | 217K | 314K | 9 | 20 | 100 | VQA |
| Fed-Scope | - | 30K | 5 | 20 | 30 | General Capability, Mathematics, Coding |
| Fed-Aya | - | 36K | 8 | 20 | 50 | Question-Answering in 12 Languages |
| Fed-LLM-Large | - | 31K | 52 | 4 | 10 | Instruction Following |

Table 12: **Benchmark details and federated learning configurations.**

| Client 1 | Client 2 | Client 3 |
|---|---|---|
| **GQA** (Hudson & Manning, 2019) : Yes/No 
● Attribute 
● Relation 
● Object 
● Global | **GQA** (Hudson & Manning, 2019) : Four-choice 
● Attribute 
● Relation 
● Category 
● Global | **GQA** (Hudson & Manning, 2019) : Two-choice 
● Attribute 
● Relation 
● Category 
● Global |
| **Client 4** | **Client 5** | **Client 6** |
| **Abstract VQA** (Antol et al., 2015) 
● Attribute 
● Number 
● Yes/No 
● Others | **SNLI-VE** (Xie et al., 2019) 
● Action 
● Scene Context & Obj Relations 
● Object-Centric 
● Commonsense | **COCOQA** (Ren et al., 2015) 
● Object 
● Number 
● Color 
● Location |
| **Client 7** | **Client 8** | **Client 9** |
| **NLVR2** (Suhr et al., 2019) 
● Presupposition Negation Universal 
● Spatial Comparative 
● Cardinality Existential 
● Coordination Coreference | **VizWiz** (Gurari et al., 2018) 
● Food, Brand, and Label Identification 
● Color and Type Identification 
● Optical Character Recognition 
● General Object Identification | **AQUA** (Garcia et al., 2020) 
● "standing" keyword 
● "sit" keyword 
● "wear", "hold", "walk", "talk" keywords 
● Other keywords |

Table 13: **Per-client task configuration of HFLB.** Datasets are partitioned into question-type subsets using keyword rules or GPT judgement.

**Metrics.** We measured the accuracy metrics, $A_{last}$ and $A_{\text{AUC}}$, based on the correct choice for multi-choice questions and the correct tokens compared to the ground-truth answer for the open-ended questions. For Fed-Aya, we used GPT to rate the generated response compared to the given ground-truth response. We use the same prompt template shown in Fig. 9 from the original paper (Singh et al., 2024). For Fed-Scope, we follow the evaluation process of the original paper (Kuang et al., 2024) and use MMLU (Hendrycks et al., 2020), GSM8K (Cobbe et al., 2021), and HumanEval (Chen et al., 2021) benchmarks to evaluate general capability, math and coding skills, respectively. For Fed-LLM-Large, we use Rouge-L (Lin, 2004) metric as it is commonly used metric to assess LLM generation quality compared to ground-truth (Long et al., 2024; Li et al., 2024c).

**Implementation Details and Hyperparameters.** We set the batch size to 4, the learning rate for Co-LoRA to $5 \times 10^{-5}$, and for other parameters to $2 \times 10^{-5}$. We use the Constant LR scheduler and the AdamW optimizer (Loshchilov & Hutter, 2019) for all datasets. We use the LoRA rank $r$ of 128 attached to all linear layers in the LLM backbone. For LLM experiments, we set the learning rate to $3 \times 10^{-4}$ and the LoRA rank to 16. For federated learning, we set the total number of communication rounds $R$ to 20, with the local training step of 100. For a fair comparison, we adjust the local training step for each baseline and FedMosaic to use the same computational cost as SFT, following (Seo et al., 2025). For the PFL-Dynamic setup, we employ memory-only training, where newly encountered samples are added to episodic memory, and training batches are retrieved solely from memory, following (Koh et al., 2023; Seo et al., 2024; 2025). For episodic memory, we adopt a memory-infinite setup (Prabhu et al., 2023; Seo et al., 2025), assuming all data can be stored in memory, reflecting that memory cost is not a bottleneck in real-world scenarios. For $\tau$, the softmax temperature parameter used in RELA, we use 0.5 in all experiments. For $\alpha$, the EMA ratio of client-wise gradient in Eq. 2, we use 0.5 in all experiments. For $\lambda$, the loss balancing coefficient between the L2 loss and the regularization term in Eq. 20, we use 0.5. For $N_s$, gradient sampling ratio for sanitized gradient $\tilde{g}_i$, we use 40%. For $\mu$, the noise scale applied to $\epsilon$, we set $\mu = 10^{-4}$. We set $N_B$, the number of layers employing Co-LoRA, to 4 in all experiments. See Sec. A.18 for detailed hyperparameter analysis.

**[Instruction]**
Please act as an impartial judge and evaluate the quality of the response provided by an AI assistant to the user question displayed below. A good answer should follow these rules:
1. It should be in the same language as the question.
2. It should answer the request in the instruction.
3. It should be factually and semantically comprehensible.
4. It should be grammatically correct and fluent.

Begin your evaluation by providing a short explanation. Be as objective as possible. After providing your explanation, you must rate the response on a scale of 1 to 10 by strictly following this format: "`[[rating]]`", for example: "Rating: `[[5]]`". A human-annotated answer is given for reference.

**[Question]**
{question}

**[The Start of Assistant's Answer]**
{answer}
**[The End of Assistant's Answer]**

**[Reference]**
{reference}

Figure 9: **Prompt template used in GPT-4 judge.**

Federated distillation baselines (*i.e.*, TAKFL, PerAda, and FedMKT) share knowledge between heterogeneous models via logits, while FedMosaic employs Co-LoRA. For the other baselines, we combine them with Fed-ET (Cho et al., 2022), which aggregates models among homogeneous clients. For $W_s$, a small-scale pre-trained MLLM used for calculating client-wise gradients in RELA, we adopt the smallest model among all client models. For example, in Table 16, where clients have three different types of heterogeneous architectures (*i.e.*, LLaVA-Llama3.2-1B, LLaVA-Llama3.2-3B, and LLaVA-Llama3.1-8B), we employ LLaVA-Llama3.2-1B as $W_s$ for all clients. This choice ensures computational efficiency and maintains gradient dimension consistency for cosine similarity calculation. Note that we can employ a lighter $W_s$ (*e.g.*, reduced size or lower-bit quantization) to further decrease the computation overhead of gradient calculation in RELA, as detailed in Sec. A.20. All experiments are executed in Python 3.10, on four Ubuntu 20.04 machines, with 8 NVIDIA RTX A6000 GPUs each. Each experiment runs on a single RTX A6000 GPU in a day.

## A.7 CLIENT MODEL CONFIGURATIONS

We summarize the model configurations for each PFL experiment, including model types and their counts, in Tab. 14. For experiments on multi-modal benchmarks, we use LLaVA-Llama3 or LLaVA-Qwen2.5 variants, while for text-only benchmarks, we use Llama-3 variants.

## A.8 COMPARISON OF COMPUTATIONAL AND MEMORY COSTS

We compare the computational cost $\mathcal{C}$ and memory cost $\mathcal{M}$ of various baselines and summarize the results in Tab. 15. Following (Seo et al., 2025), we measure computational cost in FLOPs per iteration and memory cost in Bytes. Specifically, for each baseline, we report the relative FLOPs and relative Bytes in comparison to supervised fine-tuning (SFT), which only requires a single forward and backward pass without any extra computation and memory overhead.

**Comparison of Computational Cost $\mathcal{C}$.** We first compare the computational cost of FL methods. PerAda (Xie et al., 2024) incurs approximately twice the computational cost compared to other baselines, since it sequentially updates both the personalized adapter and the local adapter. Similarly, FedDAT (Chen et al., 2024a) independently optimizes the local adapter and the dual adapter teacher

| Benchmark | Experiment | LLaVA-Llama3 | | | LLaVA-Qwen2.5 | | | Llama-3 | | |
|---|---|---|---|---|---|---|---|---|---|---|
| | | 1B | 3B | 8B | 0.5B | 1.5B | 3B | 1B | 3B | 8B |
| Multi-modal | Tab. 2 DRAKE-Dynamic | 4 | 6 | 0 | 0 | 0 | 0 | – | – | – |
| | Tab. 2 HFLB-Dynamic | 3 | 6 | 0 | 0 | 0 | 0 | – | – | – |
| | Tab. 3 DRAKE-Static | 4 | 6 | 0 | 0 | 0 | 0 | – | – | – |
| | Tab. 4 DRAKE-Dynamic | 0 | 3 | 0 | 0 | 4 | 3 | – | – | – |
| | Tab. 7 DRAKE-Dynamic | 4 | 6 | 0 | 0 | 0 | 0 | – | – | – |
| | Tab. 16 DRAKE | 3 | 5 | 2 | 0 | 0 | 0 | – | – | – |
| | Tab. 17 DRAKE-Homo | 0 | 10 | 0 | 0 | 0 | 0 | – | – | – |
| | Tab. 17 HFLB-Homo | 0 | 9 | 0 | 0 | 0 | 0 | – | – | – |
| | Tab. 18 DRAKE-Homo | 0 | 0 | 0 | 0 | 0 | 10 | – | – | – |
| | Tab. 18 DRAKE-Hetero | 0 | 0 | 0 | 3 | 2 | 5 | – | – | – |
| | Tab. 19 Llama 3B / Qwen 1.5B | 0 | 6 | 0 | 0 | 4 | 0 | – | – | – |
| | Tab. 19 Llama 1B / 3B / Qwen 1.5B | 2 | 5 | 0 | 0 | 3 | 0 | – | – | – |
| | All other analysis / ablation experiments | 4 | 6 | 0 | 0 | 0 | 0 | – | – | – |
| Text-only | Tab. 6 Fed-LLM-Large-Dynamic | – | – | – | – | – | – | 26 | 26 | 0 |
| | Tab. 7 Fed-Scope-Static | – | – | – | – | – | – | 0 | 3 | 2 |
| | Tab. 20 Fed-aya-Dynamic | – | – | – | – | – | – | 4 | 4 | 0 |
| | Tab. 20 Fed-Scope-Static | – | – | – | – | – | – | 0 | 3 | 2 |

Table 14: **Client model configuration details for each experiment.** Counts per model family/size.

| | Memory Cost $\mathcal{M}$ | | Computational Cost $\mathcal{C}$ | |
|---|---|---|---|---|
| Methods | Overhead Type | Relative $\mathcal{M}$ to SFT | Overhead Type | Relative $\mathcal{C}$ to SFT |
| SFT | - | 1.000 | - | 1.000 |
| DITTO (ICML 2021) | Dual Adapter | 1.052 | Double Forward/Backward | 2.000 |
| FedSim (ICML 2022) | Dual Adapter | 1.052 | Double Forward | 1.487 |
| FedIT (ICASSP 2024) | - | 1.000 | - | 1.000 |
| TAKFL (NeurIPS 2024) | - | 1.000 | Distill Logit Extract | 1.294 |
| PerAda (CVPR 2024) | Dual Adapter | 1.052 | Double Forward/Backward & Logit Extraction | 2.294 |
| FedDAT (AAAI 2024) | Triple Adapter | 1.104 | Double Forward/Backward | 2.564 |
| FedDPA (NeurIPS 2024) | Dual Adapter | 1.052 | Double Forward/Backward | 2.051 |
| FedMKT (COLING 2025) | Public data Logit Share | 1.002 | Logit Extraction | 1.574 |
| FedMosaic (**Ours**) | Co-LoRA | 1.053 | Last Layer Gradient Compute & Client-wise Similarity Compute | 1.161 |

Table 15: **Comparison of memory and computational costs.** FedMosaic incurs additional memory and computational costs compared to SFT, but only by approximately 5.3% and 16.1%, respectively.

(DAT), which also results in a double computational cost. FedMKT, PerAda, and TAKFL perform knowledge distillation and transfer using logits (*e.g.*, from public data), which introduces additional forward computation for logit extraction.

Our proposed FedMosaic incurs a minor additional computational cost due to (i) gradient computation from the frozen pre-trained model used for measuring task similarity (ii) dual adapter structure, and (iii) Co-LoRA alignment, but this overhead amounts to only approximately 9.8% more FLOPs compared to SFT. Specifically, gradient computation and the dual adapter structure add about 9.8% overhead during training, while Co-LoRA alignment adds 6.25% overhead before federated learning to align heterogeneous models, totaling 16.1% additional cost. Here, the Co-LoRA alignment overhead is computed as the ratio between the number of batch iterations used for Co-LoRA alignment and those used for SFT training across all clients.

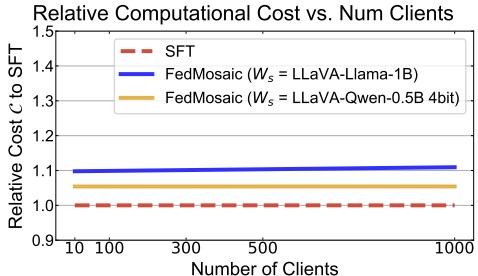

Figure 10: **Relative Computational Cost to SFT *vs*. Number of clients of FedMosaic.**

The reason for the small increase in computation during training is that (i) we apply 60% gradient compression for similarity calculation in RELA, (ii) we only use the last layer gradient of the small-scale pre-trained frozen model, and (iii) we perform gradient computation once every 10 batch iterations rather than on all batches for computational efficiency, as mentioned in Sec. 4.1. Similarly, the small overhead of Co-LoRA alignment stems from the fact that (i) it is performed only once before federated training to establish a shared initialization, and (ii) it is required only between heterogeneous model-type pairs, not across all clients, as mentioned in Sec. 4.2.2. Note that while

FedMosaic also introduces a dual adapter structure, *i.e.*, maintaining the local adapter and the global adapter separately, similar to PerAda, FedDAT, and FedDPA, we optimize the dual-adapter at once using a learnable balancing parameter in Co-LoRA, which adaptively balances local and global output. In contrast, the baselines separately optimize each adapter, thus incurring a double computational cost. Thanks to these designs, the relative overhead remains approximately at 10% even under large-scale client setups (*e.g.*, 1000 clients), as shown by the blue line in Fig. 10. Finally, the cost can be further reduced by adopting smaller or quantized $W_s$, as discussed in Sec. A.20. For example, using a LLaVA-Qwen-0.5B-4bit model as $W_s$ reduces the overhead to approximately 5% with 1000 clients (yellow line in Fig. 10), while achieving performance comparable to LLaVA-Llama-1B (Sec. A.20).

**Comparison of Memory Cost $\mathcal{M}$.** We then compare the memory cost of FL methods. FedMKT incurs a marginal additional memory overhead, as it requires storing logits from clients for knowledge aggregation. DITTO, FedSim, PerAda, FedDAT, FedDPA, as well as FedMosaic employ a dual adapter structure, which maintains both local and global adapters separately, to preserve global knowledge. It incurs additional memory cost, but since we adopt the LoRA adapter (Hu et al., 2022), which occupies significantly less memory compared to pre-trained weights (Qi et al., 2024), the dual adapter only consumes about 5% additional memory cost compared to using a single LoRA adapter.

### A.9 Details of Communication Costs in FedMosaic

The only additional transmission in FedMosaic compared to Vanilla in Tab. 7 (*i.e.*, FedMosaic w/o RELA and w/o Co-LoRA) is a single EMA-updated gradient vector per client, adding 8.6% overhead compared to sending only local LoRA parameters. However, as mentioned in Sec. 4.1, we apply gradient compression by randomly selecting only $N_s$% of the client-specific gradient vectors, thereby reducing communication costs. With $N_s = 40\%$, the communication overhead drops to 3.4%.

Moreover, while baselines transmit full LoRA modules (*i.e.*, $A \in \mathbb{R}^{r \times d}$ and $B \in \mathbb{R}^{d \times r}$) across all layers, FedMosaic freezes $A$ and $B$ in Co-LoRA layers and transmits only $P \in \mathbb{R}^{r \times r}$ and $Q \in \mathbb{R}^r$, significantly reducing 14.3% communication cost. Combining the marginal overhead from client-specific vectors with the reduction from Co-LoRA, FedMosaic consequently achieves 10.9% lower communication cost than even the most efficient baseline (*i.e.*, FedAvg), regardless of the number of clients, ensuring scalability and communication efficiency (Blue line in Fig. 11).

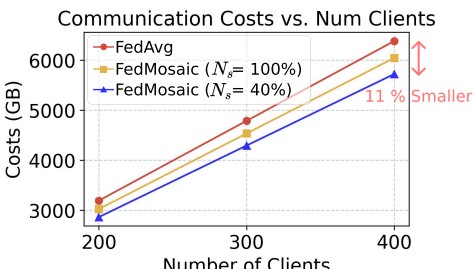

Figure 11: **Communication Cost *vs*. Number of clients of FedMosaic.**

### A.10 Empirical Analysis of Block-Wise Aggregation

To identify layer-wise correspondences between depth-heterogeneous models, we analyze representation alignment using CKA (Kornblith et al., 2019). Specifically, we measure similarity across layers within the Llama-3 family (1B, 3B, 8B) and the Qwen-2.5 family (0.5B, 1.5B, 3B), as illustrated in Fig. 12. As shown in the figure, layers with the same relative depth exhibit strong alignment, indicating approximately linear alignment within both the Llama-3 and Qwen-2.5 families. Moreover, we observe near-linear alignment even across families, *i.e.*, between Llama-3 and Qwen-2.5, despite weaker linearity than intra-family alignment. Moreover, to demonstrate that this layer-wise correlation trend generally holds across different models, not just between Llama and Qwen, we have additionally included the layer-wise correlation analysis between InternLM (Cai et al., 2024) and Llama in Fig. 13, which shows the same trend as our previous findings. This empirical analysis supports our block-wise aggregation of Co-LoRA. We provide an illustration of the block-wise Co-LoRA in Fig. 14.

### A.11 Experimental Results in More Diverse Heterogeneous PFL Scenarios

We evaluate FedMosaic in heterogeneous PFL using three different heterogeneous architectures, *i.e.*, LLaVA-Llama3.2-1B, LLaVA-Llama3.2-3B, and LLaVA-Llama3.1-8B, and summarize the results in

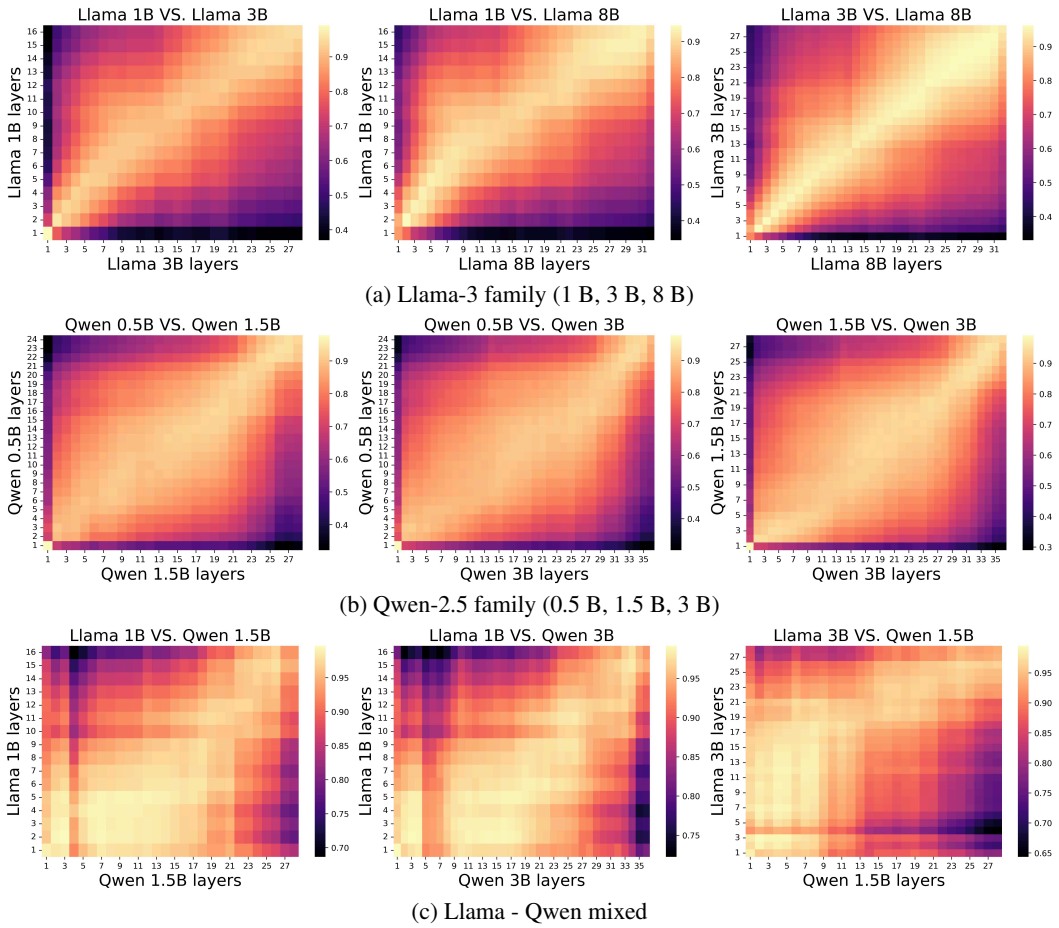

Figure 12: **Layer-wise similarity across model scales measured with CKA.** Each heat-map cell reports the centered-kernel–alignment (CKA) similarity between the hidden representations of heterogeneous multi-modal LLMs at every pair of layers. Here, lighter colors indicate higher similarity. **(a)** For the Llama-3–based heterogeneous models and **(b)** for the Qwen-2.5–based heterogeneous models, the brightest (*i.e.*, the highest similarity) band appears roughly along the main diagonal, indicating that layers with *relative depth* align most strongly. The near-linear trend supports our proposed block-wise aggregation, which transfers knowledge from smaller to larger models that have the same relative depth within the same architectural family.

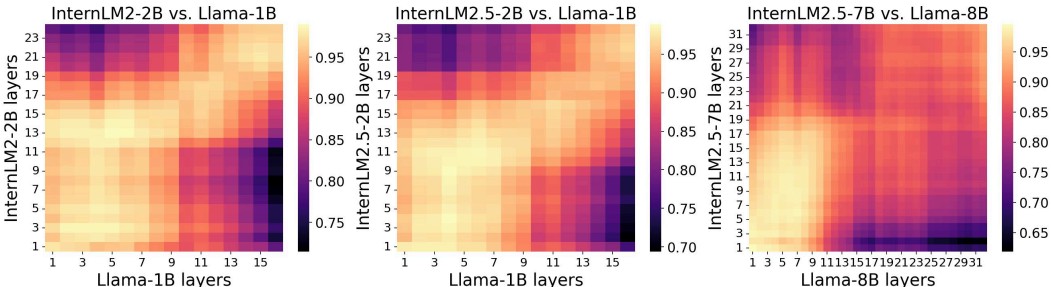

Figure 13: **Layer-wise similarity across models measured with CKA.** Each heat-map cell reports the centered-kernel–alignment (CKA) similarity between the hidden representations from InternVL (Chen et al., 2024b) using InternLM as base LLM and LLaVA using Llama as base LLM. The layers with *relative depth* align most strongly even between models

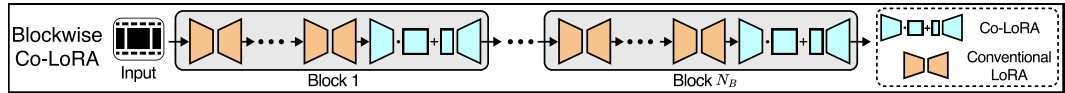

Figure 14: **Illustration of blockwise Co-LoRA.** When a model has $N_B$ Co-LoRA modules, each block employs Co-LoRA at its last layer, while the remaining layers adopt conventional LoRA. Each block contains the same number of layers.

Tab. 16. Consistent with the previous heterogeneous PFL scenario with two different types of architectures (*i.e.*, LLaVA-Llama3.2-1B and LLaVA-Llama3.2-3B), FedMosaic consistently outperforms the baselines in both PFL-Dynamic and PFL-Static settings. These results demonstrate that Co-LoRA effectively enables knowledge sharing across various heterogeneous architectures, highlighting its applicability to real-world scenarios where each client possesses individual heterogeneous models.

| | DRAKE-Dynamic | | | | DRAKE-Static | | | |
| | Self | | Others | | Self | | Others | |
| Method | $A_{last}$ ↑ | $A_{\text{AUC}}$ ↑ | $A_{last}$ ↑ | $A_{\text{AUC}}$ ↑ | $A_{last}$ ↑ | $A_{\text{AUC}}$ ↑ | $A_{last}$ ↑ | $A_{\text{AUC}}$ ↑ |
|---|---|---|---|---|---|---|---|---|
| SFT | 66.41±0.84 | 59.17±0.64 | 47.94±0.18 | 47.10±0.19 | 68.86±2.16 | 62.30±3.12 | 47.98±0.27 | 47.44±0.20 |
| DITTO (ICML 2021) | 61.56±0.01 | 56.36±0.20 | 48.19±0.26 | 47.41±0.01 | 65.48±0.97 | 60.34±0.32 | 48.49±0.03 | 48.09±0.01 |
| FedSim (ICML 2022) | 65.00±0.41 | 58.18±0.46 | 47.47±0.10 | 46.61±0.04 | 67.39±0.32 | 62.42±0.25 | 47.08±0.05 | 47.10±0.03 |
| FedIT (ICASSP 2024) | 66.18±0.22 | 58.95±0.02 | 47.54±0.02 | 46.97±0.01 | 68.78±0.22 | 64.07±0.92 | 47.34±0.55 | 47.33±0.29 |
| TAKFL (NeurIPS 2024) | 64.73±0.49 | 58.33±0.53 | 47.85±0.46 | 47.16±0.08 | 68.17±0.75 | 63.19±1.21 | 47.28±0.14 | 46.90±1.00 |
| FedDPA (NeurIPS 2024) | 63.26±0.09 | 57.31±0.02 | 48.04±0.26 | 47.21±0.07 | 67.71±0.95 | 63.03±0.24 | 48.18±0.53 | 48.22±0.13 |
| FedDAT (AAAI 2024) | 60.43±0.54 | 56.82±0.29 | 49.79±0.21 | 48.25±0.25 | 61.97±0.58 | 57.64±0.45 | 49.76±0.46 | 47.99±0.21 |
| PerAda (CVPR 2024) | 58.10±2.74 | 54.84±1.68 | 46.90±0.22 | 46.69±0.21 | 61.57±0.97 | 56.22±0.38 | 45.01±0.37 | 47.06±0.71 |
| FedMKT (COLING 2025) | 63.25±0.05 | 57.23±0.18 | 47.88±0.29 | 47.25±0.15 | 65.50±0.37 | 61.12±0.56 | 48.27±0.04 | 47.70±0.31 |
| FedMosaic (**Ours**) | **68.94±0.68** | **60.96±0.06** | **52.18±0.34** | **50.11±0.03** | **70.41±1.27** | **65.12±1.15** | **52.67±0.20** | **50.91±0.70** |

Table 16: **Quantitative comparison in heterogeneous PFL with three different types of models.** 'Self' denotes evaluation on a client's own data, while 'Others' denotes evaluation on data from other clients. 3 clients use LLaVA-Llama3.2-1B model, 5 clients use LLaVA-Llama3.2-3B model, and 2 clients use LLaVA-Llama3.1-8B model. SFT refers to supervised fine-tuning on each client's data without cross-client knowledge sharing.

## A.12 EXPERIMENT RESULTS ON HOMOGENEOUS PFL SETUP

In addition to the heterogeneous PFL scenario (Sec. 6.2), where both data distributions and model architectures vary across clients, we also evaluate FedMosaic in a homogeneous PFL setting, where clients share the same model architecture but have heterogeneous data distributions. We assume all clients use LLaVA-3B models and summarize the results in Tab.17. Consistent with results in the heterogeneous setup, FedMosaic significantly outperforms baselines in generalization ability (*i.e.*, 'Others') and even surpasses SFT in personalization performance (*i.e.*, 'Self').

## A.13 EXPERIMENT RESULTS ON QWEN-BASED LLAVA

In addition to heterogeneous PFL scenarios with LLaVA-Llama3 variants, we also compare FedMosaic with baselines using Qwen-based LLaVA models. Specifically, we use LLaVA-Qwen2.5-0.5B for the small model and LLaVA-Qwen2.5-3B for the large model. For baselines, we select the top-3 well-performing baselines in heterogeneous PFL scenarios using LLaVA-Llama3 variants, as well as supervised fine-tuning (*i.e.*, SFT). We summarize the results in Tab. 18.

As shown in the table, FedMosaic significantly outperforms baselines in both homogeneous and heterogeneous PFL scenarios, consistent with results from LLaVA-Llama3 variants. This demonstrates that our proposed Co-LoRA consistently facilitates knowledge sharing across heterogeneous models and RELA reduces interference during local model aggregation, regardless of architecture.

## A.14 EXPERIMENT RESULTS IN CROSS-FAMILY HETEROGENEOUS PFL SCENARIOS

In addition to Tab. 4, we further evaluate FedMosaic under cross-family heterogeneity using clients with either Qwen- and Llama-based LLaVAs on DRAKE-Dynamic. As shown in Tab. 19, FedMosaic consistently outperforms all baselines, demonstrating its generalizability and transferability beyond the same family heterogeneous models (*e.g.*, LLaVA-Llama-1B, LLaVA-Llama-3B).

| | DRAKE-Dynamic | | | | HFLB-Dynamic | | | |
| | Self | | Others | | Self | | Others | |
| Method | $A_{last}$ ↑ | $A_{AUC}$ ↑ | $A_{last}$ ↑ | $A_{AUC}$ ↑ | $A_{last}$ ↑ | $A_{AUC}$ ↑ | $A_{last}$ ↑ | $A_{AUC}$ ↑ |
|---|---|---|---|---|---|---|---|---|
| SFT | 67.57±0.14 | 59.18±1.26 | 49.70±0.02 | 48.04±1.18 | 80.26±1.14 | 77.98±0.49 | 63.24±0.23 | 62.90±0.60 |
| DITTO (ICML 2021) | 64.09±0.55 | 57.22±0.01 | 49.92±0.27 | 49.05±0.07 | 79.34±0.41 | 76.72±0.84 | 64.10±0.40 | 63.57±0.06 |
| FedSim (ICML 2022) | 67.05±0.98 | 60.69±0.84 | 47.94±1.69 | 47.55±1.14 | 79.60±0.77 | 76.93±0.78 | 60.85±1.15 | 60.50±0.82 |
| FedIT (ICASSP 2024) | 67.62±0.49 | 60.35±0.69 | 49.84±0.19 | 48.95±0.13 | 79.34±0.59 | 78.40±1.21 | 62.51±1.30 | 62.38±0.96 |
| TAKFL (NeurIPS 2024) | 67.02±0.04 | 59.86±0.36 | 50.03±0.06 | 48.85±0.09 | 79.45±0.69 | 77.15±0.99 | 63.01±0.66 | 62.70±0.96 |
| FedDPA (NeurIPS 2024) | 66.59±0.86 | 58.68±0.71 | 49.92±0.24 | 48.81±0.13 | 80.57±0.62 | 77.65±0.61 | 63.58±0.30 | 63.02±0.09 |
| FedDAT (AAAI 2024) | 61.95±0.90 | 57.22±0.45 | 51.50±0.13 | 50.28±0.08 | 79.58±0.53 | 76.82±1.13 | 67.33±0.13 | 66.47±0.29 |
| PerAda (CVPR 2024) | 63.78±0.70 | 57.44±0.16 | 49.79±0.25 | 49.10±0.02 | 78.88±0.96 | 76.31±1.24 | 63.35±1.26 | 63.09±0.56 |
| FedMKT (COLING 2025) | 65.20±0.43 | 58.73±0.12 | 49.68±0.29 | 48.84±0.22 | 79.43±0.69 | 77.24±0.61 | 62.66±0.38 | 62.79±0.86 |
| FedMosaic (**Ours**) | **70.80±0.23** | **62.18±0.64** | **53.93±0.36** | **51.72±0.28** | **80.75±0.14** | **78.87±0.20** | **68.30±0.69** | **67.46±0.22** |

| | DRAKE-Static | | | | HFLB-Static | | | |
| | Self | | Others | | Self | | Others | |
| Method | $A_{last}$ ↑ | $A_{AUC}$ ↑ | $A_{last}$ ↑ | $A_{AUC}$ ↑ | $A_{last}$ ↑ | $A_{AUC}$ ↑ | $A_{last}$ ↑ | $A_{AUC}$ ↑ |
|---|---|---|---|---|---|---|---|---|
| SFT | 70.56±0.27 | 66.31±0.37 | 50.37±0.04 | 49.99±0.16 | 80.89±0.85 | 80.01±0.12 | 63.40±0.41 | 63.09±0.17 |
| DITTO (ICML 2021) | 66.19±0.38 | 62.05±0.69 | 50.59±0.05 | 49.95±0.08 | 80.37±0.37 | 79.05±0.25 | 63.93±0.24 | 63.52±0.10 |
| FedSim (ICML 2022) | 68.65±0.86 | 63.96±0.78 | 49.27±0.22 | 48.85±0.14 | 80.05±0.01 | 79.36±0.79 | 61.22±0.30 | 61.31±0.12 |
| FedIT (ICASSP 2024) | 70.14±0.03 | 66.08±0.59 | 50.09±0.15 | 49.87±0.26 | 80.63±1.23 | 79.97±0.01 | 63.34±0.11 | 63.09±0.04 |
| TAKFL (NeurIPS 2024) | 68.75±0.04 | 64.39±0.07 | 49.70±0.16 | 49.60±0.03 | 79.97±0.11 | 79.41±0.81 | 63.67±0.39 | 63.30±0.25 |
| FedDPA (NeurIPS 2024) | 67.40±2.73 | 63.71±1.24 | 50.45±0.38 | 49.92±0.30 | 79.99±1.23 | 79.02±0.35 | 63.45±0.07 | 63.11±0.02 |
| FedDAT (AAAI 2024) | 64.09±1.16 | 61.10±0.60 | 52.20±0.26 | 51.47±0.04 | 79.80±0.50 | 78.43±0.68 | 68.92±0.44 | 67.66±0.27 |
| PerAda (CVPR 2024) | 65.64±0.48 | 61.93±0.58 | 50.53±0.02 | 49.94±0.05 | 80.25±0.45 | 79.01±0.26 | 64.00±0.13 | 63.58±0.31 |
| FedMKT (COLING 2025) | 67.85±0.92 | 62.93±1.16 | 49.23±1.54 | 48.77±1.69 | 79.85±0.67 | 79.72±0.60 | 63.84±0.20 | 63.39±0.07 |
| FedMosaic (**Ours**) | **72.02±0.37** | **67.18±0.13** | **54.21±0.05** | **52.94±0.05** | **81.69±0.16** | **80.54±0.09** | **68.70±0.26** | **67.75±0.23** |

Table 17: **Quantitative comparison in homogeneous PFL.** 'Self' denotes evaluation on a client's own data, while 'Others' denotes evaluation on data from other clients. All clients use LLaVA-3B models. SFT refers to supervised fine-tuning on each client's data without cross-client knowledge sharing.

| | DRAKE-Homonegeous | | | | DRAKE-Heterogeneous | | | |
| | Self | | Others | | Self | | Others | |
| Method | $A_{last}$ ↑ | $A_{AUC}$ ↑ | $A_{last}$ ↑ | $A_{AUC}$ ↑ | $A_{last}$ ↑ | $A_{AUC}$ ↑ | $A_{last}$ ↑ | $A_{AUC}$ ↑ |
|---|---|---|---|---|---|---|---|---|
| SFT | 68.79±1.36 | 59.06±0.18 | 47.87±0.40 | 47.10±0.04 | 65.79±0.17 | 58.24±0.14 | 44.40±0.86 | 44.06±0.43 |
| FedIT (ICASSP 2024) | 68.40±0.07 | 60.58±0.41 | 49.70±0.52 | 48.54±0.07 | 66.02±0.72 | 58.26±1.06 | 44.47±0.30 | 44.06±0.43 |
| FedDPA (NeurIPS 2024) | 66.73±0.42 | 60.21±0.47 | 49.39±0.07 | 48.69±0.11 | 65.43±0.64 | 57.79±0.08 | 44.48±0.43 | 44.53±0.26 |
| FedMKT (COLING 2025) | 65.95±0.35 | 58.81±0.64 | 49.73±0.09 | 48.52±0.24 | 62.85±0.99 | 56.71±0.33 | 44.73±1.35 | 44.24±0.75 |
| FedMosaic (**Ours**) | **70.38±0.34** | **62.84±0.23** | **54.25±0.33** | **51.85±0.20** | **67.36±0.21** | **59.95±0.71** | **50.43±0.01** | **48.51±0.29** |

Table 18: **Quantitative comparison in PFL-Dynamic using Qwen-based LLaVA.** 'Self' denotes evaluation on a client's own data, while 'Others' denotes evaluation on data from other clients. In DRAKE-Homogeneous, all clients use LLaVA-Qwen2.5-3B models, while in DRAKE-Heterogeneous, 3 clients use LLaVA-Qwen2.5-0.5B model, 2 clients use LLaVA-Qwen2.5-1.5B model, and 5 client uses LLaVA-Qwen2.5-3B model. SFT refers to supervised fine-tuning on each client's data without cross-client knowledge sharing.

## A.15 EXPERIMENTAL RESULTS ON TEXT-ONLY BENCHMARKS

In addtion to Fed-LLM-Large (Sec.6.2), we evaluate FedMosaic on other text-only PFL benchmarks, Fed-Aya and Fed-Scope, and summarize the results in Tab. 20. As shown in the table, FedMosaic outperforms baselines in both personalization and generalization, consistent with results from other MLLM PFL benchmarks, such as DRAKE and HFLB.

## A.16 CLIENT-WISE ACCURACY

In addition to reporting the average accuracy across all clients in Sec. 6.2, we report client-wise accuracy in both homogeneous and heterogeneous PFL setups.

**Homogeneous PFL Setups.** We first compare client-wise performance in the homogeneous PFL-Dynamic setups on DRAKE and HFLB, and summarize the results in Tab.21 and Tab.22, respectively. As shown in the tables, personalization performance is improved for all clients compared to SFT (*i.e.*, supervised fine-tuning on local data) except Client 8 in Tab. 22. We highlight this result because SFT is a strong baseline for personalization (Ghari & Shen, 2024), as we also show in Tab. 17, where it outperforms all baselines except for FedMosaic. Moreover, supervised fine-tuning may be

| | Llama 3B / Qwen 1.5B | | | | Llama 1B / Llama 3B / Qwen 1.5B | | | |
| | Self | | Others | | Self | | Others | |
| Method | $A_{last}$ ↑ | $A_{\text{AUC}}$ ↑ | $A_{last}$ ↑ | $A_{\text{AUC}}$ ↑ | $A_{last}$ ↑ | $A_{\text{AUC}}$ ↑ | $A_{last}$ ↑ | $A_{\text{AUC}}$ ↑ |
|---|---|---|---|---|---|---|---|---|
| SFT | 68.41±0.39 | 61.04±0.33 | 48.48±0.15 | 47.75±0.07 | 67.24±0.07 | 60.42±0.33 | 47.48±0.02 | 46.73±0.10 |
| DITTO (ICML 2021) | 64.34±0.35 | 57.72±0.09 | 48.35±0.38 | 47.63±0.22 | 63.78±0.37 | 57.11±0.78 | 47.88±0.29 | 47.09±0.09 |
| FedSim (ICML 2022) | 66.40±0.94 | 59.08±0.53 | 47.32±0.59 | 46.74±0.33 | 65.24±0.21 | 58.33±0.24 | 46.73±0.64 | 46.07±0.41 |
| FedIT (ICASSP 2024) | 67.81±0.73 | 60.46±0.61 | 48.37±0.24 | 47.41±0.06 | 67.21±0.51 | 60.05±0.07 | 47.25±0.04 | 46.62±0.03 |
| TAKFL (NeurIPS 2024) | 65.41±0.27 | 58.52±0.07 | 47.51±0.62 | 47.34±0.32 | 66.13±1.23 | 58.47±0.44 | 47.70±0.83 | 46.70±0.41 |
| FedDPA (NeurIPS 2024) | 65.92±0.49 | 58.73±0.01 | 48.30±0.20 | 47.41±0.19 | 65.93±0.20 | 58.42±0.07 | 47.67±0.08 | 46.64±0.05 |
| FedDAT (AAAI 2024) | 63.86±1.33 | 58.41±0.63 | 50.08±0.34 | 49.23±0.09 | 63.16±1.65 | 57.74±0.78 | 49.13±0.33 | 48.16±0.15 |
| PerAda (CVPR 2024) | 64.11±0.73 | 57.89±0.54 | 48.52±0.51 | 47.76±0.27 | 62.87±1.89 | 57.07±0.62 | 47.54±0.04 | 46.87±0.01 |
| FedMKT (COLING 2025) | 65.38±0.99 | 58.57±0.16 | 47.60±0.49 | 47.29±0.04 | 64.29±1.34 | 58.31±0.43 | 47.13±0.35 | 46.71±0.37 |
| FedMosaic (**Ours**) | **70.62±0.29** | **63.33±0.26** | **52.56±0.08** | **50.76±0.01** | **69.64±0.77** | **62.09±0.51** | **51.69±0.15** | **49.85±0.13** |

Table 19: **Quantitative comparison in cross-family heterogeneous PFL on DRAKE-dynamic.** 'Self' denotes evaluation on a client's own data, while 'Others' denotes evaluation on data from other clients. 'Llama 3B / Qwen 1.5B' experiment is with 6 clients using LLaVA-Llama3.2-3B model and 4 clients using LLaVA-Qwen2.5-1.5B model, while 'Llama 1B / Llama 3B / Qwen 1.5B' experiment is with 2 clients using LLaVA-Llama3.2-1B model, 5 clients using LLaVA-Llama3.2-3B model, and 3 clients using LLaVA-Qwen2.5-1.5B model. SFT refers to supervised fine-tuning on each client's data without cross-client knowledge sharing.

| | Fed-Aya | | | | Fed-Scope | | | |
| | Self | | Others | | Self | | Others | |
| Method | $A_{last}$ ↑ | $A_{\text{AUC}}$ ↑ | $A_{last}$ ↑ | $A_{\text{AUC}}$ ↑ | $A_{last}$ ↑ | $A_{\text{AUC}}$ ↑ | $A_{last}$ ↑ | $A_{\text{AUC}}$ ↑ |
|---|---|---|---|---|---|---|---|---|
| SFT | 2.34±0.08 | 2.17±0.03 | 2.01±0.02 | 1.99±0.01 | 25.21±1.31 | 29.35±0.61 | 25.83±1.03 | 27.30±0.73 |
| FedSim (ICML 2022) | 1.99±0.03 | 1.98±0.04 | 1.92±0.01 | 1.92±0.03 | 21.13±0.97 | 24.03±0.45 | 15.57±0.87 | 18.59±0.30 |
| FedIT (ICASSP 2024) | 2.27±0.08 | 2.18±0.02 | 2.19±0.11 | 2.10±0.04 | 25.06±1.36 | 28.89±0.66 | 25.86±0.94 | 27.94±0.44 |
| TAKFL (NeurIPS 2024) | 2.34±0.12 | 2.17±0.06 | 2.19±0.10 | 2.08±0.03 | 26.18±1.03 | 27.84±0.57 | 28.53±0.33 | 28.27±0.19 |
| FedDPA (NeurIPS 2024) | 2.39±0.09 | 2.27±0.02 | 2.20±0.05 | 2.13±0.01 | 24.93±1.06 | 27.51±0.55 | 26.72±0.67 | 27.69±0.27 |
| FedDAT (AAAI 2024) | 2.14±0.14 | 2.09±0.04 | 1.98±0.08 | 1.96±0.04 | 24.93±0.55 | 27.51±0.28 | 26.72±0.53 | 27.69±0.33 |
| FedMKT (COLING 2025) | 2.31±0.06 | 2.12±0.05 | 2.06±0.02 | 1.96±0.00 | 24.92±0.96 | 27.51±0.63 | 29.55±0.43 | 30.57±0.19 |
| FedMosaic (**Ours**) | **2.51±0.01** | **2.32±0.03** | **2.25±0.01** | **2.17±0.02** | **30.58±0.84** | **32.68±0.64** | **31.01±1.46** | **32.50±0.62** |

Table 20: **Quantitative comparison of heterogeneous LLM clients on the text-only benchmarks.** In Fed-aya experiment, 4 clients use Llama-3.2-1B and 4 clients use Llama-3.2-3B. In Fed-Scope experiment, 3 clients use Llama-3.2-3B and 2 clients use Llama-3.1-8B.

sufficient for personalization in some tasks (Woźniak et al., 2024; Mosbach et al., 2021), as seen with Client 6 in Tab. 21, where there are only marginal improvements compared to SFT. However, for more challenging tasks, properly leveraging knowledge from other clients significantly improves personalization, *e.g.*, FedMosaic shows a gain of 8.2% in $A_{last}$ for Client 9 and 6.4% for Client 7, as well as a 7.1% improvement in $A_{\text{AUC}}$ for Client 5. In summary, even though there is only a 1–3% gain in the average performance across clients, this is because in clients where SFT is sufficient for personalization, the gain from PFL seems smaller. Consistent with the PFL-Dynamic setup, FedMosaic outperforms SFT in the PFL-Static scenario, as shown in Tab. 23.

| | $A_{last}$ / $A_{\text{AUC}}$ | | | | | | | | | |
| Method | Client 1 | Client 2 | Client 3 | Client 4 | Client 5 | Client 6 | Client 7 | Client 8 | Client 9 | Client 10 |
|---|---|---|---|---|---|---|---|---|---|---|
| SFT | 81.90 / 62.11 | 65.23 / 59.28 | 67.38 / 60.31 | 70.28 / 60.40 | 63.69 / 62.07 | 76.05 / 67.35 | 63.40 / 54.17 | 66.93 / 59.31 | 62.70 / 52.20 | 58.16 / 54.58 |
| FedMosaic | **82.73 / 66.11** | **67.31 / 63.76** | **68.97 / 62.53** | **72.60 / 67.51** | **68.67 / 63.58** | **76.98 / 68.56** | **69.83 / 58.86** | **70.11 / 60.08** | **70.89 / 54.89** | **59.88 / 55.89** |
| *Difference* | +0.83 / +4.01 | +4.98 / +1.51 | +2.08 / +4.48 | +1.59 / +2.23 | +2.31 / +7.12 | +0.93 / +1.21 | +6.43 / +4.69 | +3.18 / +0.77 | +8.19 / +2.69 | +1.71 / +1.31 |

Table 21: **Per-client performance of FedMosaic vs. SFT in a homogeneous PFL-Dynamic setup on DRAKE.** The *Difference* row shows the performance gain/loss of FedMosaic compared to SFT. All clients use the LLaVA-Llama3.2-3B model.

| | $A_{last}$ / $A_{\text{AUC}}$ | | | | | | | | |
| Method | Client 1 | Client 2 | Client 3 | Client 4 | Client 5 | Client 6 | Client 7 | Client 8 | Client 9 |
|---|---|---|---|---|---|---|---|---|---|
| SFT | 82.28 / 79.13 | 96.26 / 96.59 | 52.73 / 50.22 | **82.16 / 81.52** | 72.79 / 72.93 | 86.65 / 84.74 | 76.04 / 73.79 | 81.13 / 79.83 | **90.58 / 83.02** |
| FedMosaic | **82.77 / 79.79** | **96.60 / 97.08** | **52.99 / 50.88** | 81.89 / **81/98** | **74.31 / 74.95** | **88.20 / 86.55** | **76.78 / 74.74** | **81.94 / 80.33** | 90.00 / **83.25** |
| *Difference* | +0.49 / +0.67 | +0.34 / +0.48 | +0.26 / +0.67 | -0.27 / +0.46 | +1.52 / +2.02 | +1.55 / +1.81 | +0.74 / +0.94 | +0.81 / +0.50 | -0.58 / +0.23 |

Table 22: **Per-client performance of FedMosaic vs. SFT in a Homogeneous PFL-Dynamic setup on HFLB.** The *Difference* row shows the performance gain/loss of FedMosaic compared to SFT. All clients use the LLaVA-Llama3.2-3B model.

| | $A_{last}$ / $A_{AUC}$ | | | | | | | | |
|---|---|---|---|---|---|---|---|---|---|
| Method | Client 1 | Client 2 | Client 3 | Client 4 | Client 5 | Client 6 | Client 7 | Client 8 | Client 9 |
| SFT | 83.09 / 81.54 | 96.62 / 96.70 | 51.81 / 51.38 | 80.62 / 80.32 | 72.34 / 73.66 | 92.21 / 90.82 | 77.56 / 75.12 | **82.66 / 80.74** | **91.04** / 89.83 |
| FedMosaic | **83.21 / 81.98** | **97.96 / 97.68** | **53.29 / 52.24** | **81.59 / 81.92** | **76.20 / 74.54** | **92.34 / 91.13** | **77.77 / 75.59** | 81.80 / 80.15 | **91.08 / 89.99** |
| *Difference* | +0.12 / +0.44 | +1.33 / +0.98 | +1.48 / +0.86 | +0.97 / +1.60 | +3.86 / +0.87 | +0.13 / +0.31 | +0.21 / +0.46 | -0.86 / -0.59 | +0.05 / +0.16 |

Table 23: **Per-client performance of FedMosaic vs. SFT in a homogeneous PFL-Static setup on HFLB.** The *Difference* row shows the performance gain/loss of FedMosaic compared to SFT. All clients use the LLaVA-Llama3.2-3B model.

**Heterogeneous PFL Setups.** Consistent with the results in the homogeneous PFL setups, FedMosaic enhances personalization ability in most clients, as shown in Tab.24 and Tab.25. As shown in the tables, not only do clients using smaller models (*i.e.*, LLaVA-Llama3.2-1B, LLaVA-Llama3.2-3B) benefit from knowledge sharing through PFL, but clients with larger models (*i.e.*, LLaVA-Llama3.2-8B) also see significant improvements, *e.g.*, a 6.5% improvement in $A_{last}$ and 9.4% improvement in $A_{AUC}$ for Client 9 in Tab. 24. We believe this is due to Co-LoRA effectively sharing knowledge between heterogeneous architectures, allowing both smaller and larger models to provide meaningful information to each other.

| | $A_{last}$ / $A_{AUC}$ | | | | | | | | | |
|---|---|---|---|---|---|---|---|---|---|---|
| | LLaVA-Llama3.2-**1B** | | | LLaVA-Llama3.2-**3B** | | | | | LLaVA-Llama3.1-**8B** | |
| Method | Client 1 | Client 2 | Client 3 | Client 4 | Client 5 | Client 6 | Client 7 | Client 8 | Client 9 | Client 10 |
| SFT | 77.31 / 66.56 | 61.22 / 57.80 | 67.59 / 62.22 | 77.84 / 71.02 | 66.82 / 57.36 | 66.56 / 56.91 | 66.49 / 64.58 | 62.06 / 58.61 | 64.63 / 57.17 | 72.05 / 69.91 |
| FedMosaic | **78.26 / 67.94** | **61.35 / 59.93** | **67.61 / 63.08** | **78.56 / 74.71** | **72.35 / 60.50** | **71.45 / 62.36** | **67.90 / 65.63** | 61.88 / 59.48 | **71.12 / 66.58** | **73.65 / 70.97** |
| *Difference* | +0.95 / +1.37 | +0.13 / +2.13 | +0.02 / +0.85 | +0.73 / +3.70 | +5.53 / +3.14 | +4.89 / +5.45 | +1.41 / +1.05 | -0.19 / +0.87 | +6.49 / +9.41 | +1.60 / +1.05 |

Table 24: **Per-client performance of FedMosaic vs. SFT in a heterogeneous PFL-Static setup on DRAKE.** The *Difference* row shows the performance gain/loss of FedMosaic compared to SFT. Clients 1-3 use LLaVA-Llama3.2-1B, Clients 4-8 use LLaVA-Llama3.2-3B, and Clients 9-10 use LLaVA-Llama3.1-8B.

| | $A_{last}$ / $A_{AUC}$ | | | | | | | | |
|---|---|---|---|---|---|---|---|---|---|
| | LLaVA-Llama3.2-**1B** | | | | LLaVA-Llama3.2-**3B** | | | | |
| Method | Client 1 | Client 2 | Client 3 | Client 4 | Client 5 | Client 6 | Client 7 | Client 8 | Client 9 |
| SFT | 80.69 / 77.18 | **97.41** / 95.25 | 50.95 / 47.72 | 82.39 / 81.49 | 73.99 / 73.00 | **92.10** / 88.23 | 76.58 / 74.19 | 81.00 / 79.94 | 89.32 / 83.12 |
| FedMosaic | **81.69 / 77.53** | 96.46 / **96.18** | **51.38 / 47.85** | **82.70 / 82.41** | **74.49 / 75.09** | 91.80 / **88.36** | **76.95 / 74.92** | **82.24 / 80.25** | **89.94 / 83.62** |
| *Difference* | +1.00 / +0.36 | -0.96 / +0.93 | +0.43 / +0.13 | +0.32 / +0.92 | +0.49 / +2.08 | -0.30 / +0.13 | +0.36 / +0.73 | +1.24 / +0.31 | +0.62 / +0.50 |

Table 25: **Per-client performance of FedMosaic vs. SFT in a heterogeneous PFL-Dynamic setup on HFLB.** The *Difference* row shows the performance gain/loss of FedMosaic compared to SFT. Clients 1-3 use LLaVA-Llama3.2-1B, while Clients 4-9 use LLaVA-Llama3.2-3B.

## A.17 DETAILS OF PUBLIC DATA $D_p$

For the public data $D_p$, we use a subset of the MLLM's pretraining data, as briefly mentioned in Sec. 4.2.2. Specifically, since our experiments are based on LLaVA, we adopt its instruction tuning dataset, LLaVA-Instruct-158K (Liu et al., 2023b), as $D_p$. For the alignment details, we randomly sample 5,000 examples from $D_p$ and align Co-LoRA for 1 epoch using a batch size of 4, a learning rate of $5 \times 10^{-5}$, and the Adam optimizer with cosine scheduler. To further assess the effect of $D_p$, we explore the effect of $D_p$'s size and its distributional alignment with the client data.

We first assess the effect of public data size in Tab 26. As shown, using too little data results in insufficient alignment of Co-LoRA, leading to degraded performance. Interestingly, using too much public data also causes slight performance drops, likely due to overfitting to the $D_p$ distribution, which hinders generalizable alignment to clients' distributions. Considering this trade-off, we selected a moderate dataset size (*i.e.*, 5,000 samples).

Next, we assess robustness under domain mismatch between $D_p$ and the clients' distribution. Specifically, we use ChatbotIT (Zheng et al., 2023) (text-only NLP benchmark) and Visual Storytelling (Li et al., 2024d) (multi-sentence outputs) as $D_p$, both differing from the clients' multi-modal, short-answer tasks. To ensure a fair comparison, we fix the size of $D_p$ to 5,000 samples. As shown in Tab. 27, using Visual Storytelling shows comparable performance with LLaVA-Instruct-158K. Similarly, using text-only NLP benchmark shows performance comparable to (and even surpassing)

| Size of $D_p$ | Self | | Others | |
|---|---|---|---|---|
| | $A_{last}$ ↑ | $A_{\text{AUC}}$ ↑ | $A_{last}$ ↑ | $A_{\text{AUC}}$ ↑ |
| 625 | 67.25±0.08 | 59.17±0.05 | 51.02±0.04 | 49.05±0.09 |
| 1250 | 67.97±0.04 | **59.86±0.16** | 51.33±0.03 | **49.58±0.07** |
| 2500 | **68.11±0.06** | 59.76±0.04 | 51.35±0.31 | 49.52±0.13 |
| 5000 | 67.96±0.05 | 59.83±0.15 | **51.46±0.04** | 49.56±0.06 |
| 10000 | 67.44±0.18 | 59.34±0.16 | 51.04±0.16 | 49.37±0.03 |

Table 26: **Effect of public data $D_p$ size.** FedMosaic performs best on moderate-sized datasets (*i.e.*, 1,250-5,000)

| Dataset $D_p$ | Self | | Others | |
|---|---|---|---|---|
| | $A_{last}$ ↑ | $A_{\text{AUC}}$ ↑ | $A_{last}$ ↑ | $A_{\text{AUC}}$ ↑ |
| LLaVA-Instruct-158K (Liu et al., 2023b) | 67.96±0.05 | 59.83±0.15 | **51.46±0.04** | 49.56±0.06 |
| ChatbotIT (Zheng et al., 2023) | **68.42±0.18** | **60.17±0.26** | 51.16±0.22 | **49.78±0.13** |
| Visual Storytelling (Li et al., 2024d) | 67.90±0.09 | 59.86±0.12 | 51.06±0.05 | 49.41±0.07 |

Table 27: **Effect of domain gap between $D_p$ and clients' data.** FedMosaic shows consistent performance across multiple benchmarks, even on text-only NLP benchmarks (*i.e.*, ChatbotIT).

LLaVA-Instruct-158K, demonstrating resilience of FedMosaic to distribution misalignment between $D_p$ and clients' data.

## A.18    HYPERPARAMETER ANALYSIS

Since dataset-specific hyperparameter search is undesirable under distribution shifts (*i.e.*, Dynamic setup), where future data are unknown, we adopt a single set of hyperparameters across all benchmarks and setups, determined from the DRAKE-Dynamic setup.

**Effect of Communication Rounds $R$ and Local Steps.**    We analyze the trade-off by varying the number of communication rounds and local steps under a fixed training budget. As shown in Tab. 28, more rounds (*i.e.*, fewer local steps per round) degrade personalization ('Self'). We attribute this to insufficient local adaptation, which weakens the personalized models and reduces the quality of shared knowledge during communication, ultimately limiting improvements in generalizability. Conversely, significantly reducing rounds (*i.e.*, increasing local steps) improves personalization but harms generalization ('Others') due to limited inter-client knowledge sharing, reducing generalizability. To balance this trade-off, we adopt a moderate setting with 20 rounds and 100 local steps per round.

**Effect of Low Rank $r$ in Co-LoRA.**    We study how varying the rank $r$ in Co-LoRA affects performance, and summarize the results in Fig. 15. As shown in the figure, while both excessively high and low values of $r$ lead to degraded performance, a broad range of intermediate values shows stable performance. While a larger $r$ increases shareable capacity ($\mathbb{R}^{r \times r}$), it may cause overfitting (Lin et al., 2024; Cho et al., 2024). In contrast, a smaller $r$ helps mitigate overfitting but may introduce capacity limitations, resulting in suboptimal performance (He et al., 2022). By balancing the trade-off, we select an adequate low rank $r = 128$.

**Effect of $N_B$ in Co-LoRA.**    We study the effect of $N_B$, the number of layers employing Co-LoRA, and summarize the results in Tab. 29. In a model $W$ with $|W|$ layers, increasing $N_B$ reduces the layers using conventional LoRA to $|W| - N_B$. Since Co-LoRA has fewer trainable parameters (*i.e.*, $r^2 + r$) than conventional LoRA (*i.e.*, $r \times (d_I + d_O)$), increasing $N_B$ decreases the total number of learnable parameters in $W$, as $A \in \mathbb{R}^{r \times d_I}$ and $B \in \mathbb{R}^{d_O \times r}$ are frozen, with only $P \in \mathbb{R}^{r \times r}$ and $Q \in \mathbb{R}^r$ being trainable. Despite the reduced number of parameters, as shown in the SFT performance in the table, performance remains stable even as $N_B$ increases. We attribute this to orthogonal and frozen $A$ and $B$, which maximize and preserve the expressiveness of Co-LoRA (Theorem 1), despite the lower trainable parameter count.

However, there is a trade-off in $N_B$ in the PFL setup: increasing $N_B$ improves generalizability (*e.g.*, $N_B = 2$ vs. $N_B = 4$ in $A_{\text{AUC}}$ of 'Others' in FedMosaic) but can degrade personalization (*e.g.*, $N_B = 4$ vs. $N_B = 8$ in $A_{\text{last}}$ and $A_{\text{AUC}}$ of 'Self' in FedMosaic). This is because, while increasing

| Total Rounds | Local Steps per Round | Self | | Others | |
|---|---|---|---|---|---|
| | | $A_{last}$ ↑ | $A_{\text{AUC}}$ ↑ | $A_{last}$ ↑ | $A_{\text{AUC}}$ ↑ |
| 5 | 400 | 66.66±0.89 | **60.21±0.61** | 47.36±0.13 | 46.65±0.01 |
| 10 | 200 | 65.66±0.47 | 58.58±0.29 | 47.44±0.03 | 46.63±0.26 |
| 20 | 100 | **67.86±0.51** | 59.83±0.15 | **51.36±0.04** | 49.46±0.06 |
| 40 | 50 | 67.05±0.54 | 59.49±0.11 | 51.10±0.11 | 49.42±0.02 |
| 80 | 25 | 66.83±0.38 | 59.23±0.06 | 51.16±0.04 | **49.47±0.21** |

Table 28: **Effect of communication rounds and local steps on DRAKE under Fixed Total Training Cost (*i.e.*, Total rounds × Local steps per round).**

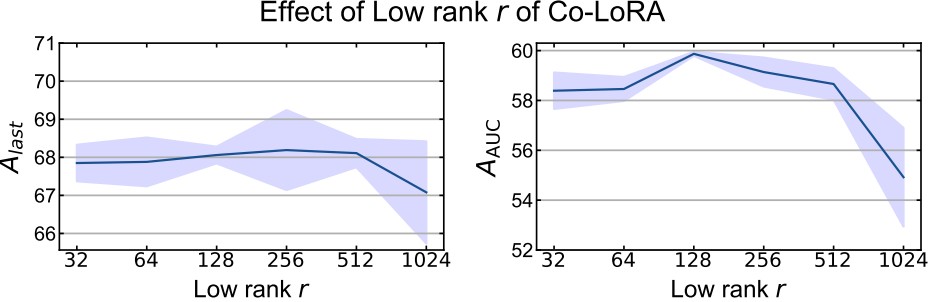

Figure 15: **Accuracy on different low rank $r$ values in Co-LoRA on DRAKE-Dynamic setup.** Extremely low or high ranks degrade performance, but a wide intermediate range maintains stable accuracy.

$N_B$ enables heterogeneous models to share knowledge across more layers, the number of trainable parameters for local training decreases due to the reduction in the number of conventional LoRA modules. As a result, by balancing this trade-off, we employ a moderate value of $N_B$, *i.e.*, $N_B = 4$.

| Method | Self | | Others | |
|---|---|---|---|---|
| | $A_{last}$ ↑ | $A_{\text{AUC}}$ ↑ | $A_{last}$ ↑ | $A_{\text{AUC}}$ ↑ |
| SFT ($N_B = 2$) | 66.14±0.02 | 57.02±0.09 | 47.75±0.42 | 46.02±0.14 |
| SFT ($N_B = 4$) | 66.24±0.68 | 57.71±0.27 | 47.63±0.15 | 46.62±0.14 |
| SFT ($N_B = 8$) | 66.21±0.62 | 57.87±0.20 | 47.86±0.29 | 46.76±0.20 |
| FedMosaic ($N_B = 2$) | **68.07±0.10** | **59.90±0.68** | 51.07±0.06 | 48.78±0.15 |
| FedMosaic ($N_B = 4$) | 67.86±0.51 | 59.83±0.16 | 51.26±0.04 | **49.36±0.08** |
| FedMosaic ($N_B = 8$) | 67.52±0.77 | 59.39±0.23 | **51.27±0.27** | 49.26±0.14 |

Table 29: **Accuracy on different number of blocks ($N_B$) in Co-LoRA in heterogeneous PFL-dynamic setup on DRAKE.** Given a model $W$ with $|W|$ layers, using $N_B$ blocks means that $N_B$ layers adopt Co-LoRA, while the remaining $|W| - N_B$ layers use conventional LoRA.

**Effect of Orthogonality-regularization Scale $\lambda$ in Co-LoRA.** We study how varying the orthogonality-regularization scale $\lambda$ in Co-LoRA alignment process (Eq. 20) affects performance, and summarize the results in Fig. 16. We observe that the 'Self' performance does not fluctuate on larger $\lambda$, but reduces for small $\lambda$. This indicates that insufficient weighting of the orthogonality-regularization term weakens the enforced orthogonality among $A$ matrices from heterogeneous models, thereby reducing the representational capacity (*i.e.*, Span) of Co-LoRA, as mentioned in Theorem 1.

**Effect of Temperature $\tau$ in RELA.** We analyze the effect of the softmax temperature $\tau$ in RELA (Eq. 4), which controls the aggregation sharpness, *i.e.*, lower $\tau$ focuses aggregation on similar clients, while higher $\tau$ aggregates weights across more diverse ones. Interestingly, as shown in Fig. 17, neither decreasing nor increasing the temperature $\tau$ consistently improves personalization or generalization. Instead, we observe two trends: (i) Extremely low $\tau$ degrades both personalization and generalization (ii) Increasing $\tau$ too much reduces personalization, while generalization gradually stabilizes.

This is because low $\tau$ limits the diversity of aggregation, restricting knowledge sharing even among moderately relevant clients. In contrast, high $\tau$ encourages sharing across dissimilar tasks, which can

Effect of Orthogonality regularization scale $\lambda$

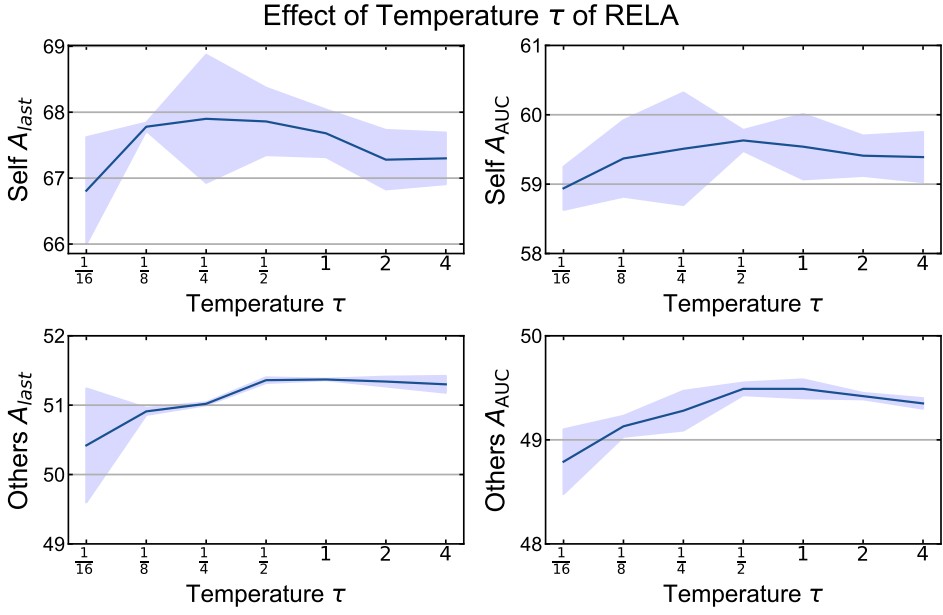

Figure 16: **Accuracy on different L2 regularization scale $\lambda$ values in Co-LoRA on DRAKE-Dynamic setup.** Extremely low $\lambda$ degrades performance, but a wide intermediate range maintains stable accuracy.

reduce personalizability due to the inclusion of unrelated knowledge in the global model. However, increasing $\tau$ does not continuously improve generalization; it converges after a certain point. We believe this is because, although a higher $\tau$ promotes aggregation of more diverse knowledge, it also increases parameter interference (*e.g.*, sign conflicts) when combining models trained on distinct tasks (Yeh et al., 2023; Ding et al., 2024). Consequently, we adopt a moderate temperature of $\tau = 0.5$ for all experiments.

Figure 17: **Accuracy under different temperatures $\tau$ on DRAKE-Dynamic setup.** Extremely low or high temperatures degrade performance, but a wide intermediate range maintains stable accuracy.

**Effect of Gradient sampling ratio $N_s$ in RELA.** We analyze the effect of gradient sampling ratio $N_s$ and summarize the results in Fig. 18. As shown, sampling only a very small number of dimensions from the decayed client-wise gradient $\hat{g}_i$ to construct the sanitized gradient $\tilde{g}_i$ significantly degrades performance, as it undermines the representativeness and informativeness of the compressed gradient vector (Li et al., 2024b). However, using up to $N_s = 40\%$ of the gradient dimensions maintains comparable performance while reducing both communication costs and privacy risks from gradient inversion. Accordingly, we consistently set $N_s = 40\%$ for RELA in all experiments.

**Effect of decaying EMA ratio $\alpha$ in RELA.** We analyze the effect of gradient deacying EMA ratio $\alpha$ and summarize the results in Fig. 19. Too small $\alpha$ ignores current task information, while too large

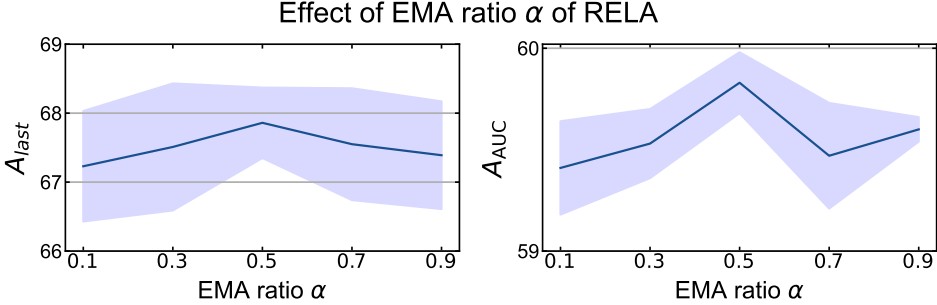

Figure 18: **Accuracy under different gradient sampling ratio $N_s$ on DRAKE-Dynamic setup.** Extremely low gradient sampling ratio degrades performance, but a wide range maintains stable accuracy.

$\alpha$ ignores previously learned tasks. Balancing the trade-off, we consistently set $\alpha = 0.5$ for RELA in all experiments.

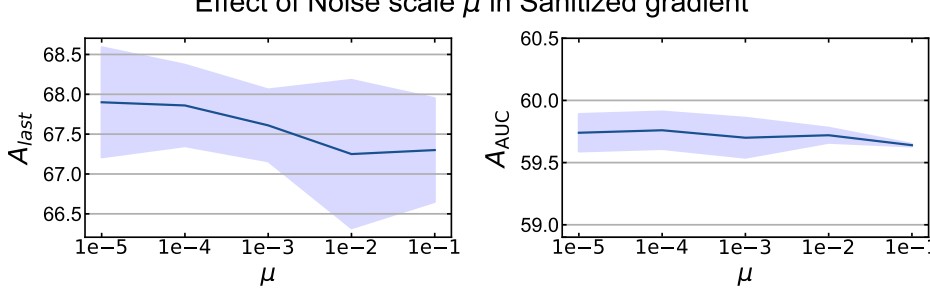

Figure 19: **Accuracy under different gradient decaying EMA ratio $\alpha$ on DRAKE-Dynamic setup.** Balance between current and old ($\alpha = 0.5$) shows the best result.

**Effect of noise scale $\mu$ in RELA sanitized gradient.** We also study the noise scale $\mu$ in sanitized gradient $\tilde{g}$ (Eq. 3) in RELA and visualize the results in Fig. 20. We clearly see the decreasing trend in the results when we increase the noise scale. While stronger random noise enhances privacy by better concealing sensitive information, excessive noise severely distorts the gradient signal, undermining the reliable estimation of task relevance based on gradient similarity.

### Effect of Noise scale $\mu$ in Sanitized gradient

Figure 20: **Accuracy under different noise scale $\mu$ on DRAKE-Dynamic setup.** Strong noise distorts the gradient signal, undermining the reliable estimation of task relevance based on gradient similarity.

**Effect of Gradient computing interval $m$ in RELA.** We analyze the effect of computing interval (*i.e.*, computing on every $m$ batches) of client-wise gradient $\hat{g}_i$ during local training in both performance and computational cost, and summarizes the results in Fig. 21. As shown in Fig. 21, computing gradients more frequently (*i.e.*, smaller $m$) improves performance but increases computational cost,

whereas computing them too infrequently (*i.e.*, larger $m$) reduces cost but significantly degrades accuracy. Since a wide range of $m$ values (*i.e.*, 1~20) maintains stable performance, we adopt $m = 10$ as a balanced choice that offers comparable accuracy with reduced computational overhead.

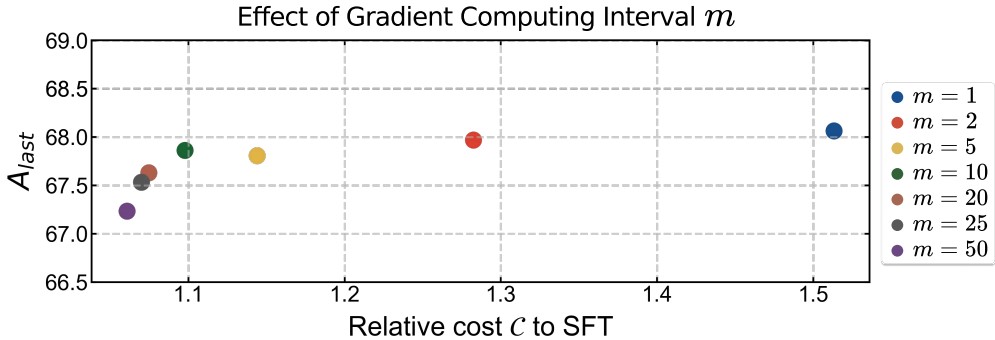

Figure 21: **Accuracy *vs*. Computational Cost under different gradient computing interval $m$ on DRAKE-Dynamic setup.** Computing gradient on every $m = 10$ batches during local training shows the best balance between the performance and computational cost.

**Effect of choice of Gradient layer in RELA.** We also analyze how the choice of model layers from which per-client gradient vectors are collected for RELA affects performance, computational cost, and communication cost. As shown in Fig. 22-(a), employing only the last 1 layer achieves comparable performance to employing gradients of more layers (*e.g.*, Last 8 layers, All layers) with incurring minimal computational cost. Moreover, utilizing gradients from more layers increases communication cost, as the dimensionality of the vectors transmitted from clients to the server grows accordingly. As shown in Fig. 21-(b), this leads to higher communication overhead when more layers are included. Based on this empirical evidence, which demonstrates that the last-layer gradient provides a cost-efficient approximation to full-layer gradients while achieving nearly identical performance, we track only the last-layer gradient in RELA.

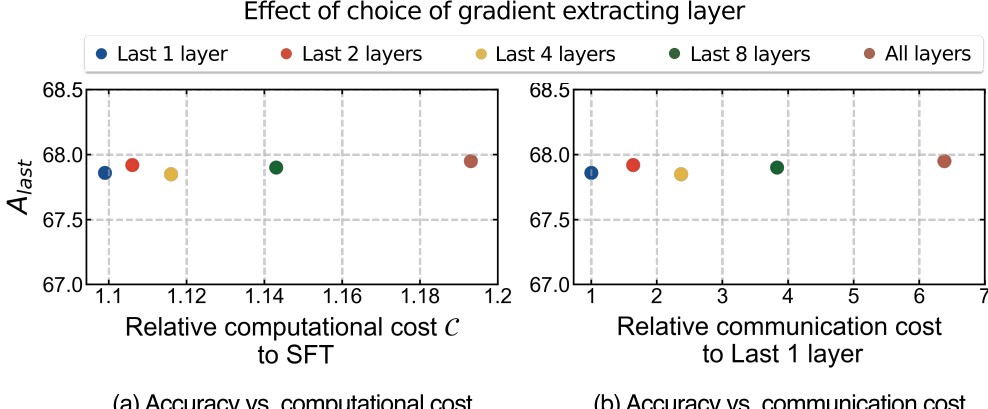

(a) Accuracy vs. computational cost    (b) Accuracy vs. communication cost

Figure 22: **Accuracy *vs*. Computational/Communication Cost under different gradient extracting layers on DRAKE-Dynamic setup.** Note that we use the last Linear layer parameter for 'Last 1 layer' while using all parameters in Transformer layers for others. The results demonstrate that the last 1 layer gradient provides a cost-efficient approximation to full-layer gradients, achieving nearly identical performance.

## A.19 ABLATION STUDY OF RELA

Our proposed RELA measures task similarity among clients using client-wise gradients to construct a similarity-aware customized global model for each client. Specifically, for the $i$-th client, we maintain a decayed gradient $\hat{g}_i$, updated via exponential moving average (EMA) from the current gradient $g_i$,

which is computed using the current batch and a small, frozen pre-trained model. Note that we use the last-layer gradient of the frozen model, not the training model, as mentioned in Sec. 4.1.

We ablate RELA by comparing four aggregation variants: (1) Equal-weight aggregation, which ignores task similarity and aggregates local models uniformly; (2) Training model gradients, using current batch gradients from the trainable model; (3) Frozen model gradients, using current batch gradients from the frozen pre-trained model; and (4) Decayed frozen gradient (*i.e.*, RELA), which maintains an EMA of gradients from the frozen pre-trained model to capture model knowledge shifts. We summarize the results in Tab. 30.

As shown in the table, model aggregation considering task similarity measured by client-wise gradients generally improves the 'Self' accuracy. This demonstrates the loss of information in Equal-weight aggregation due to parameter interference. While using gradients from the training model ('Training model gradient') shows improved performance compared to Equal-weight aggregation, it still suffers from data heterogeneity, as gradient similarities from models trained on heterogeneous data may not capture actual similarity (Tang et al., 2020; Evans et al., 2024). Using a frozen model's gradients from the current batch ('Frozen model gradients') can address this limitation, but cannot reflect model knowledge shifts under distribution shifts, as mentioned in Sec. 4.1. In contrast, RELA effectively captures task similarity under data heterogeneity with distribution shifts, thus outperforming other aggregation strategies.

| Method | Self | |
| --- | --- | --- |
| | $A_{last}$ ↑ | $A_{\text{AUC}}$ ↑ |
| Equal-weight aggregation | 66.99±0.77 | 58.39±0.38 |
| Training model gradients | 67.16±0.74 | 59.38±0.53 |
| Frozen model gradients | 67.24±1.27 | 59.45±0.74 |
| Decayed frozen gradients (**RELA**) | **67.86±0.51** | **59.83±0.16** |

Table 30: **Ablation of components in RELA in Heterogeneous PFL-dynamic setup on DRAKE.** 'Equal-weight aggregation' refers to aggregating local models with equal weight, 'Training model gradients' refers to using gradients from the current batch computed with the training model, 'Frozen model gradients' refers to using gradients from the current batch computed with the frozen pre-trained model, and 'Decayed frozen gradients' refer to EMA-estimated gradients from the frozen model (*i.e.*, RELA).

## A.20 RELA WITH LIGHTER $W_s$

To extract per-client gradients for RELA, we employ the smallest MLLM among all clients' models (*e.g.*, LLaVA-1.5-1B). To further reduce the computational overhead of RELA, we can utilize an even smaller model (*e.g.*, LLaVA-Qwen-0.5B) or apply lower-bit quantization (*e.g.*, 4-bit and 8-bit), as shown in Tab. 31. The results show that FedMosaic maintains consistent performance even with smaller models and lower-bit quantization, demonstrating that computational costs can be further reduced without sacrificing performance.

| $W_s$ | Relative computational costs ↓ | Self | | Others | |
| --- | --- | --- | --- | --- | --- |
| | | $A_{last}$ ↑ | $A_{\text{AUC}}$ ↑ | $A_{last}$ ↑ | $A_{\text{AUC}}$ ↑ |
| LLaVA-Llama3.2-1B (16bit) | 1.0 | **67.86±0.51** | **59.83±0.15** | 51.16±0.04 | 49.36±0.08 |
| LLaVA-Qwen-0.5B (16bit) | 0.52 | 67.84±0.19 | 59.42±0.75 | 51.40±0.12 | 49.37±0.19 |
| LLaVA-Qwen-0.5B (8bit) | 0.26 | 67.36±0.14 | 59.22±0.43 | **51.74±0.06** | **49.56±0.07** |
| LLaVA-Qwen-0.5B (4bit) | 0.13 | 67.23±0.49 | 59.41±0.13 | 51.34±0.42 | 49.43±0.09 |

Table 31: **Effect of $W_s$'s model size and quantization on DRAKE-Dynamic.** Relative computational costs denote the ratio of computation compared to LLaVA-1.5-1B (16-bit). Comparable performance is maintained with both a smaller model (*i.e.*, 0.5B) and quantized models (*i.e.*, 4-bit, 8-bit).

## A.21 DIFFERENT WEIGHT ALIGNMENT METHODS IN CO-LORA

To ensure shared initialization across heterogeneous architectures, we align $A$ matrices from heterogeneous architectures (*i.e.*, $A_i \in \mathbb{R}^{r \times d_i}$, $A_j \in \mathbb{R}^{r \times d_j}$) using L2 loss, and $B$ matrices (*i.e.*, $B_i \in \mathbb{R}^{d'_i \times r}$, $B_j \in \mathbb{R}^{d'_j \times r}$) using canonical correlation analysis (CCA), as detailed in Sec. 4.2.2. We ablate the effects of aligning $A$ and $B$ matrices, and summarize the results in Tab. 32. Note that the results in the table report the average performance across all clients, which can make the overall average improvements appear small, as mentioned in Sec. 6.2.

As shown, alignment improves both personalization ('Self') and generalization ('Others') performance. This occurs because aligning both $A$ and $B$ in Co-LoRA ensures heterogeneous models share initialization and follow consistent optimization paths, allowing aggregation without weight interference (Wortsman et al., 2022a;b; Yadav et al., 2023). In contrast, misalignment in either matrix introduces initialization discrepancies between models, leading to weight interference during aggregation and degrading performance (Jordan et al., 2023; Neyshabur et al., 2020; Stoica et al., 2025).

| Initialization | | Self | | Others | |
|---|---|---|---|---|---|
| A | B | $A_{last}$ ↑ | $A_{\text{AUC}}$ ↑ | $A_{last}$ ↑ | $A_{\text{AUC}}$ ↑ |
| Random | Random | 67.95±1.71 | 63.50±0.66 | 47.71±0.08 | 47.46±0.05 |
| Random | Aligned (CCA) | 68.22±1.84 | 63.59±0.62 | 47.80±0.06 | 47.56±0.08 |
| Aligned (L2) | Random | 68.29±1.57 | 63.58±0.60 | 47.72±0.14 | 47.41±0.04 |
| Aligned (L2) | Aligned (CCA) | **68.73±1.92** | **63.82±0.59** | **48.04±0.04** | **47.57±0.08** |

Table 32: **Accuracy on different weight alignment in Co-LoRA under Heterogeneous PFL-Static setup on DRAKE.** 'Random' initializes matrices randomly, while Aligned (L2) and Aligned (CCA) use matrices aligned by L2 loss and canonical correlation analysis (CCA), respectively.

## A.22 COMPARISON WITH SIMILARITY-AWARE MODEL AGGREGATION

**Federated Learning**  We are not the first to try to aggregate client models based on task similarity, but most existing approaches are not applicable to large language model (LLM) federated learning scenarios. pFedSim (Tan et al., 2023) measures the classifier similarity among clients and uses it for weighted model aggregation, which is infeasible for LLM fine-tuning where the pre-trained classifier remains frozen. pFedHR (Wang et al., 2023) computes similarity between client models based on the output logits and performs similarity-aware layer stitching. However, its logit extractions relies on public data, which (as acknowledged by the authors) raises public data sensitivity concerns: the discrepancy between private and public data may leads to inaccurate similarity estimate. Flashback (Aljahdali et al., 2024) conducts dynamic knowledge distillation where teacher logits are adaptively weighted by client-wise label count, but this method is limited to single-dataset classification FL tasks. There is also another approach like FEDLAW (Li et al., 2023d) that adaptively aggregates models based on the optimized learnable weights instead of client similarity. However, it incurs high computational costs and remains highly sensitive to the choice of public data.

RELA, on the contrary, measures the task similarity among clients using sanitized last layer gradients. These gradients from each private data effectively capture task characteristics with minimal additional computation and communication cost, while preserving privacy through EMA updates, noise injection, and gradient compression.

**Multi-Task Learning**  Federated learning (FL) and personalized federated learning (PFL) are not the only paradigms that learn heterogeneous tasks concurrently. Multi-task learning (MTL) trains a single model on multiple tasks simultaneously in a single compute node, not in separate private nodes as in FL. MTL methods, such as NBS (Navon et al., 2022) and Rotograd (Javaloy & Valera, 2022), leverage inter-task relationships to dynamically weight gradients, thereby mitigating interference and promoting a balanced update direction across tasks. This is similar to the motivation of our aggregation method, *i.e.*, RELA, which aims to reduce the interference between multiple tasks. However, these techniques are not directly applicable to personalized federated learning due to differing objectives: similarity-aware MTL strategies target *generalizability* by reshaping task-specific gradients to improve a single shared model, whereas RELA focuses on *personalization*. It

provides each client with a customized global model by down-weighting contributions from irrelevant tasks, rather than enforcing a single globally shared one, thereby providing knowledge beneficial for personalization, while still enhancing generalizability as a byproduct.

We also empirically compare RELA with MTL methods (Navon et al., 2022). Note that Rotograd(Javaloy & Valera, 2022) is incompatible with PFL because it aligns gradient directions by optimizing rotation heads using samples from multiple tasks, which is not feasible under federated data isolation and privacy constraints. Therefore, we only compare with NBS, which determines task weights via a Nash bargaining objective that maximizes total loss reduction. In PFL setting, we apply NBS weighting to adapter parameters rather than raw gradients for a shared global model, since FL aggregates parameters. As shown in Tab. 33, NBS shows lower 'Self' performance, while maintaining similar generalization ('Others') performance. We believe NBS's weighting mechanism focuses on harmonizing parameters from multiple tasks to minimize the overall loss, but fails to preserve or share task-relevant information necessary for each client's personal tasks, resulting in poor 'Self' performance.

| Method | Self | | Others | |
| --- | --- | --- | --- | --- |
| | $A_{last}$ ↑ | $A_{\text{AUC}}$ ↑ | $A_{last}$ ↑ | $A_{\text{AUC}}$ ↑ |
| FedMosaic w/ RELA | **67.86±0.51** | **59.83±0.15** | **51.16±0.04** | **49.36±0.08** |
| FedMosaic w/ NBS (Navon et al., 2022) | 66.89±0.43 | 58.98±0.20 | 51.07±0.09 | 49.09±0.06 |

Table 33: **Quantitative comparison with MTL method.**

## A.23 EXTENDED FAST ADAPTATION EVALUATIONS

In addition to the fast adaptation evaluation in Sec.6.2, we further assess fast adaptation on additional unseen tasks from DRAKE. We summarize the results in Fig.23. As shown in the figure, models initialized with FedMosaic adapt significantly faster than those with random initialization or other PFL baselines, demonstrating that FedMosaic enhances generalizability by effectively sharing knowledge in heterogeneous PFL setups.

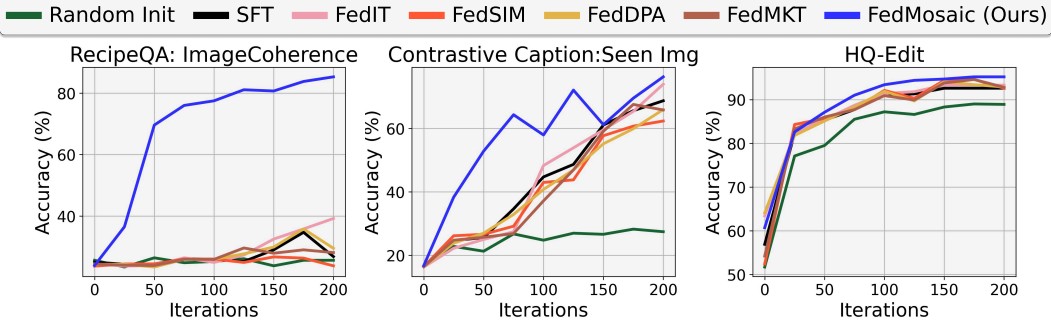

Figure 23: **Comparison of adaptation speed.** We use FedMosaic's unseen tasks as downstream tasks. Random init starts from randomly initialized models, while other baselines are initialized from aggregated local models trained on DRAKE using each respective FL baseline.

## A.24 EXPERIMENTAL RESULTS ON RANK HETEROGENEOUS PFL SCENARIOS

In the main experiments, we assume that all clients use the same LoRA rank $r$, which ensures the consistent dimensions for $P \in \mathbb{R}^{r \times r}$ and $Q \in \mathbb{R}^r$ in Co-LoRA across heterogeneous models. It enables knowledge sharing across heterogeneous models, but it also imposes the constraint that all clients must adopt the same rank $r$. To support knowledge sharing among heterogeneous architectures with both different parameter dimensions and different LoRA ranks, Co-LoRA can be readily combined with prior heterogeneous FL methods (Cho et al., 2024) that allow clients to use LoRA modules with heterogeneous ranks but shared dimensions.

**HETLoRA (Cho et al., 2024).** HETLoRA proposes to align the rank dimensions of heterogeneous local LoRA modules by zero-padding, aggregate them with sparsity-based weighting, and finally distribute the aggregated update using truncation. For example, assume two clients $\mathcal{I} = \{i, j\}$ have heterogeneous LoRA adapters, $A_i \in \mathbb{R}^{d \times r_i}$, $B_i \in \mathbb{R}^{r_i \times d}$ and $A_j \in \mathbb{R}^{d \times r_j}$, $B_j \in \mathbb{R}^{r_j \times d}$, where $r_i < r_j$. HETLoRA first zero-pads the smaller-rank LoRA module to $r_{\max} = \max\{r_i, r_j\}$, e.g., zero-pad $A_i[:, r_i : r_{\max}]$ and $B_i[r_i : r_{\max}, :]$. Then, it aggregates the LoRA modules as

$$\bar{A} = \sum_{k \in \mathcal{I}} \alpha_k A_k, \qquad \bar{B} = \sum_{k \in \mathcal{I}} \alpha_k B_k, \tag{24}$$

where the aggregation weights $\alpha_k$ are defined proportional to the Frobenius norm $\|\Delta W_k\|_F$ of the reconstructed full-rank update $\Delta W_k = B_k A_k$, e.g., $\alpha_i = \frac{\|\Delta W_i\|_F}{\sum_{k \in \mathcal{I}} \|\Delta W_k\|_F}$. Finally, the aggregated global LoRA module is distributed to each client via dimension truncation to match each client's LoRA rank. For example, for client $i$ having rank $r_i < r_{\max}$ and LoRA matrix $B_i \in \mathbb{R}^{d \times r_i}$, the client keeps the first $r_i$ columns of the aggregated $\bar{B}$ matrix and discards the remaining $r_{\max} - r_i$ columns.

**Combining HETLoRA with Co-LoRA.** We integrate HETLoRA with Co-LoRA to enable knowledge sharing among clients with both model and rank heterogeneity. For example, consider two clients $\mathcal{I} = \{i, j\}$ with heterogeneous $P$ and $Q$ modules in Co-LoRA, *i.e.*, $P_i \in \mathbb{R}^{r_i \times r_i}$, $Q_i \in \mathbb{R}^{r_i}$ and $P_j \in \mathbb{R}^{r_j \times r_j}$, $Q_j \in \mathbb{R}^{r_j}$, with $r_i < r_j$. Similar to HETLoRA's aggregation process, we zero-pad $P_i, Q_i$ to rank $r_{\max}(= r_j)$, *i.e.*, zero-pad $P_i[r_i : r_{\max}, r_i : r_{\max}]$ and $Q_i[r_i : r_{\max}]$. With the rank dimension aligned $P$ and $Q$ modules, we aggregate them to obtain the global module $\bar{P}$ and $\bar{Q}$ by combining Eq. 24 and RELA as

$$\bar{P}_i = \sum_{j=1}^{N} \hat{w}_{ij}\, P_j, \quad \bar{Q}_i = \sum_{j=1}^{N} \hat{w}_{ij}\, Q_j, \quad \hat{w}_{ij} = \frac{w_{ij}\alpha_j}{\sum_{n=1}^{N} w_{in}\alpha_n}, \tag{25}$$

where $w_{ij}$ is the client-wise task relevance weight in RELA in Eq. 4. Finally, the $\bar{P}$ and $\bar{Q}$ are distributed to each client via dimension truncation to match each client's rank, *e.g.* truncating the $r_{\max} - r_i$ rows and columns of $\bar{P}$ and $r_{\max} - r_i$ rows of $\bar{Q}$.

With the combination of two orthogonal components, *i.e.*, Co-LoRA and HETLoRA, which enable knowledge sharing under both model and rank heterogeneity, we evaluate a setting where clients use different models with heterogeneous LoRA ranks. Specifically, clients adopt heterogeneous models (LLaVA-Llama3.2-1B or LLaVA-Llama3.2-3B) and three different LoRA ranks $\{64, 128, 256\}$. For a fair comparison, we also compare against other PFL baselines combined with HETLoRA. As shown in Tab. 34, FedMosaic combined with HETLoRA significantly outperforms all baselines with the same HETLoRA integration. These results highlight that Co-LoRA is not limited to heterogeneous architectures with a shared rank, but can also handle heterogeneous ranks across heterogeneous architectures when paired with existing hetero-rank LoRA FL approaches, further extending its applicability to real-world clients.

## A.25 Experimental Results on Non-I.I.D. Heterogeneous PFL Scenarios

We further validate the effectiveness of FedMosaic on standard non-i.i.d. personalized federated learning setups, where clients have different distributions of the same dataset, rather than each client tackling a distinct dataset. To this end, we randomly choose 10 sub-tasks from DRAKE, merge them into a single dataset, and split it into 10 unbalanced subsets by allocating a portion of each task using a Dirichlet distribution with concentration parameter $\alpha$. We then assign each subset to one of the 10 clients. Specifically, we use MagicBrush, VSR, and NLVR2 from the visual relation sub-group, 2 IRFLs, Bongard-HOI, and Bongard-OpenWorld from the multi-modal reasoning sub-group, and 2 IconQAs and Co-Instruct from the VQA sub-group in DRAKE. Following Xie et al. (2024); Morafah et al. (2024), we evaluate on $\alpha \in \{1.0, 0.5, 0.1\}$, with 20 rounds and 100 local steps. As

| Method | Self | | Others | |
|---|---|---|---|---|
| | $A_{last}$ ↑ | $A_{\text{AUC}}$ ↑ | $A_{last}$ ↑ | $A_{\text{AUC}}$ ↑ |
| SFT | 65.35±0.62 | 57.89±0.27 | 46.98±0.25 | 46.26±0.20 |
| DITTO (ICML 2021) | 60.68±0.09 | 54.69±0.13 | 47.20±0.11 | 46.46±0.16 |
| FedSim (ICML 2022) | 63.46±0.88 | 56.57±0.31 | 46.03±0.29 | 45.58±0.12 |
| FedIT (ICASSP 2024) | 65.15±0.71 | 57.82±0.43 | 47.02±0.49 | 46.12±0.35 |
| TAKFL (NeurIPS 2024) | 63.34±0.84 | 56.59±0.13 | 47.52±0.20 | 46.83±0.34 |
| FedDPA (NeurIPS 2024) | 63.66±0.80 | 56.12±0.46 | 46.97±0.44 | 46.22±0.23 |
| FedDAT (AAAI 2024) | 58.90±0.62 | 55.30±0.20 | 48.02±0.17 | 47.22±0.01 |
| PerAda (CVPR 2024) | 60.29±0.43 | 54.46±0.41 | 47.25±0.23 | 46.36±0.29 |
| FedMKT (COLING 2025) | 61.35±0.41 | 55.07±0.12 | 46.47±0.67 | 46.06±0.33 |
| FedMosaic (**Ours**) | **67.71±0.33** | **59.28±0.15** | **52.40±0.16** | **48.07±0.15** |

Table 34: **Quantitative comparison in rank heterogeneous PFL-dynamic on DRAKE.** 4 clients use LLaVA-Llama3.2-1B model, and 6 clients use LLaVA-Llama3.2-3B model, with each client randomly choosing rank $r$ from $\{64, 128, 256\}$, resulting in 3 clients with $r = 64$, 5 clients with $r = 128$, and 2 clients with $r = 256$. 'Self' denotes evaluation on a client's own data, while 'Others' denotes evaluation on data from other clients. SFT refers to supervised fine-tuning on each client's data without cross-client knowledge sharing.

shown in Tab. 35, FedMosaic shows superior performance compared to PFL baselines even on traditional non-i.i.d. PFL, demonstrating its strong generalization capability. This highlights that the performance gains of FedMosaic in the main experiments do not arise from our data heterogeneity setup, but from the general effectiveness of FedMosaic.

| Method | $\alpha = 1.0$ | | $\alpha = 0.5$ | | $\alpha = 0.1$ | |
|---|---|---|---|---|---|---|
| | $A_{last}$ ↑ | $A_{\text{AUC}}$ ↑ | $A_{last}$ ↑ | $A_{\text{AUC}}$ ↑ | $A_{last}$ ↑ | $A_{\text{AUC}}$ ↑ |
| SFT | 66.77±0.05 | 62.38±0.58 | 68.25±0.40 | 63.73±1.04 | 71.40±0.99 | 66.90±0.37 |
| DITTO | 62.00±0.12 | 58.37±0.49 | 62.34±0.12 | 59.52±0.38 | 66.80±0.95 | 63.46±0.02 |
| FedSIM | 63.73±0.03 | 59.82±0.27 | 67.44±0.33 | 62.77±0.78 | 69.90±1.97 | 65.03±1.51 |
| FedIT | 66.89±0.77 | 62.57±0.32 | 67.65±0.58 | 63.46±1.27 | 72.07±0.36 | 67.17±0.31 |
| TAKFL | 65.18±0.46 | 63.57±0.59 | 66.65±0.37 | 62.08±0.29 | 69.88±1.69 | 65.21±1.19 |
| FedDPA | 64.34±0.27 | 60.34±0.35 | 65.58±2.72 | 61.91±1.72 | 68.56±0.27 | 64.58±0.03 |
| FedDAT | 62.99±0.01 | 59.19±0.09 | 63.01±1.47 | 59.95±0.65 | 66.43±0.14 | 63.16±0.22 |
| PerADA | 61.73±0.61 | 58.20±0.68 | 62.88±1.09 | 59.77±0.76 | 66.41±0.61 | 63.27±0.28 |
| FedMKT | 64.87±0.30 | 60.67±0.12 | 65.14±0.10 | 60.81±0.39 | 69.04±0.83 | 64.78±0.33 |
| FedMosaic (**Ours**) | **69.12±0.39** | **64.72±0.02** | **70.20±0.64** | **64.68±0.20** | **73.12±1.32** | **68.39±0.89** |

Table 35: **Quantitative comparison in heterogeneous non-i.i.d. PFL on DRAKE.** We conduct PFL on Dirichlet non-i.i.d. splits with $\alpha \in \{1.0, 0.5, 0.1\}$. 4 clients use LLaVA-Llama3.2-1B model, and 6 clients use LLaVA-Llama3.2-3B model. We report the average accuracy across all clients.

### A.26 EXPERIMENTAL RESULTS ON HETEROGENEOUS PFL WITH SMALL-SCALE MODELS

In addition to our large-scale experiments with billion-parameter models in heterogeneous PFL scenarios, we also evaluate heterogeneous PFL using small-scale models. Specifically, we consider a heterogeneous PFL with model heterogeneity, where two clients use T5-small (60M) and the remaining two use T5-base (220M) (Raffel et al., 2020). For data heterogeneity, each client independently learns a different sub-task from the GLUE benchmark (Wang et al., 2019). To simulate distribution shifts, each client learns two sub-tasks sequentially: client 1: MNLI→QQP, client 2: MRPC→RTE, client 3: SST2→QNLI, client 4: COLA→MRPC. We use the AdamW optimizer with a learning rate $3 \times 10^{-4}$ and a batch size 32 for local training in all clients, updating only the LoRA parameters while freezing all pre-trained weights. Following prior T5 model training frameworks (Wang et al., 2024; Zhang et al., 2025b), we set the LoRA rank to 32, LoRA alpha to 8. We evaluate two combinations of federated learning rounds and local steps and summarize the results in Tab. 36. As shown in the table,

| | 6 rounds, 100 local steps | | | | 20 rounds, 50 local steps | | | |
| | Self | | Others | | Self | | Others | |
| Method | $A_{last}$ ↑ | $A_{\text{AUC}}$ ↑ | $A_{last}$ ↑ | $A_{\text{AUC}}$ ↑ | $A_{last}$ ↑ | $A_{\text{AUC}}$ ↑ | $A_{last}$ ↑ | $A_{\text{AUC}}$ ↑ |
|---|---|---|---|---|---|---|---|---|
| SFT | 60.94±1.35 | 50.96±0.57 | 26.19±0.80 | 24.49±0.32 | 62.88±0.36 | 51.20±0.22 | 26.94±1.34 | 25.26±0.50 |
| DITTO | 59.22±1.08 | 47.89±0.64 | 24.68±0.14 | 24.29±0.05 | 60.40±1.38 | 48.99±0.39 | 25.36±0.41 | 24.01±0.18 |
| FedSIM | 60.66±0.41 | 50.70±0.12 | 25.33±0.78 | 24.24±0.05 | 61.88±0.91 | 49.55±0.11 | 26.39±0.66 | 23.90±0.09 |
| FedIT | 60.53±0.53 | 50.76±0.04 | 25.98±0.62 | 23.93±0.36 | 62.34±0.64 | 50.85±0.10 | 26.92±1.58 | 24.40±0.52 |
| TAKFL | 60.49±0.54 | 49.88±0.15 | 25.37±0.24 | 23.60±0.35 | 61.05±1.68 | 49.89±0.57 | 26.02±0.09 | 23.66±0.07 |
| FedDPA | 61.04±0.48 | 50.19±0.13 | 25.78±0.08 | 23.84±0.24 | 61.47±0.66 | 49.09±0.32 | 25.77±0.58 | 24.19±0.33 |
| FedDAT | 51.13±0.27 | 43.25±0.08 | 24.29±0.09 | 24.41±0.01 | 53.32±0.10 | 44.65±0.38 | 23.69±0.40 | 23.39±0.06 |
| PerADA | 59.53±1.01 | 47.99±0.60 | 24.69±0.22 | 24.31±0.06 | 60.80±0.93 | 49.05±0.16 | 25.33±0.13 | 24.11±0.30 |
| FedMKT | 60.29±0.45 | 49.14±0.64 | 25.41±0.42 | 23.63±0.21 | 62.21±0.08 | 50.25±0.68 | 24.82±0.28 | 23.27±0.15 |
| FedMosaic (**Ours**) | **71.20±1.54** | **55.28±1.11** | **40.33±0.04** | **31.20±0.60** | **78.24±0.14** | **60.32±0.02** | **37.14±0.82** | **31.12±0.33** |

Table 36: **Quantitative comparison in heterogeneous PFL-dynamic with small-scale models.** We evaluate heterogeneous PFL using small-scale T5 models, where T5-small serves as the smaller model and T5-base as the larger model on the GLUE benchmark. Two clients use T5-small and two clients use T5-base. We evaluate heterogeneous PFL under two settings: 6 rounds with 100 local steps, and 20 rounds with 50 local steps. 'Self' denotes evaluation on a client's own data, while 'Others' denotes evaluation on data from other clients.

FedMosaic significantly outperforms baselines in both PFL settings. These small-scale experiments demonstrate that Co-LoRA and RELA can be applied effectively regardless of model sizes.

A.27 ADDITIONAL DETAILS OF DRAKE

We curate DRAKE using open-sourced datasets from the Internet. We refer to diverse multi-modal benchmarks, such as DEMON (Li et al., 2024c), SEED-Bench-2 (Li et al., 2024a), Co-Instruct (Wu et al., 2024b), and HFLB (Chen et al., 2024a).

DRAKE consists of 40 distinct tasks, where for each task, we subsample 10,000 training and 1,000 test samples, if the dataset is large; otherwise, we split the data into training and test sets with an 8:2 ratio, totaling 375k images and 274k questions. We illustrate the overall structure of DRAKE as a tree diagram in Fig. 24. As shown in the figure, DRAKE consists of three sub-groups, *i.e.*, visual relation, multi-modal reasoning, and VQA, each comprising multiple fine-grained tasks. We also provide detailed per-client task configuration of DRAKE in Tab. 37.

**Visual Relation Tasks**   Visual relation tasks focus on capturing relationships among visual objects, consist of 12 distinct tasks. We categorize them into three splits: *fashion-relation*, *dual-image relation*, and *spatial/temporal*. The fashion-relation split includes the cloth-counting and three-/four-image variants of Fashion200K (Han et al., 2017) as well as the query–reference caption split of FashionIQ (Wu et al., 2021). The dual-image relation split is comprised of on NLVR2 (Suhr et al., 2019), CIRR (Liu et al., 2021), the two- and four-image true/false subsets of VISION (Bai et al., 2023), and MagicBrush (Zhang et al., 2023a). The spatial/temporal split combines single-frame spatial reasoning in VSR (Liu et al., 2023a) with temporal understanding tasks from SEED-Bench-2 (Li et al., 2024a).

**Multi-modal Reasoning Tasks**   Multi-modal reasoning tasks require the integration of visual cues with common-sense knowledge, comprising 12 distinct tasks. We include four tasks from IRFL (Yosef et al., 2023) that examine the figurative interpretation of images paired with non-literal language. We also include five additional tasks derived from the positive–negative image groups in Bongard-HOI (Jiang et al., 2022) and Bongard-OpenWorld (Wu et al., 2024d). These tasks require fine-grained discrimination between highly similar but contrasting visual sets. We incorporate three reasoning challenges from COMICS (Iyyer et al., 2017), MIT-States (Isola et al., 2015), and VizWiz (Gurari et al., 2018), all sourced from DEMON (Li et al., 2024c).

**VQA Tasks**   The *VQA* sub-group includes 16 diverse VQA tasks that differ significantly from LLaVA's pre-training datasets. We include novel question types, such as image-quality queries in Co-Instruct (Wu et al., 2024b), textbook-style diagram interpretation in TQA (Kembhavi et al., 2017), DVQA (Kafle et al., 2018), and knowledge-grounded QA tasks from SEED-Bench-2 (Li et al., 2024a). We also incorporate unfamiliar image domains, including IconQA (Lu et al., 2021) and WCVQA (Winata et al., 2024). We partition Co-Instruct by question type (*e.g.*, Yes/No, How/what

questions), IconQA by difficulty and question category (*e.g.*, Multi-choice or Short answering, kindergarten-level or grade 1-level questions), while we divide WCVQA using their original split.

| Client 1 | Client 2 |
|---|---|
| • Fashion200K (Han et al., 2017): ClothCombination
• FashionIQ (Wu et al., 2021)
• VISION (Bai et al., 2023): 2_Img
• MagicBrush (Zhang et al., 2023a) | • Co-Instruct (Wu et al., 2024b): 2_ImgCompare
• SEED–Bench–2 (Li et al., 2024a): KGQA
• IconQA (Lu et al., 2021): ShortAnswerEasy
• WCVQA (Winata et al., 2024): DishName |
| **Client 3** | **Client 4** |
| • Co-Instruct (Wu et al., 2024b): HowWhat
• DVQA (Kafle et al., 2018)
• IconQA (Lu et al., 2021): ShortAnswerHard
• WCVQA (Winata et al., 2024): ContextDishName | • Co-Instruct (Wu et al., 2024b): 3_ImgCompare
• SEED–Bench–2 (Li et al., 2024a): InstanceQA
• IconQA (Lu et al., 2021): MultiChoiceEasy
• WCVQA (Winata et al., 2024): AdvContextDishName |
| **Client 5** | **Client 6** |
| • Fashion200K (Han et al., 2017): ColorConsistency
• NLVR2 (Suhr et al., 2019): Subset1
• VISION (Bai et al., 2023): 4_Img
• VSR (Liu et al., 2023a) | • Fashion200K (Han et al., 2017): StyleConsistency
• NLVR2 (Suhr et al., 2019): Subset2
• CIRR (Liu et al., 2021)
• SEED–Bench–2 (Li et al., 2024a): TemporalQA |
| **Client 7** | **Client 8** |
| • IRFL (Yosef et al., 2023): MetaphorSimileMatching
• Bongard–HOI (Jiang et al., 2022): ActionDetection
• VizWiz (Gurari et al., 2018)
• MIT–States (Isola et al., 2015) | • IRFL (Yosef et al., 2023): FigurativeVerification
• IRFL (Yosef et al., 2023): IdiomMatching
• Bongard–OpenWorld (Wu et al., 2024d): ConceptDetection
• COMICS (Iyyer et al., 2017): Dialogue |
| **Client 9** | **Client 10** |
| • IRFL (Yosef et al., 2023): PhraseSelection
• Bongard–OpenWorld (Wu et al., 2024d): ConceptQuery
• Bongard–HOI (Jiang et al., 2022): ConceptQuery
• Bongard–HOI (Jiang et al., 2022): ActionIncoherence | • Co-Instruct (Wu et al., 2024b): YesNO
• TQA (Kembhavi et al., 2017)
• IconQA (Lu et al., 2021): MultiChoiceHard
• WCVQA (Winata et al., 2024): Location |

Table 37: **Per-client task configuration of DRAKE.** DRAKE consists of 10 clients, each with 4 distinct tasks. Client 1, 5, 6 tackle visual relation tasks, Client 7, 8, 9 handle multi-modal reasoning tasks, and Client 2, 3, 4, 10 focus on VQA tasks, as illustrated in Fig. 24.

**Unseen Tasks**  The unseen tasks include novel task types (DreamSim (Fu et al., 2023b), Image-CoDe (Krojer et al., 2022)) and novel visual domains with familiar task format (RecipeQA (Yagcioglu et al., 2018) subsets proposed in DEMON (Li et al., 2024c), and HQ-Edit (Hui et al., 2025)). In addition, we split the Contrast-Caption subset of Mantis (Jiang et al., 2024; Yu et al., 2022) into two tasks: one with images overlapping LLaVA's pre-training distribution and another with previously unseen images.

A.28 COMPARISON WITH OTHER FL BENCHMARKS.

A number of federated learning (FL) benchmarks exist, but most lack critical aspects required for FL of multi-modal foundation models, which DRAKE addresses, with the main differences summarized in Tab. 1.

Benchmarks like NonIID-50 (Yoon et al., 2021), LEAF-FCL (Qi et al., 2023), and MNIST-Shuffle (Wuerkaixi et al., 2024) simulate distribution shifts under FL. However, they are single-task and single-modal (*i.e.*, image classification) only, thus cannot reflect real-world task diversity. HC-FMTL (Lu et al., 2024) spans multiple vision tasks, such as depth estimation and semantic segmentation, but still remains unimodal.

On the language side, several text-only FL benchmarks have recently been proposed to target LLM federated learning scenarios. Fed-SNI (Collins et al., 2023) and Fed-FLAN (Long et al., 2024) cover diverse NLP tasks, while Fed-Aya (Ye et al., 2024) focuses on multilingual instruction following. FEDLEGAL (Zhang et al., 2023b) curates a FL benchmark for the privacy-sensitive legal domain where federated learning is necessary. Despite their breadth, they are also limited to a single modality and assume static data distribution, while data distribution often shifts over time in real-world.

For multi-modal PFL, we find only one prior benchmark: HFLB (Chen et al., 2024a). Although HFLB includes 7 different datasets, the multi-modal task diversity is limited, and they are mostly

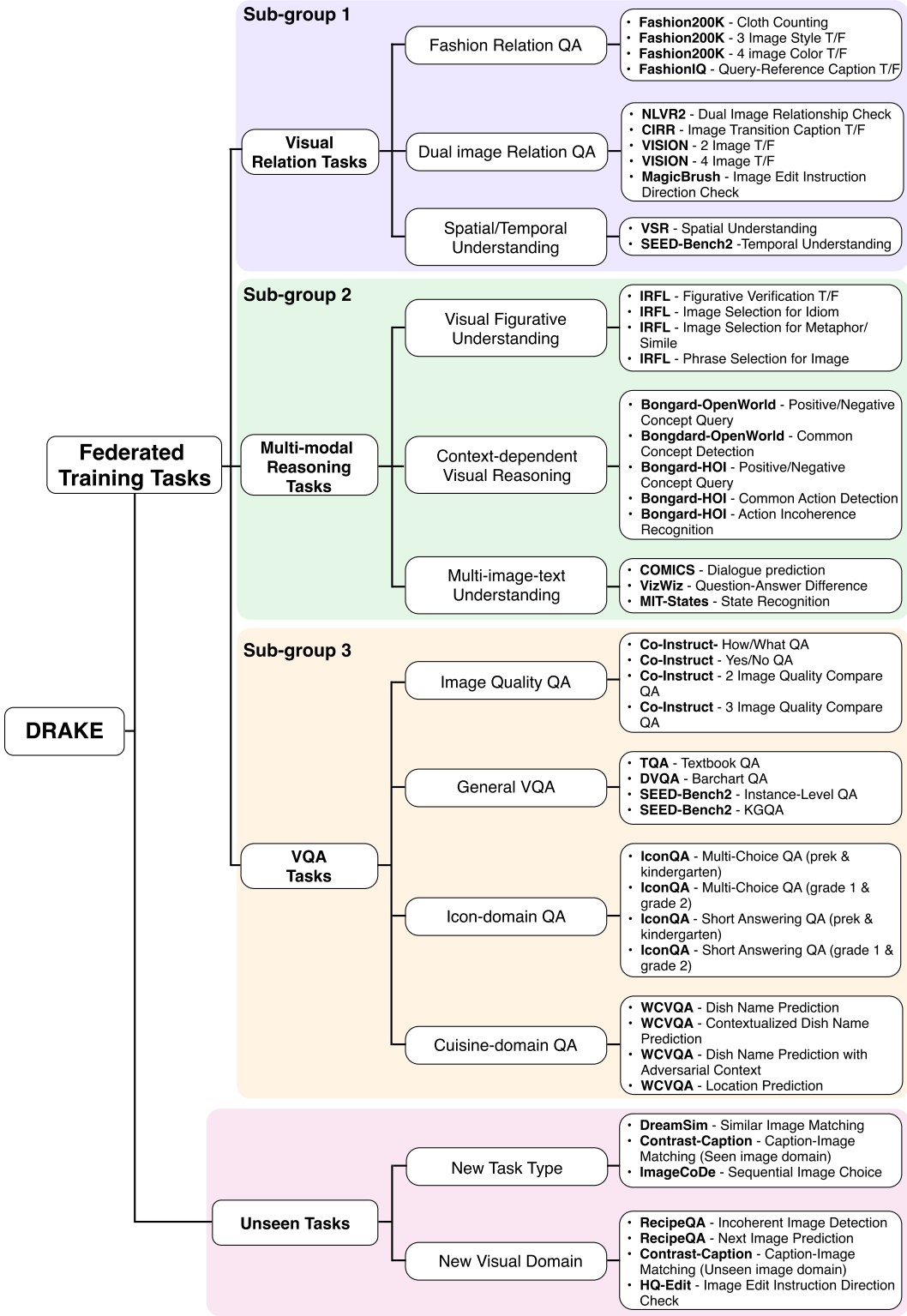

Figure 24: **Detailed configuration of the proposed DRAKE benchmark.**

single-image settings. Other multi-modal FL benchmarks, *e.g.*, FedMultimodal (Feng et al., 2023), FHBench (Wang et al., 2025), and FedMLLM (Xu et al., 2024), are primarily comprised of only

a classification task on different modalities or split one or two VQA datasets, which is insufficient for evaluation personalization under diverse heterogeneous tasks. Importantly, existing benchmarks generally lack an unseen task split to test the generalization ability of the model.

In contrast, DRAKE offers 40 different multi-modal tasks (including multi-image tasks), which is suitable for personalization evaluation and temporal distribution shift. Moreover, DRAKE additionally provides 7 unseen tasks for out-of-distribution generalization evaluation.

## A.29    Justification for Using a Smaller Model as the Pivot

During the Co-LoRA alignment, we freeze the smaller model as a pivot because, over time, new models that become available are typically larger and stronger, rather than smaller and weaker ones. In this scenario, using the larger model as the pivot would cause the pivot to change whenever a new model is introduced, requiring re-alignment of all previously aligned models. For example, when aligning LLaVA-1B and LLaVA-3B, if we freeze the larger model (*i.e.*, LLaVA-3B) as the pivot, the introduction of LLaVA-8B would make it the new pivot. Consequently, both LLaVA-1B and LLaVA-3B would need to be updated during alignment with LLaVA-8B, which would break the previously established alignment between LLaVA-1B and LLaVA-3B.

Moreover, we note that this strategy does not affect performance. As shown in Tab. 38, employing the larger model as the pivot yields comparable performance to our strategy. In summary, we adopt this approach not because it offers better performance, but because it provides a stable pivot for alignment when larger models are newly deployed.

| Co-LoRA alignment scenario | Pivot model | Self | | Others | |
|---|---|---|---|---|---|
| | | $A_{last}$ ↑ | $A_{\text{AUC}}$ ↑ | $A_{last}$ ↑ | $A_{\text{AUC}}$ ↑ |
| LLaVA-Llama3.2-1B / LLaVA-Llama3.2-3B | small (1B) | **67.86**±**0.51** | **59.83**±**0.15** | 51.16±0.04 | **49.36**±**0.08** |
| | large (3B) | 67.71±0.27 | 59.54±0.05 | **51.46**±**0.37** | 49.29±0.19 |
| LLaVA-Qwen2.5-1.5B / LLaVA-Llama3.2-3B | small (1.5B) | 70.62±0.29 | 63.33±0.26 | **52.56**±**0.08** | 50.76±0.01 |
| | large (3B) | **70.83**±**0.35** | **63.40**±**0.25** | 52.55±0.21 | **50.82**±**0.01** |

Table 38: **Effect of pivot model selection for Co-LoRA alignment in heterogeneous PFL-dynamic on DRAKE.** During the alignment process, we freeze the pivot model and update the other model to align with it. The accuracy remains consistent regardless of the chosen pivot model. 'Self' denotes evaluation on a client's own data, while 'Others' denotes evaluation on data from other clients.

## A.30    Privacy Guarantee of Sanitized Gradient $\tilde{g}$

FedMosaic requires transmitting per-client gradient vectors to the server for RELA, which may expose clients to privacy attacks such as gradient inversion (Mo et al., 2021). To mitigate these concerns, we employ (i) using only the last-layer gradient, (ii) exponential moving average (EMA) updates, and (iii) gradient sampling with noise addition for sanitization. To assess the privacy protection of our sanitized gradient $\tilde{g}$, we evaluate its resistance under a strong gradient inversion attack, LAMP (Balunovic et al., 2022). LAMP is designed to reconstruct text inputs by searching for an text sequence that produces the same gradient as the target gradient, alternating between continuous and discrete optimization steps. In the continuous optimization step, LAMP updates the text embedding so that its gradient matches the target gradient. In the discrete optimization step, it maps the continuous optimized text embedding back to tokens, generates multiple candidates by applying operations such as token swapping or moving, and then selects the best candidate with the lowest perplexity to update the text embedding. By iteratively alternating between these two steps, LAMP gradually recovers an input that is both close to the target gradient and linguistically natural.

Under this attack, we compare our sanitized gradient against several recently proposed defense approaches for shared gradients in the (personalized) federated learning domain, such as Clipping + Noise (Bietti et al., 2022), SVD truncation (Luo et al., 2025), Orthogonal projection (Zhang et al., 2025a), and Fisher Information (Jhunjhunwala et al., 2024). Clipping + Noise clips the gradient norms and add Gaussian noise to guarantees a differential-privacy. SVD truncation retains only the

| Defense Method | Rouge-L $\downarrow$ |
|---|---|
| Full layer gradient | 0.2952 |
| Clipping + Noise (Bietti et al., 2022) | 0.2720 |
| SVD Truncation (Luo et al., 2025) | 0.2688 |
| Fisher Information (Jhunjhunwala et al., 2024) | 0.2128 |
| Orthogonal Projection (Zhang et al., 2025a) | 0.1614 |
| Last layer gradient | 0.1442 |
| Last layer gradient + EMA aggregated | 0.0844 |
| Last layer gradient + EMA aggregated + Noise | 0.0699 |
| Last layer gradient + EMA aggregated + Noise + Compression (sanitized gradient $\tilde{g}_i$) | **0.0653** |

Table 39: **Comparison of privacy-defense methods under gradient inversion attack.** We report the recovered text Rouge-L scores, where lower Rouge-L indicates better protection against privacy leakage from gradient inversion attacks. We set the batch size to 4, noise scale $\mu$ to $1 \times 10^{-4}$, and compression ratio to 40%.

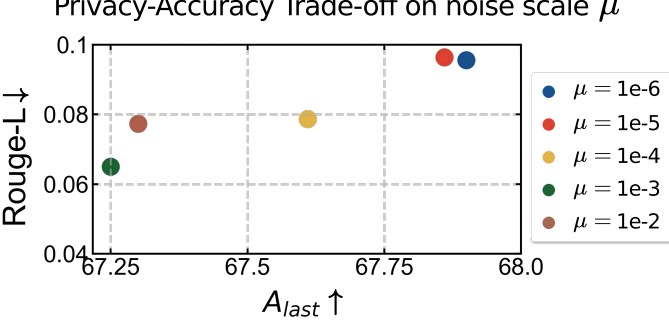

Figure 25: **Privacy-Accuracy trade-off on noise scale $\mu$.** Lower Rouge-L score indicates better protection against gradient inversion attack, and higher $A_{last}$ indicates better PFL performance. Increasing noise scale leads to stronger privacy protection but incurs accuracy drop.

principal components of the gradient and Orthogonal projection projects normalized random vector on the gradient subspace, rather than full gradients. Jhunjhunwala et al. (2024) use Fisher information to weight local modules while mitigating privacy concerns, approximating it with the diagonal of the Fisher Information Matrix, *i.e.*, the squared gradient, which discards the sign information.

We summarize the results of LAMP on the Llama3.2-1B model using GLUE-COLA dataset in Tab. 39, reporting the Rouge-L score of recovered input texts. As shown in the table, our sanitized gradient effectively protects private information from the attack compared to baselines. Specifically, using only the last-layer gradient significantly reduces reconstruction success compared to using full-layer gradients as well as other defense baselines, demonstrating that last-layer gradients not only reduce communication and computation cost but also mitigate privacy risks. Moreover, EMA-based gradient accumulation provides additional protection by mixing gradients along the temporal axis. Finally, noise addition and compression for sanitization further obscure input information, making the sanitized gradient $\tilde{g}$ substantially more resistant to privacy attacks.

Finally, we analyze the privacy-accuracy trade-off on noise scale $\mu$. As shown in Fig. 25, adding more Gaussian noise hinders text recovery, but increasing the noise level also degrades performance by reducing the representativeness and informativeness of the gradients. Based on this privacy-accuracy trade-off, we adopt a moderate noise scale ($1 \times 10^{-4}$) in RELA.

### A.31 CLIENT-WISE TASK RELEVANCE VISUALIZATION

To analyze task relevance within DRAKE, we compute a task-relevance matrix using the sanitized gradients $\hat{g}$ employed in RELA. For ease of visualization, tasks belonging to the same group are placed adjacently. Specifically, each task corresponds to the clients listed in Tab. 37 as follows: tasks

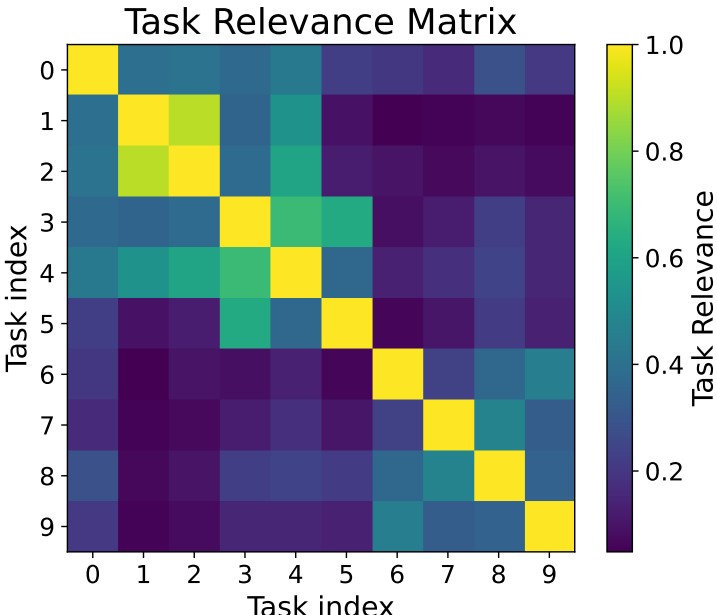

Figure 26: **Task Relevance Matrix of DRAKE.** The task relevance is measured by sanitized gradient $\tilde{g}_i$ from DRAKE-Static setup. Each task is the per-client task summarized in Tab. 37, where 0-2 tasks tackle visual relation tasks, 3-5 tasks handle multi-modal reasoning tasks, and the rest focus on VQA tasks. Tasks within the same task-category exhibit notably high mutual relevance.

0–2 (clients 1, 5, 6) correspond to visual relation, tasks 3–5 (clients 7, 8, 9) to multi-modal reasoning, and tasks 6–9 (clients 4, 10, 3, 2) to VQA.

As shown in Fig. 26, tasks within the same group exhibit noticeably higher mutual relevance than those across different groups. Moreover, tasks 0–5 show relatively high cross-group relevance, likely because they all involve multi-image inputs, whereas tasks 6–9 primarily use single-image inputs, and this shared input structure contributes to their similarity.

### A.32    Experimental Results on Heterogeneous PFL with Asynchronous Distribution Shift

We simulate data distribution shift in PFL (*i.e.*, dynamic setup) by sequentially introducing new task for the fixed number of rounds. Thus, all clients experience the distribution shift simultaneously (i.e., at the same round) in the experiments. For instance, in both HFLB-Dynamic and DRAKE-Dynamic experiments, each client learns 4 tasks sequentially over 20 rounds, with a distribution shift occurring every 5 rounds. Although FedMosaic shows its effectiveness under syncrhonized distribution-shift, it can naturally be extended to an *asynchronized distribution-shift* setup, where distribution shifts occur at different times across clients. To demonstrate that FedMosaic works robustly regardless of the timing of distribution shifts, we evaluate FedMosaic under the asynchronized distribution-shift setup. Specifically, instead of assigning every client a fixed 5 rounds per task, each client randomly adopts one of three task-round patterns: (5,5,5,5), (3,7,3,7), or (7,3,7,3), where each number denotes the number of rounds allocated to each task in order. As shown in Tab. 40, FedMosaic consistently outperforms other baselines even under this asynchronous distribution-shift scenario, demonstrating its stability to heterogeneity in the timing of distribution shifts.

### A.33    Limitations and Future Work

While we evaluate FedMosaic across diverse multi-modal FL benchmarks (*e.g.*, HFLB and our proposed DRAKE) and text-only NLP benchmarks (*e.g.*, Fed-Aya, Fed-Scope, and Fed-LLM-Large), it has not yet been tested on other modalities such as speech and time series. Extending evaluations

| Method | Self | | Others | |
|---|---|---|---|---|
| | $A_{last}$ ↑ | $A_{\text{AUC}}$ ↑ | $A_{last}$ ↑ | $A_{\text{AUC}}$ ↑ |
| SFT | 66.23±0.26 | 58.35±0.28 | 46.94±0.34 | 46.36±0.28 |
| DITTO (ICML 2021) | 60.11±0.80 | 54.60±0.15 | 47.27±0.28 | 46.43±0.10 |
| FedSim (ICML 2022) | 64.71±0.49 | 57.62±0.25 | 46.44±0.13 | 45.91±0.08 |
| FedIT (ICASSP 2024) | 66.77±0.25 | 58.71±0.24 | 46.92±0.19 | 46.32±0.15 |
| TAKFL (NeurIPS 2024) | 65.47±0.95 | 58.07±0.52 | 47.14±0.23 | 46.46±0.02 |
| FedDPA (NeurIPS 2024) | 62.40±0.82 | 56.26±0.21 | 47.33±0.24 | 46.53±0.12 |
| FedDAT (AAAI 2024) | 59.06±0.35 | 54.79±0.21 | 48.90±0.09 | 47.83±0.06 |
| PerAda (CVPR 2024) | 59.82±0.39 | 54.49±0.16 | 47.18±0.17 | 46.43±0.06 |
| FedMKT (COLING 2025) | 61.76±0.06 | 55.79±0.15 | 46.76±0.18 | 46.30±0.07 |
| FedMosaic (**Ours**) | **68.30±0.45** | **60.23±0.07** | **51.96±0.26** | **49.63±0.10** |

Table 40: **Quantitative comparison in heterogeneous PFL-dynamic on DRAKE with asynchronous distribution shift among clients.** 4 clients use LLaVA-Llama3.2-1B model, and 6 clients use LLaVA-Llama3.2-3B model, with each client randomly choosing one of three task-round patterns: (5,5,5,5), (3,7,3,7), or (7,3,7,3), where each number denotes the number of rounds allocated to each task in order. 'Self' denotes evaluation on a client's own data, while 'Others' denotes evaluation on data from other clients. SFT refers to supervised fine-tuning on each client's data without cross-client knowledge sharing.

to these domains would better capture real-world FL scenarios and represent an important direction for future work.

Moreover, although LoRA is most commonly used in transformer-based models, it is not restricted to them. Specifically, prior studies (Yeh et al., 2023; Ding et al., 2024) have demonstrated its applicability to convolutional layers, highlighting its architecture-agnostic nature. Therefore, exploring the use of Co-LoRA beyond transformers to enable knowledge transfer across broader model families represents an important direction for future work.

### A.34 IMPACT STATEMENTS

This work focuses on federated learning, which enables model training in a distributed manner, eliminating the need to share raw data and thereby mitigating data privacy risks. However, since we assume that the model is trained under distribution shifts, there is a potential for the unintended amplification of issues such as model bias and ethical misalignment. We are committed to taking all necessary measures to address these risks, although tackling these concerns is not the primary focus of this work.

### A.35 DETAILED ALGORITHM OF FEDMOSAIC

Algorithm 1 provides a pseudocode for the proposed FedMosaic framework. We further provide detailed algorithms for initializing adapters (Algorithm 2) and aligning adapters across heterogeneous clients (Algorithms 3 and 4).

**Algorithm 1:** `FedMosaic`

```
# Input
#   Number of clients N, Number of client model types K,
#     Set of frozen pre-trained models 𝒲 = {W₁,...,W_K},
#     Mapping of client to model V : {1,...,N} → 𝒲, Number of rounds
 R,
#     Batch size B, Learning rate η, Gradient frequency f, EMA
 ratio a, Temperature τ, Gaussian noise std μ, Gradient
 subsample ratio N_s
#     Small-scale pre-trained model 𝒲_S, Regularization
 coefficient λ
#     Training data streams of T tasks for N clients
 {𝒟₁,𝒟₂,...,𝒟_N},
#       where 𝒟_i = {𝒟_i¹,𝒟_i²,...,𝒟_i^T}, Number of Co-LoRA blocks N_B,
#     Public dataset 𝒟_p, Conventional-LoRA, Co-LoRA

E = InitializeAdaptersForModels(𝒲, N_B, K) # (Alg. 2)
E = AlignAdapter(𝒲, E, 𝒟_p, K, λ) # (Alg. 3, Alg. 4)

# Initialization for each client
for i in range(N) do
  M[i] = [] # Initialize episodic memory
  # Initialize local and global adapters with the aligned
   adapter E
  L[i] = E[i]
  G[i] = E[i]
  # Initialize gradient vector with zeros
  g[i] = torch.zeros(d_[len(V(i))])
  # Initialize gating parameters for each layer in L[i]
  beta[i] = [0.0 for _ in range(len(L[i]))]

# Set Random subsample indices for last layer gradient
subsample_index = random_choice(last_layer_grad size, N_s)

for task t in range(T) do
  for round r in range(R) do
    # Client-side
    for client index i in range(N) do
      W_i = V(i)
      g_t = []
      for j, (x_j, y_j) in 𝒟_i^t do
        M[i].append((x_j, y_j))
        x, y = random_choice(M[i], B)
        # Compute loss with mixed local-global adapter
        loss = CrossEntropy(Forward(W_i, L[i], G[i], beta[i],
         x), y)
        L[i] -= η * Grad(L[i], loss)
        if i % f == 0 then
          loss = CrossEntropy(Forward(𝒲_S, x), y)
          g_last = LastLayerGrad(𝒲_S, loss)
          # Subsample gradient and add gaussian noise
          eps ∼ N(0,I) # I ∈ ℝ^(last_layer_grad size)
          g_sanitized = g_last[subsample_index] + μ *
           eps[subsample_index]
          g_t.append(g_sanitized)
      g[i] = (1 - a) * g[i] + a * Average(g_t)

    # Server-side
    # Aggregate global adapters using SIMA
    G = SIMA(L, g)
```

**Algorithm 2:** `InitializeAdaptersForModels`$(\mathcal{W}, N_B, K)$

```
// Initialize the list of adapter lists for each model
E = []
```
**for** `k in range(1, K)` **do**
    `// Initialize adapter list for model` $\mathcal{W}_k$
    `E_k = []`
    `// Block size per model  (Co-LoRA assigned)`
    `B_k = len(`$\mathcal{W}$`[k]) //` $N_B$
    `// Remaining layers not evenly divisible`
    `B_r = len(`$\mathcal{W}$`[k]) %` $N_B$
    **for** `l in range(1, len(`$\mathcal{W}$`[k]))` **do**
        **if** $\exists$ `n in {1..`$N_B$`} s.t.` $l == n * B\_k + B\_r$ **then**
            `// Append  Co-LoRA`
            `E_k.append( Co-LoRA)`
        **else**
            `// Append conventional LoRA`
            `E_k.append(Conventional-LoRA)`
    `// Add adapter list to full collection`
    `E.append(E_k)`
```
return E
```

---

**Algorithm 3:** `AlignAdapter(W, E, ` $\mathcal{D}_p$ `, ` $K$ `, ` $\lambda$ `)` `- Part 1`

---

```
# Define hook functions to capture outputs during a forward
 pass
Function hook_fn(module, input, output):
  └ self.lora_outputs.append(output)

### Combine Co-LoRA with the pre-trained model
E_indices = []
for i in range(K) do
  │ E_index = []
  │ for j in range(len(W[i])) do
  │   │ if E[i][j] == Co-LoRA then
  │   │   │ # Attach Co-LoRA to pre-trained model layers
  │   │   │ W[i][j] = Attach(E[i][j], W[i][j])
  │   │   └ E_index.append(j)
  └ E_indices.append(E_index)

### Align Co-LoRA A matrices
W_p = W[0] # Set pivot model
W_p.freeze() # Freeze pivot model
for i in range(1, K) do
  │ W_i = W[i]
  │ ## Step 1.  Attach hooks to LoRA A and perform a forward pass
  │ # Register hooks to capture the outputs of LoRA A matrix at
  │  Co-LoRA attached layers
  │ for j in E_indices[0] do
  │   └ W_p[j].A.register_forward_hook(hook_fn)
  │ for j in E_indices[i] do
  │   └ W_i[j].A.register_forward_hook(hook_fn)
  │
  │ ## Step 2.  Alignment of A using L2 loss
  │ total_loss = 0
  │ for (x, y) in D_p do
  │   │ W_p.forward_with_lora_scaled(x, scale=0)
  │   │ W_i.forward_with_lora_scaled(x, scale=0)
  │   │ total_loss = 0
  │   │ # Compute L2 loss for each layer using outputs captured by
  │   │  hooks
  │   │ for j in range(len(lora_outputs[i])) do
  │   │   │ # Compute L2 between LoRA A outputs of W_i and W_p at
  │   │   │  layer j
  │   │   │ loss_l2 = L2(W_p.lora_outputs[j], W_i.lora_outputs[j])
  │   │   │ # Compute Regularization loss to prevent deviation from
  │   │   │  orthogonality (Eq. 21)
  │   │   │ loss_reg = Reg(W_i[j].A)
  │   │   └ total_loss += loss_l2 + λ * loss_reg
  │   └ W_i -= η * Grad(W_i, total_loss)
  │
  │ ## Step 3.  Post-process A to enforce Orthogonality
  │ for j in E_indices[i] do
  │   │ # Use Singular Value Decomposition to get the closest
  │   │  orthogonal matrix
  │   │ U, S, Vt = SVD(W_i[j].A)
  │   └ W_i[j].A = U*Vt
```

---

**Algorithm 4:** `AlignAdapter(`$\mathcal{W}$`, `$E$`, `$\mathcal{D}_p$`, `$K$`, `$\lambda$`)` `- Part 2`

```
### Align Co-LoRA B matrices
for i in range(1, K) do
    W_i = W[i]
    ## Step 1.  Attach hooks to LoRA B and perform a forward pass
    # Register hooks to capture the outputs of B in Co-LoRA
     attached layers
    for j in E_indices[0] do
        W_p[j].B.register_forward_hook(hook_fn)
    for j in E_indices[i] do
        W_i[j].B.register_forward_hook(hook_fn)

    # Forward pass to collect outputs of the LoRA B matrices for
     CCA
    for (x, y) in D_p do
        W_p.forward_with_lora_scaled(x, scale=0)
        W_i.forward_with_lora_scaled(x, scale=0)

    ## Step 2.  Apply CCA to align B matrices
    for j in range(len(W_i)) do
        # Extract B matrix outputs from hooks
        X_p = W_p.lora_outputs[j]
        X_i = W_i.lora_outputs[j]

        # Compute CCA projection vectors
        Π_p, Π_i = CCA(X_p, X_i)

        # Initialize B using the CCA transformation
        W_i[j].B = ((Π_i)⁻¹)ᵀ · (Π_p)ᵀ · W_p[j].B

E_align = []
for i in range(K) do
    for j in range(len(W[i])) do
        if E[i][j] == Co-LoRA then
            E[i][j] = W[i][j]
    E_align.append(E[i])

return E_align
```