# OpenReview forum: "Co-LoRA: Collaborative Model Personalization on Heterogeneous Multi-Modal Clients"
_ICLR.cc/2026/Conference — ICLR 2026 Poster_

### Official Review · Reviewer_oYUk · 2025-10-15

**Soundness:** 4
**Presentation:** 4
**Contribution:** 4
**Rating:** 10
**Confidence:** 4

**Summary:**

The paper tackles personalized federated learning (PFL) under realistic heterogeneity: data heterogeneity where each client has distinct multi-modal tasks with temporal shifts, and model heterogeneity where clients use different model families and sizes. It introduces FedMosaic, which combines relevance-guided aggregation and PQ-LoRA to enable selective knowledge sharing and cross-architecture parameter sharing.
The authors also release DRAKE, a multi-modal PFL benchmark with 40 tasks spanning VQA, visual reasoning, visual relations, including multi-image inputs and unseen task evaluation under distribution shifts.
Empirically, FedMosaic outperforms strong baselines across heterogeneous/static/dynamic and cross-family settings on Self (personalization) and Others (generalization), and improves fast adaptation on unseen tasks; ablations show both RELA and PQ-LoRA contribute meaningfully.

**Strengths:**

- Well-posed problem + realistic setup. The paper motivates that most PFL work oversimplifies heterogeneity; here, clients differ in both data and model (families and depths/sizes), which is closer to practice (agentic AI, device constraints).

- Clear algorithmic design. The proposed method comes with clear motivation and design solution accordinly. For instance, RELA computes client-wise gradients on a small frozen model. It applies EMA decay to track shifting client knowledge and also adds Gaussian noise + random subsampling for privacy and bandwidth.

- Benchmark contribution. DRAKE covers multi-modal, multi-image tasks, temporal shifts, and unseen evaluation; the table contrasts prior FL benchmarks along these axes.

- Strong and granular evidence. The experiments are comprehensive with different settings like heterogeneous (same-family) PFL and cross-family heterogeneity, also analyize the performance from per-client view and fast adaptation possibility. Detailed ablations about adding PQ-LoRA improves Others, adding RELA further lifts Self/Others, are also provided.

**Weaknesses:**

While RELA applies EMA, noise, and subsampling to the last-layer gradients from privacy perspective, an explicit comparison to baselines with stronger privacy guarantees would strengthen the claim.

**Questions:**

- I'm curious about the task relavance between the tasks in your DRAKE benchmark, like showing the relavance matrix.

---

> ### Author Response · Authors · 2025-11-19
> **Response to Reviewer oYUk**
>
> Dear Reviewer oYUk,
>
> We are grateful for the reviewer's strongly positive assessment. We also appreciate your suggestions regarding the privacy guarantee of RELA and the detailed analysis of task relevance in DRAKE benchmark. Below, we address these suggestions one by one.
>
> > While RELA applies EMA, noise, and subsampling to the last-layer gradients from privacy perspective, an explicit comparison to baselines with stronger privacy guarantees would strengthen the claim.
>
> $\to$ Thank you for the great suggestion! We have added a comparison with baselines focusing on privacy guarantees in Table G below (details in Sec. A.30 of the revision). To evaluate the privacy protection offered by our gradient sanitization in RELA, we conduct gradient inversion attacks [1] on Llama-1B using the GLUE-COLA dataset. As shown in Table G below, **RELA’s components, including EMA, noise addition, and gradient subsampling, significantly prevent the recovery of input text**. Specifically, using only the last-layer gradient significantly reduces reconstruction success compared to using full-layer gradients, demonstrating that last-layer gradients not only reduce communication and computation cost but also mitigate privacy risks. We have included these results in Sec. A.30, Tab. 39 of the revision.
>
>
> **Table G. Comparison of privacy-defense methods under gradient inversion attack**
> |Defense Method|Recovered text Rouge-L $\downarrow$|
> |-|-|
> |Full layer gradient|0.2952|
> |Clipping+Noise|0.2720|
> |SVD Truncation|0.2688|
> |Fisher Information|0.2128|
> |Orthogonal Projection|0.1614|
> |||
> |Last layer gradient|0.1442|
> |+EMA aggregation|0.0844|
> |+Noise|0.0699|
> |+Subsampling|**0.0653**|
>
> > I'm curious about the task relevance between the tasks in your DRAKE benchmark, like showing the relevance matrix.
>
> $\to$ Thank you for the insightful suggestion. We have added a task relevance matrix for the DRAKE benchmark in Sec. A.31 of the revision. The relevance between tasks is computed using the sanitized gradients $\tilde{g}$ employed in RELA. In Fig. 26, tasks 0-2 correspond to visual relation tasks, tasks 3-5 correspond to multi-modal reasoning tasks, and tasks 6-9 correspond to VQA. The matrix clearly shows that tasks within the same group exhibit noticeably higher mutual relevance than those across different groups. Moreover, tasks 0–5 also exhibit relatively high cross-group relevance, likely because these tasks primarily involve multiple-image inputs, whereas tasks 6–9 mostly involve single-image inputs. This shared input structure may contribute to the observed relevance.
>
>
> [1] Balunovic et al., LAMP: Extracting Text from Gradients with Language Model Priors, NeurIPS 2022

---

### Official Review · Reviewer_UdQR · 2025-10-17

**Soundness:** 3
**Presentation:** 3
**Contribution:** 2
**Rating:** 4
**Confidence:** 4

**Summary:**

This work addresses both data and model heterogeneity in Personalized Federated Learning (PFL). The authors propose FedMosaic, a framework that jointly mitigates these challenges through two core components: RELA and PQ-LoRA. RELA (Relevance-guided Aggregation) constructs client-specific global models by weighting updates based on task relatedness, enabling effective knowledge sharing among similar clients while reducing interference across unrelated tasks. PQ-LoRA introduces dimension-invariant low-rank adapters whose parameters depend only on rank $r$, allowing efficient and architecture-agnostic knowledge sharing across heterogeneous models. To more accurately capture real-world task heterogeneity and distribution shifts, they further introduce DRAKE, a comprehensive multi-modal federated learning benchmark.

**Strengths:**

1. Experimental results demonstrate consistent improvements over strong baselines, indicating the effectiveness of the approach.

2. The appendix further provides a thorough and extensive suite of experiments, supporting the validity and robustness of the reported findings.

3. The paper is generally well-written and clearly organized, making the technical ideas easy to follow.

**Weaknesses:**

1.	The comparison against prior works using non-IID splits of a single dataset may not be entirely equitable. The contextual settings differ significantly (maybe the earlier studies targeted models specialized for single-domain or unimodal tasks, rather than fine-tuning billion-parameter foundation models). Moreover, the motivation for exploring multi-modal tasks in PFL requires further clarification. What are the practical or deployment-oriented use cases where clients naturally possess distinct modalities? At present, the setup appears somewhat hypothetical, with each client operating on different data and architectures. In such a scenario, the incentive for federated participation is not clearly articulated.

2. The novelty of RELA is not fully evident. The client-wise gradient update formulation $\hat{g}_i^{(t)} = (1 - \alpha) \hat{g}_i^{(t-1)} + \alpha g^{(t)}_i$ closely resembles a first-order exponential moving average (EMA), similar to adaptive optimization methods such as Adam. Furthermore, the addition of a sanitization or noise component introduces privacy-related implications that warrant more rigorous analysis. If differential privacy–like noise is applied, the paper should evaluate its robustness against gradient-based privacy attacks and report accuracy trade-offs with and without the noise injection.

3. The paper attempts to address multiple orthogonal challenges simultaneously (data heterogeneity, model heterogeneity, privacy), which can dilute the focus of the contribution. A clearer ablation or modular analysis could help isolate the effects of each component. Currently it's unclear how impactful the "computing gradients at every $m$ batch is" or how impactful (accuracy-wise) the sanitized gradients are.

**Questions:**

1. In the related work section, most PFL citations are listed without discussion. It would be helpful to briefly summarize the current state of the field: What approaches do recent state-of-the-art methods adopt, and what limitations does FedMosaic specifically address beyond them?

2. The preliminaries conclude with a PFL objective, but the formulation of the global model objective is unclear. How does the given objective differ from the standard local objective, and why does it include terms dependent on other clients’ models?

3. In Equation (1) and Figure 2, are the gradients computed with respect to the frozen weights $W_s$?

4. The rationale for computing only the last-layer gradient (based on the proportionality of preceding gradients via the chain rule) requires further justification or empirical support. Are there results related to it in the appendix?

5. The paper mentions that gradients $g_i$ are computed every $m$ batch iterations rather than every batch. What is the observed accuracy trade-off with and without this optimization?

6. For PQ-LoRA, does the method assume that all clients use the same low-rank dimension $r$? If so, the approach still enforces a degree of architectural homogeneity. Given that the core challenge is model heterogeneity, how can we justify $P$ and $Q$ modules remaining dimensionally the same?

7. How are the LoRA parameters $A$ and $B$ trained?

---

> ### Author Response · Authors · 2025-11-19
> **Response to Reviewer UdQR (1/7)**
>
> Dear Reviewer UdQR,
>
> We sincerely thank you for your helpful feedback and insightful comments. We address your comments and questions below.
>
> > For PQ-LoRA, does the method assume that all clients use the same low-rank dimension $r$? If so, the approach still enforces a degree of architectural homogeneity. Given that the core challenge is model heterogeneity, how can we justify $P$ and $Q$ modules remaining dimensionally the same?
>
> $\to$ Thank you again for the great comment! You are correct that our main setting assumes a uniform LoRA rank across clients. To address this limitation, we integrate PQ-LoRA with HETLoRA [1], a heterogeneous FL method that allows clients to use LoRA modules with different ranks on the same backbone model (see Sec. A.24 for details). We evaluate a setting where clients use heterogeneous models with three different LoRA ranks (64, 128, 256) by integrating PQ-LoRA with HETLoRA, and summarize the results in Table B below. As shown, FedMosaic combined with HETLoRA substantially outperforms all baselines with the same HETLoRA integration. These results highlight that **PQ-LoRA is not limited to heterogeneous architectures with a shared rank, but can also handle heterogeneous ranks across heterogeneous architectures** when paired with existing hetero-rank LoRA FL approaches, further extending its applicability to real-world clients. We have included these results in Sec. A.24, Tab. 34 of the revision.
>
> **Table B. Comparison in rank heterogeneous PFL-Dynamic on DRAKE**
> |Method|Self $A_{last} \uparrow$|Self $A_\text{AUC} \uparrow$|Others $A_{last} \uparrow$|Others $A_\text{AUC} \uparrow$|
> |-|-|-|-|-|
> |SFT|65.35$\pm$0.62|57.89$\pm$0.27|46.98$\pm$0.25|46.26$\pm$0.20|
> |DITTO|60.68$\pm$0.09|54.69$\pm$0.13|47.20$\pm$0.11|46.46$\pm$0.16|
> |FedSim|63.46$\pm$0.88|56.57$\pm$0.31|46.03$\pm$0.29|45.58$\pm$0.12|
> |FedIT|65.15$\pm$0.71|57.82$\pm$0.43|47.02$\pm$0.49|46.12$\pm$0.35|
> |TAKFL|63.34$\pm$0.84|56.59$\pm$0.13|47.52$\pm$0.20|46.83$\pm$0.34|
> |FedDPA|63.66$\pm$0.80|56.12$\pm$0.46|46.97$\pm$0.44|46.22$\pm$0.23|
> |FedDAT|58.90$\pm$0.62|55.30$\pm$0.20|48.02$\pm$0.17|47.22$\pm$0.01|
> |PerAda|60.29$\pm$0.43|54.46$\pm$0.41|47.25$\pm$0.23|46.36$\pm$0.29|
> |FedMKT|61.35$\pm$0.41|55.07$\pm$0.12|46.47$\pm$0.67|46.06$\pm$0.33|
> |FedMosaic (Ours)|**67.71$\pm$0.33**|**59.28$\pm$0.15**|**52.40$\pm$0.16**|**48.07$\pm$0.15**|
>
> > The rationale for computing only the last-layer gradient (based on the proportionality of preceding gradients via the chain rule) requires further justification or empirical support. Are there results related to it in the appendix?
>
> $\to$ Thank you for the great comment! We have added empirical support for using only the last-layer gradient instead of the full-layer gradient in Sec.A.18 of the revision. As shown in Fig.22, using only the last-layer gradient achieves comparable performance compared to using gradients from multiple layers (e.g., 2, 4, 8), and even all layers while significantly reducing computational costs. **These empirical results support that the last-layer gradient provides a cost-efficient approximation to the full-layer gradients.**

---

> ### Author Response · Authors · 2025-11-19
> **Response to Reviewer UdQR (2/7)**
>
> > The novelty of RELA is not fully evident. The client-wise gradient update formulation
>  closely resembles a first-order exponential moving average (EMA), similar to adaptive optimization methods such as Adam.
>
> $\to$ To clarify, our client-wise gradient update is indeed formulated in an EMA manner, as noted in L202, and the novelty lies not in EMA itself, which is widely used across many domains, but in leveraging it for two core aspects of real-world FL: **handling distribution shifts and improving privacy preservation**. In RELA, we employ EMA specifically to: (i) track per-client gradients while accounting for data distribution shifts over time, inspired by the exponential decay of past information in forgetting, and (ii) improve resilience against privacy attacks (e.g., gradient inversion) by mixing gradients over times rather than utilizing per-sample gradients, as shown in Table C below. We believe these observed effects of EMA in RELA provide a promising direction for future research on further mitigating privacy risks and tackling distribution shifts in federated learning.
>
> Moreover, we also note that EMA is only one component of RELA, alongside client-specific weighted aggregation, gradient sampling and noise addition for sanitization, and the use of last-layer gradients. **These components collectively enable RELA to improve personalization under data heterogeneity while simultaneously mitigating privacy risks in federated learning, rather than relying on any single component alone.** We have provided more detail about this experiment in Sec. A.30 and included these results in Tab. 39 of the revision.
>
> **Table C. Comparison of privacy-defense methods under gradient inversion attack**
> |Defense Method|Recovered text Rouge-L $\downarrow$|
> |-|-|
> |Full layer gradient|0.2952|
> |Clipping+Noise|0.2720|
> |SVD Truncation|0.2688|
> |Fisher Information|0.2128|
> |Orthogonal Projection|0.1614|
> |||
> |Last layer gradient|0.1442|
> |+EMA aggregation|0.0844|
> |+Noise|0.0699|
> |+Subsampling|**0.0653**|
>
> > The addition of a sanitization or noise component introduces privacy-related implications that warrant more rigorous analysis. If differential privacy–like noise is applied, the paper should evaluate its robustness against gradient-based privacy attacks and report accuracy trade-offs with and without the noise injection.
>
> $\to$ We appreciate the reviewer’s thoughtful comment. We first clarify that we have already reported the effects of the noise scale $\mu$ in Fig. 20 (Sec. A.18), where accuracy decreases with larger noise scales. Regarding robustness to gradient-based privacy attacks and the privacy–accuracy trade-off, we have additionally conducted gradient inversion attacks [2] on Llama-1B using the GLUE-COLA dataset, and summarize the results in Sec. A.30 of the revision. As shown in Fig. 25, adding more Gaussian noise hinders text recovery, but increasing the noise level also degrades performance by reducing the representativeness and informativeness of the gradients. Based on this privacy-accuracy trade-off, we adopt a moderate noise scale ($1\times10^{-4}$) in RELA.
>
> > The paper attempts to address multiple orthogonal challenges simultaneously (data heterogeneity, model heterogeneity, privacy), which can dilute the focus of the contribution. A clearer ablation or modular analysis could help isolate the effects of each component. Currently, it's unclear “how impactful (accuracy-wise) the sanitized gradients are” or “how impactful the computing gradients at every $m$ batch is”.
>
> $\to$ We appreciate your feedback. We have already provided a detailed analysis of the impact of sanitized gradients in Sec. A.18, and we have also included an additional analysis of the gradient computation interval ($m$) in Fig. 21 (Sec. A.18) of the revision.
>
> Regarding the impact of sanitized gradients (i.e., the effects of noise scale $\mu$), as shown in Fig. 20, increasing the noise level degrades performance due to reduced representativeness and informativeness of the gradients, despite enhancing privacy protection (shown in Table C). Considering this trade-off, we adopt $\mu = 1 \times 10^{-4}$, which effectively balances accuracy and privacy.
>
> Regarding the additional analysis on the gradient computation interval ($m$), as shown in Fig. 21, computing gradients more frequently (i.e., smaller $m$) improves performance but increases computational cost, whereas computing them too infrequently (i.e., larger $m$) reduces cost but significantly degrades accuracy. Since a wide range of $m$ values (i.e., 1~20) maintains stable performance, we adopt $m = 10$ as a balanced choice that offers comparable accuracy with reduced computational overhead. We appreciate your feedback enhancing our comprehensive analysis!

---

> ### Author Response · Authors · 2025-11-19
> **Response to Reviewer UdQR (3/7)**
>
> > In the related work section, most PFL citations are listed without discussion. It would be helpful to briefly summarize the current state of the field: What approaches do recent state-of-the-art methods adopt, and what limitations does FedMosaic specifically address beyond them?
>
> $\to$ We thank the reviewer for this helpful suggestion. We have revised the related work section to more clearly summarize recent PFL approaches and the limitations that FedMosaic addresses.
>
> > The preliminaries conclude with a PFL objective, but the formulation of the global model objective is unclear. How does the given objective differ from the standard local objective, and why does it include terms dependent on other clients’ models?
>
> $\to$ Sorry for the confusion! We have revised our PFL objective to explicitly incorporate the global objective, clearly distinguishing it from the standard local objective by including the global module received from the server in addition to the client-specific local term. The dependence across clients in the objective arises because the global adapter $G$ is shared among all clients, providing shared global knowledge, thus, we express the objective as $\min_{\lbrace L_i, \dots, L_N \rbrace} \frac{1}{N} J_i$, following [3, 4, 5, 6].
>
> > In Equation (1) and Figure 2, are the gradients computed with respect to the frozen weights $W_s$?
>
> $\to$ Yes, we calculate the gradient of frozen weights, as we noted in Lines 193-196.
>
> > How are the LoRA parameters $A$ and $B$ trained?
>
> $\to$ LoRA parameters $A$ and $B$ are trained as follows: (i) in conventional LoRA, they are trained using the standard auto-regressive objective (i.e., cross-entropy loss), consistent with other PFL baselines, and (ii) in PQ-LoRA, they are frozen during federated learning, with the alignment step completed beforehand (L303), while only the $P$ and $Q$ modules are trained. We provide the details of PQ-LoRA alignment in Sec. 4.2.2. If you have any additional concerns regarding the training of $A$ and $B$, please let us know. Thank you!
>
> > The motivation for exploring multi-modal tasks in PFL requires further clarification. What are the practical or deployment-oriented use cases where clients naturally possess distinct modalities? At present, the setup appears somewhat hypothetical, with each client operating on different data and architectures. In such a scenario, the incentive for federated participation is not clearly articulated.
>
> $\to$ We appreciate the reviewer’s thoughtful question. We respectively argue that our proposed multi-modal PFL setup, which tackles (i) data heterogeneity, (ii) model heterogeneity and (iii) distribution shifts, is **well aligned with realistic and practical deployment scenarios**, as also highlighted by **Reviewers GXyk, oYUk, and pywM**. For further clarification, we describe below our motivation behind multi-modal PFL, and real-world applications where our framework is directly applicable.
>
> **[motivation of multi-modal PFL]**
>
> The motivation for multi-modal PFL is that Multimodal LLMs (MLLMs) are less foundational (i.e., achieve weaker generalization capabilities) than their uni-modal counterparts (e.g., LLMs for text and ViTs for vision), often demonstrating poor zero-shot generalization and even underperforming uni-modal foundation models on uni-modal tasks [7, 8, 9]. In contrast, uni-modal foundation models (e.g., LLMs) already achieve near-human zero-shot performance on a wide range of downstream tasks [10, 11], reducing the need for additional personalization. The larger performance gap between personalized MLLMs and off-the-shelf MLLMs, compared to the relatively small gap in uni-modal settings, motivates our focus on personalization in the multi-modal scenario.
>
> Moreover, since training solely on specific tasks can degrade generalization, and training large foundation models for personalization requires substantial computational cost, we aim to accelerate personalization while preserving generalization through collaborative learning by PFL. In addition, because real-world tasks are inherently multi-modal rather than uni-modal, this further motivates our focus on multi-modal PFL, as we illustrate with several practical examples below.

---

> ### Author Response · Authors · 2025-11-19
> **Response to Reviewer UdQR (4/7)**
>
> **[practical applications of multi-modal PFL]**
>
> Recently emerging **Model Context Protocol (MCP) [12]-based agents** provide a practical FL use case [13], where each client naturally handles multi-modal data. Using MCP, each user usually solves multi-modal instruction-following tasks, such as redesigning layouts from mockups and user guides, and searching store goods using reference images, using vision-language models connected to support tools (e.g., file systems, UIs, apps).
> Difference in purposes and tool usage induce task heterogeneity, while diverse hardware configurations introduce model heterogeneity (from small on-device agents to larger GPU-hosted ones). Since centralized fine-tuning is infeasible with user-specific data due to privacy risks, multi-modal PFL provides a practical solution, enabling users with similar agentic usage patterns to benefit from one another without sharing raw data. We believe DRAKE, a task-diverse multi-modal PFL benchmark with distribution shifts across VQA and instruction-following tasks, can closely approximate this setting.
>
> Not only the MCP AI agent, **home-assistant robots** operate in a multi-modal setting too (egocentric vision + text instructions) and continually learn new behaviors and adapt to evolving environments for personalization [14, 15]. Each robot’s training data is user-specific (i.e., data heterogeneity) and evolves over time (i.e., distribution shifts). Furthermore, due to differences in economic conditions, usage patterns, and purchase timelines, users naturally own robots with varying hardware and model architectures, creating model heterogeneity as newer robot models become available over time. In such scenarios, one robot may encounter new behaviors or environments that other robots have already experienced similar situations. Sharing knowledge across robots can therefore accelerate personalization and improve generalization to unseen scenarios. Furthermore, because egocentric videos captured by robots often contain highly sensitive information (e.g., private documents, personal identifiers), centralizing data is infeasible, making PFL a necessary and practical solution for privacy-preserving knowledge sharing.
>
> **Hospitals** provide another practical application of our setup. As new patients arrive and research on emerging diseases advances, data distributions shift continuously. Hospitals also differ in specialties and patient populations, resulting in distinct tasks (e.g., diabetes prediction, AIDS diagnosis), and they deploy different foundation models based on their resources and requirements, creating model heterogeneity. Because medical data are highly privacy-sensitive, centralizing datasets is infeasible. In this context, federated learning enables hospitals to share knowledge across heterogeneous models and datasets, yielding more accurate and generalizable medical AI systems, much like physicians collaborate across departments through consultations to achieve better diagnoses and treatments.
>
> We believe that our FedMosaic framework, which explicitly accounts for multiple heterogeneities and distribution shifts, offers a foundation for extending PFL research to diverse real-world applications.

---

> ### Author Response · Authors · 2025-11-19
> **Response to Reviewer UdQR (5/7)**
>
> > The comparison against prior works using non-IID splits of a single dataset may not be entirely equitable. The contextual settings differ significantly (maybe the earlier studies targeted models specialized for single-domain or unimodal tasks, rather than fine-tuning billion-parameter foundation models).
>
> $\to$ For appropriate and equitable comparisons, we carefully choose baselines that have been **developed under conditions as similar as possible** and applicable to our scenarios  (i.e., distribution-, modality-, and mode size- agnostic methods). If the reviewer has suggestions for other baselines we should compare against, that they would find more fair, we'd like to receive concrete pointers. Furthermore, to show that FedMosaic’s superiority stems from genuine robustness rather than unfair comparison, **we evaluate it under standard PFL settings: (i) non-i.i.d. splits and (ii) small-scale models, and (iii) uni-modal (text-only NLP) LLM results.**
>
> **[Selecting the most equitable baselines for our setup]**
>
> To the best of our knowledge, we are the first to evaluate PFL under the full combination of data heterogeneity, model heterogeneity, foundation models, and multi-modal, whereas prior studies typically adopt simplified setups (e.g., addressing only subsets of these factors or oversimplifying heterogeneity), as also noted by **Reviewers oYUk and pywM**. Therefore, among existing methods, we choose baselines that most closely align with our setting.
>
> Specifically, regarding model scales, many of our baselines also target billion-parameter foundation models (e.g., FedDPA, FedIT, FedMKT using Llama-7B) or million-parameter models (e.g., TAKFL, FedDAT). Although several baselines (e.g., DITTO, FedSim, PerAda) were originally evaluated with small-scale models (e.g., ResNet), their formulations are not limited to specific architectures, thus, we included them in our comparison. In contrast, we intentionally exclude methods [16, 17, 18] designed exclusively for specific architectures (e.g., channel-wise gradient differentiation [17], which is only applicable to CNN-based models), to maintain a technically appropriate comparison.
>
> Similarly, regarding modality and task, the baselines we compare against are designed to be broadly applicable rather than tailored to a specific task or modality. For example, FedMKT shares logits computed on public data, and PerAda trains a global model using logits produced by local models on the same public data. These mechanisms are independent of any particular modality or task setup.
>
>
> **[Comparison under standard non-i.i.d. splits, small-scale models, and uni-modal setup]**
>
> Finally, we respectfully argue that **the superiority of FedMosaic in multi-modal PFL setup does not stem from any unfair comparison, but rather from its general effectiveness across data splits, architectures, and modalities**. To substantiate this claim, we additionally evaluate FedMosaic under standard PFL settings used in baselines, including (i) standard non-i.i.d. splits and (ii) experiments with small-scale models. We also emphasize that (iii) **uni-modal (text-only NLP) experiments with LLMs are already reported in Tab.6.**
>
> For the standard non-i.i.d. splits, we merge 10 tasks in the DRAKE benchmark and partition them into client subsets using Dirichlet distributions ($\alpha = 0.1, 0.5, 1.0$), following the setups used in baselines (e.g., PerAda and TAKFL). As shown in Table D, FedMosaic also outperforms baselines in this setup. In addition, we evaluate FedMosaic with small-scale models using the T5 architecture [19] (60M parameters) and summarize the results in Table E and Table F. FedMosaic again surpasses baselines in this setting. For the uni-modal setup, FedMosaic consistently outperforms baselines in the text-only NLP domain, the original experimental setup used by baselines (FedDPA, FedIT, and FedMKT), as shown in Table 6. Together, these results demonstrate that FedMosaic’s superiority does not arise from inequitable comparisons, but rather reflects its robust and generalizable effectiveness across diverse modalities, data distributions, and model architectures. These results can be found in the following sections of the revision:
> - Table D: Sec. A.25, Tab. 35
> - Table E and Table F: Sec. A.26, Tab. 36

---

> ### Author Response · Authors · 2025-11-19
> **Response to Reviewer UdQR (6/7)**
>
> **Table D. Comparison in heterogeneous non-i.i.d. PFL on DRAKE**
> |Dirichlet $\alpha$|$\alpha=1.0$||$\alpha=0.5$||$\alpha=0.1$||
> |-|-|-|-|-|-|-|
> |**Method**|$A_{last} \uparrow$|$A_\text{AUC} \uparrow$|$A_{last} \uparrow$|$A_\text{AUC} \uparrow$|$A_{last} \uparrow$|$A_\text{AUC} \uparrow$|
> |SFT|66.77$\pm$0.05|62.38$\pm$0.58|68.25$\pm$0.40|63.73$\pm$1.04|71.40$\pm$0.99|66.90$\pm$0.37|
> |DITTO|62.00$\pm$0.12|58.37$\pm$0.49|62.34$\pm$0.12|59.52$\pm$0.38|66.80$\pm$0.95|63.46$\pm$0.02|
> |FedSIM|63.73$\pm$0.03|59.82$\pm$0.27|67.44$\pm$0.33|62.77$\pm$0.78|69.90$\pm$1.97|65.03$\pm$1.51|
> |FedIT|66.89$\pm$0.77|62.57$\pm$0.32|67.65$\pm$0.58|63.46$\pm$1.27|72.07$\pm$0.36|67.17$\pm$0.31|
> |TAKFL|65.18$\pm$0.46|63.57$\pm$0.59|66.65$\pm$0.37|62.08$\pm$0.29|69.88$\pm$1.69|65.21$\pm$1.19|
> |FedDPA|64.34$\pm$0.27|60.34$\pm$0.35|65.58$\pm$2.72|61.91$\pm$1.72|68.56$\pm$0.27|64.58$\pm$0.03|
> |FedDAT|62.99$\pm$0.01|59.19$\pm$0.09|63.01$\pm$1.47|59.95$\pm$0.65|66.43$\pm$0.14|63.16$\pm$0.22|
> |PerADA|61.73$\pm$0.61|58.20$\pm$0.68|62.88$\pm$1.09|59.77$\pm$0.76|66.41$\pm$0.61|63.27$\pm$0.28|
> |FedMKT|64.87$\pm$0.30|60.67$\pm$0.12|65.14$\pm$0.10|60.81$\pm$0.39|69.04$\pm$0.83|64.78$\pm$0.33|
> |FedMosaic (Ours)|**69.12$\pm$0.39**|**64.72$\pm$0.02**|**70.20$\pm$0.64**|**64.68$\pm$0.20**|**73.12$\pm$1.32**|**68.39$\pm$0.89**|
>
> **Table E. Comparison in heterogeneous PFL-dynamic with small-scale models (2 clients T5-small, 2 clients T5-base): total 6 rounds, 100 local steps per round**
> |Method|Self $A_{last} \uparrow$|Self $A_\text{AUC} \uparrow$|Others $A_{last} \uparrow$|Others $A_\text{AUC} \uparrow$|
> |-|-|-|-|-|
> |SFT|60.94$\pm$1.35|50.96$\pm$0.57|26.19$\pm$0.80|24.49$\pm$0.32|
> |DITTO|59.22$\pm$1.08|47.89$\pm$0.64|24.68$\pm$0.14|24.29$\pm$0.05|
> |FedSIM|60.66$\pm$0.41|50.70$\pm$0.12|25.33$\pm$0.78|24.24$\pm$0.05|
> |FedIT|60.53$\pm$0.53|50.76$\pm$0.04|25.98$\pm$0.62|23.93$\pm$0.36|
> |TAKFL|60.49$\pm$0.54|49.88$\pm$0.15|25.37$\pm$0.24|23.60$\pm$0.35|
> |FedDPA|61.04$\pm$0.48|50.19$\pm$0.13|25.78$\pm$0.08|23.84$\pm$0.24|
> |FedDAT|51.13$\pm$0.27|43.25$\pm$0.08|24.29$\pm$0.09|24.41$\pm$0.01|
> |PerADA|59.53$\pm$1.01|47.99$\pm$0.60|24.69$\pm$0.22|24.31$\pm$0.06|
> |FedMKT|60.29$\pm$0.45|49.14$\pm$0.64|25.41$\pm$0.42|23.63$\pm$0.21|
> |FedMosaic (Ours)|**71.20$\pm$1.54**|**55.28$\pm$1.11**|**40.33$\pm$0.04**|**31.20$\pm$0.60**|
>
> **Table F. Quantitative comparison in heterogeneous PFL-dynamic with small-scale models (2 clients T5-small, 2 clients T5-base): total 20 rounds, 50 local steps per round**
> |Method|Self $A_{last} \uparrow$|Self $A_\text{AUC} \uparrow$|Others $A_{last} \uparrow$|Others $A_\text{AUC} \uparrow$|
> |-|-|-|-|-|
> |SFT|62.88$\pm$0.36|51.20$\pm$0.22|26.94$\pm$1.34|25.26$\pm$0.50|
> |DITTO|60.40$\pm$1.38|48.99$\pm$0.39|25.36$\pm$0.41|24.01$\pm$0.18|
> |FedSIM|61.88$\pm$0.91|49.55$\pm$0.11|26.39$\pm$0.66|23.90$\pm$0.09|
> |FedIT|62.34$\pm$0.64|50.85$\pm$0.10|26.92$\pm$1.58|24.40$\pm$0.52|
> |TAKFL|61.05$\pm$1.68|49.89$\pm$0.57|26.02$\pm$0.09|23.66$\pm$0.07|
> |FedDPA|61.47$\pm$0.66|49.09$\pm$0.32|25.77$\pm$0.58|24.19$\pm$0.33|
> |FedDAT|53.32$\pm$0.10|44.65$\pm$0.38|23.69$\pm$0.40|23.39$\pm$0.06|
> |PerADA|60.80$\pm$0.93|49.05$\pm$0.16|25.33$\pm$0.13|24.11$\pm$0.30|
> |FedMKT|62.21$\pm$0.08|50.25$\pm$0.68|24.82$\pm$0.28|23.27$\pm$0.15|
> |FedMosaic (Ours)|**78.24$\pm$0.14**|**60.32$\pm$0.02**|**37.14$\pm$0.82**|**31.12$\pm$0.33**|

---

> ### Author Response · Authors · 2025-11-20
> **Response to Reviewer UdQR (7/7)**
>
> [1] Cho et al., Heterogeneous LoRA for Federated Fine-tuning of On-Device Foundation Models, EMNLP 2024 \
> [2] Balunovic et al., LAMP: Extracting Text from Gradients with Language Model Priors, NeurIPS 2022 \
> [3] Shamsian et al., Personalized Federated Learning using Hypernetworks, ICML 2021 \
> [4] Zhang et al., Personalized Federated Learning with First Order Model Optimization, ICLR 2021 \
> [5] Jiang et al., Test-Time Robust Personalization for Federated Learning, ICLR 2023 \
> [6]  Tamirisa et al., FedSelect: Personalized Federated Learning with Customized Selection of Parameters for Fine-Tuning, CVPR 2024 \
> [7] Fu et al., BLINK: Multimodal Large Language Models Can See but Not Perceive, ECCV 2024 \
> [8] Tahan et al., UniBench: Visual Reasoning Requires Rethinking Vision-Language Beyond Scaling, NeurIPS 2024 D&B Track \
> [9] Tong et al., Eyes Wide Shut? Exploring the Visual Shortcomings of Multimodal LLMs, CVPR 2024 \
> [10] OpenAI, OpenAI o1: Learning to Reason with LLMs, 2024 \
> [11] Latif et al., Can OpenAI o1 outperform humans in higher-order cognitive thinking, arXiv 2024 \
> [12] Antropic, Introducing the Model Context Protocol, 2024 (https://www.anthropic.com/news/model-context-protocol) \
> [13] BytePlus, MCP federated implementation, 2025 (https://www.byteplus.com/en/topic/541259?title=mcp-federated-implementation) \
> [14] Kim et al., Online Continual Learning for Interactive Instruction Following Agents, ICLR 2024 \
> [15] Yang et al., Task-agnostic Lifelong Robot Learning with Retrieval-based Weighted Local Adaptation, arXiv 2024 \
> [16] Oh et al., FedBABU: Towards Enhanced Representation for Federated Image Classification, ICLR 2022 \
> [17] Xia et al., Personalized Semantics Excitation for Federated Image Classification, ICCV 2023 \
> [18] Chen et al., Optimizing Personalized Federated Learning through Adaptive Layer-Wise Learning, IJCAI 2025 \
> [19] Raffel et al., Exploring the Limits of Transfer Learning with a Unified Text-to-Text Transformer, JMLR 2020

---

> > ### Comment · Reviewer_UdQR · 2025-11-21
> >
> > Thank you to the authors for their detailed response.
> >
> > 1. Would it be possible to see the results of HETLoRA (without PQ-LoRA) in Table B (Comparison in rank heterogeneous PFL-Dynamic on DRAKE), to get a clearer idea on the improvements brought by PQ-LoRA?
> >
> > 2. Is my understanding correct that LoRA parameters $A$ and $B$ are trained on the server? If so, how is the training data split for the server's proxy dataset and clients' datasets?
> >
> > 3. Thank you for providing practical applications for the proposed work. However, I still doubt that any hospital would rely on a single unified model to train on fundamentally different diseases. Is there any evidence on that actually happening?
> >
> > Likewise, it seems like it's unrealistic to assume that home robots could autonomously annotate or label the diverse and nuanced data they encounter. These assumptions gloss over the substantial operational burden of data curation, annotation quality control, and domain-expert oversight that real-world systems necessarily depend on.
> >
> > Besides, would it be possible to get details on the memory usage of FedMosaic? I want to understand whether MLLM training on such edge hardware is feasible in terms of memory.
> >
> > 4. May I get more details on how distribution shifts are simulated? Do all clients change their distributions (or tasks) simultaneously?
> >
> > 5. I would still argue that the inclusion of EMA for handling distribution shifts is not novel, there are works which incorporate adaptive optimizers to achieve the same [1, 2].
> >
> > [1] Flash: Concept Drift Adaptation in Federated Learning (Panchal et al., ICML 2023)
> >
> > [2] Adaptive Federated Learning in Presence of Concept Drift (Canonaco et al., IJCNN 2021)
> >
> > Similarly, adding of noise to achieve resilience against gradient inversion attacks has been extensively explored in differential privacy + FL literature.
> >
> > 6. The claims of the work still feel conflated, because support for multiple modalities or multiple images is fundamentally a property of the underlying foundation models used in this work, not of the proposed method. In Table 1, the authors compare FedMosaic against several prior approaches, but it is unclear whether all of those methods use the same foundation model as this work does.
> >
> > Even for the claim of "Multi-Data Sources," this capability largely stems from the underlying foundation multi-modal LLMs. This proposed work can leverage it because it is training such models, whereas prior methods may not demonstrate this property simply because their primary goal was not to train foundation-level multi-modal LLMs. In that sense, the mere act of training or fine-tuning a new set of models is not, by itself, a novel contribution.

---

> > > ### Author Response · Authors · 2025-11-23
> > > **Follow-up Response to Reviewer UdQR (5/5)**
> > >
> > > [1] Li et al., Privacy-preserving federated brain tumour segmentation, MICCAI 2019 \
> > > [2] Dayan et al., Federated learning for predicting clinical outcomes in patients with COVID-19, Nature Medicine 2021 \
> > > [3] Riaz et al., Association between obesity and cardiovascular outcomes: a systematic review and meta-analysis of Mendelian randomization studies, JAMA Network 2018 \
> > > [4] Bosetti et al., Diabetes, antidiabetic medications, and pancreatic cancer risk: an analysis from the International Pancreatic Cancer Case-Control Consortium, Annals of oncology 2014 \
> > > [5] Irpan et al., Do As I Can, Not As I Say: Grounding Language in Robotic Affordances, Google, CoRL 2022 \
> > > [6] Everyday Robots (https://x.company/projects/everyday-robots/?utm_source=chatgpt.com) \
> > > [7] Wang et al., Personalization in Human-Robot Interaction through Preference-based Action Representation Learning, ICRA 2025 \
> > > [8] Ding et al., Learning a Universal Human Prior for Dexterous Manipulation from Human Preference, RSS Workshop 2023 \
> > > [9] Kinova Jaco assistive robotic arm (https://assistive.kinovarobotics.com/product/jaco-robotic-arm) \
> > > [10] Wimer et al., PPHR: A Personalized AI System for Proactive Robots, IHCI 2023 \
> > > [11] Rengarajan et al., Federated Ensemble-Directed Offline Reinforcement Learning, NeurIPS 2024 \
> > > [12] Miao et al., FedVLA: Federated Vision-Language-Action Learning with Dual Gating
> > > Mixture-of-Experts for Robotic Manipulation, ICCV 2025 \
> > > [13]  AIEdge-X300-RTX30 - https://www.nexcom.com/Products/multi-media-solutions/ai-edge-computer/nvidia-solutions/ai-edge-computer-aiedge-x-300-rtx30/OrderingInformation?utm_source=chatgpt.com \
> > > [14] Nuvo-8208GC Series - https://www.neousys-tech.com/en/product/product-lines/edge-ai-gpu-computing/nuvo-8208gc-intel-8th-gen-dual-nvidia-rtx-2080ti-gpu-computing-platform \
> > > [15] Zhao et al., A Federated Framework for LLM-based Recommendation, ACL 2025 Findings \
> > > [16] Wang et al., pFedSAM: Personalized Federated Learning of Segment Anything Model for Medical Image Segmentation, arXiv 2025

---

> ### Author Response · Authors · 2025-11-23
> **Follow-up Response to Reviewer UdQR (1/5)**
>
> Thank you very much for your thoughtful follow-up questions and for actively engaging in the discussion. We sincerely appreciate the detailed clarifications and experimental requests you have raised. They greatly help us improve the quality of our work. Moreover, we would be glad to further address any additional concerns you may have. We hope the following response addresses the reviewer’s questions.
>
>
> > Would it be possible to see the results of HETLoRA (without PQ-LoRA) in Table B (Comparison in rank heterogeneous PFL-Dynamic on DRAKE), to get a clearer idea on the improvements brought by PQ-LoRA?
>
> $\to$ Of course! To illustrate the improvements introduced by PQ-LoRA, we compare three variants: HETLoRA, HETLoRA + RELA, and HETLoRA + RELA + PQ-LoRA (our full method). As shown in Table H, combining HETLoRA with RELA improves personalization performance (Self $A_{last}$, Self $A_{\text{AUC}}$) by facilitating knowledge sharing among clients focusing on similar tasks. Adding PQ-LoRA on top of this (HETLoRA + RELA + PQ-LoRA) further boosts personalization and **yields substantial gains in generalization performance** (Others $A_{last}$, Others $A_{\text{AUC}}$), demonstrating that **PQ-LoRA enables effective knowledge sharing even across models with heterogeneous ranks**.
>
> **Table H. Comparison in rank heterogeneous PFL-Dynamic on DRAKE**
> |Method|Self $A_{last}$|Self $A_\text{AUC}$|Others $A_{last}$|Others $A_\text{AUC}$|
> |-|-|-|-|-|
> |HETLoRA|65.15$\pm$0.71|57.82$\pm$0.43|47.02$\pm$0.49|46.12$\pm$0.35|
> |HETLoRA + RELA|66.11$\pm$0.78|58.25$\pm$0.45|47.69$\pm$0.11|46.50$\pm$0.25|
> |HETLoRA + RELA +  PQ-LoRA (Ours) |**67.71$\pm$0.33**|**59.28$\pm$0.15**|**52.40$\pm$0.16**|**48.07$\pm$0.15**|
>
>
> > Is my understanding correct that LoRA parameters $A$  and $B$ are trained on the server? If so, how is the training data split for the server's proxy dataset and clients' datasets?
>
> $\to$  (We believe the reviewer is referring to parameters $A$ and $B$ in PQ-LoRA.) Yes, these parameters are trained (i.e., aligned) on the server **only once before collaborative training to align heterogeneous models**. Importantly, the server’s training data are not split from any client datasets. Instead, it uses a publicly available dataset that is disjoint from all clients’ downstream tasks, and whose specific choice has negligible impact on performance. As detailed in Sec. A.17, the public data can range from an MLLM’s pre-training data to visual storytelling datasets (requiring multi-sentence outputs, unlike clients’ single-sentence tasks), or even text-only (i.e., uni-modal) NLP data, without degrading performance. This is because the server-side training (i.e., aligning) of $A$ and $B$ aims not to optimize for any particular downstream task, but simply to ensure that heterogeneous models share a common initialization, making the alignment robust to the choice of public data.
>
> Note that, in conventional LoRA, the parameters $A$ and $B$ are trained separately on each client, and the server only aggregates them without performing any training, consistent with other baselines.
>
> > I would still argue that the inclusion of EMA for handling distribution shifts is not novel, there are works that incorporate adaptive optimizers to achieve the same [1, 2]. Similarly, adding of noise to achieve resilience against gradient inversion attacks has been extensively explored in differential privacy + FL literature.
>
> $\to$ Thank you for raising this concern! We agree that using EMA to handle distribution shifts and adding noise for resilience to gradient inversion attacks are not novel on their own. However, we respectfully note that **the core contribution of RELA is task-relevance-guided aggregation, which enables effective knowledge sharing among clients focusing on similar tasks, with EMA and noise serving only as auxiliary components to address challenges that arise when using gradients to measure task similarity across clients**.
>
> Moreover, we clarify that EMA is not used solely for handling distribution shifts, but rather plays a major role in reducing privacy risks by mixing gradients across time, achieving a 5.98% inversion reduction (Table C). Similarly, we note that noise addition is not a standalone main component for preventing gradient inversion attacks and plays only a small additional role (1.45% inversion reduction) alongside EMA and gradient subsampling. In contrast, the use of last-layer gradients is the primary mechanism that mitigates inversion risks (15.1% inversion reduction) while also lowering computational cost, as shown in Table C.

---

> ### Author Response · Authors · 2025-11-23
> **Follow-up Response to Reviewer UdQR (2/5)**
>
> > Thank you for providing practical applications for the proposed work.
> >> However, I still doubt that any hospital would rely on a single unified model to train on fundamentally different diseases. Is there any evidence on that actually happening?
>
> $\to$ We first would like to clarify that our heterogeneous PFL setup does not assume a single unified model. Instead, we assume that hospitals maintain their own heterogeneous models tailored to their respective clinical focuses, and the goal is to enable collaborative learning across these heterogeneous models, not to merge them into a single unified model.
>
> Moreover, existing real-world deployments of federated learning in hospitals have primarily focused on a single clinical task shared across institutions [1, 2], where FL is used to aggregate distributed datasets for the same disease (e.g., tumor segmentation or COVID-19). We argue that this constraint is not due to a lack of need for collaboration across different diseases, but rather due to a methodological limitation: current FL algorithms are unable to support effective knowledge sharing when clients tackle different tasks using different model architectures tailored to those tasks. While we agree that collaboratively training models for fundamentally different diseases is neither necessary nor realistic, it is well-established that **many diseases are interrelated or share underlying physiological patterns (e.g., Obesity and Heart disease [3], ​​Diabetes mellitus and Pancreatic cancer [4]). As physicians collaborate across departments through consultations to achieve better diagnoses and treatments, we believe that FedMosaic can help bridge the effectiveness gap between federated models and real-world clinical decision-making by enabling collaboration among clients working on different diseases and using different architectures.**
>
>
> >> Likewise, it seems like it's unrealistic to assume that home robots could autonomously annotate or label the diverse and nuanced data they encounter. These assumptions gloss over the substantial operational burden of data curation, annotation quality control, and domain-expert oversight that real-world systems necessarily depend on.
>
> $\to$ Thank you for raising this valuable concern! We would like to clarify that personalizing a deployed home robot to a specific user does not necessarily require large-scale expert annotations or autonomous labeling. Instead, **personalization can often be achieved through weak or indirect supervision within RL frameworks, where the robot improves its policy based on simple user-provided feedback, such as binary success/failure signals or preference indications for a given instruction**.
>
> For example, SayCan [5] demonstrates that, given an LLM and a pre-trained robot policy, a robot can learn a high-level instruction (e.g., “I spilled my drink, can you help me?”) by selecting an appropriate skill from pre-trained simple actions (e.g., find a sponge, find a cleaner), considering the user’s environment. To train the system to follow user instructions effectively, the user only needs to provide minimal feedback, simply indicating whether the robot’s behavior succeeded or failed, which allows the robot to adapt to user-specific preferences without requiring expert-labeled data or robot-driven annotation. Notably, their experiments were conducted on real robots (Google’s Everyday Robots [6]), rather than being restricted to simulated environments.
>
> Similarly, several works have shown that robot agents can be aligned using human preference signals alone [7, 8]. For instance, in [7], a real robot for feeding tasks (the Kinova Jaco assistive robotic arm [9]) learns user-aligned behaviors simply by asking the user which of two possible arm trajectories they prefer, again without requiring expert annotations.
>
> Based on this efficient and convenient personalization process, users can easily adapt their own robot agents to their individual needs. Building on such personalization, recent works [10, 11, 12] explore improving performance through knowledge sharing across personalized robots working on different tasks, but these approaches generally assume homogeneous robot models. **We believe that FedMosaic offers a meaningful step toward enabling knowledge sharing even across heterogeneous robot agents, making collaborative learning more broadly applicable and more scalable to real-world scenarios.**

---

> ### Author Response · Authors · 2025-11-23
> **Follow-up Response to Reviewer UdQR (3/5)**
>
> > Besides, would it be possible to get details on the memory usage of FedMosaic? I want to understand whether MLLM training on such edge hardware is feasible in terms of memory.
>
> $\to$ We appreciate the reviewer’s thoughtful question. We summarize the GPU memory usage of FedMosaic as well as other PFL baselines in Tables I and J. As shown, **training a 1B MLLM with FedMosaic requires less than 10 GB, while training a 3B MLLM requires less than 20 GB**. Given that recent edge devices provide sufficient GPU memory, for example, AIEdge-X300-RTX30 [13] supports NVIDIA RTX 30-series GPUs (e.g., RTX 3090 with 24 GB), and the Nuvo-8208GC Series [14] even supports dual RTX 30-series GPUs (e.g., 24 GB each, totaling 48 GB), **training MLLMs on edge hardware with FedMosaic is indeed feasible**.
>
> Regarding the comparison of GPU memory consumption during training with other baselines, FedMosaic incurs only a small overhead compared to SFT, while using less memory than baselines that require training both local and global adapters (i.e., dual adapters). Note that although FedMosaic also employs a dual-adapter structure, its additional memory usage remains small because: (i) the global adapter is frozen, while other baselines train both, (ii) in PQ-LoRA, the $A$ and $B$ matrices are frozen while only the $P$ and $Q$ matrices are updated, and (iii) RELA uses only the ‘last-layer gradient’ of the frozen weight $W_s$.
>
> **Table I. GPU memory usage during local training of LLaVA-Llama-1B model**
> |Method|Memory usage type|GPU memory usage (GB)|
> |-|-|-|
> |SFT|Single adapter train|8.065|
> |DITTO|Dual adapter train|9.932|
> |FedSim|Dual adapter train|11.836|
> |FedIT|Single adapter train|8.065|
> |TAKFL|Single adapter train|8.065|
> |FedDPA|Dual adapter train|9.933|
> |FedDAT|Triple adapter train|10.679|
> |PerADA|Dual adapter train|9.932|
> |FedMKT|Single adapter train|8.065|
> |FedMosaic (Ours) |Single adapter train w/ PQ-LoRA|9.076|
>
> **Table J. GPU memory usage during local training of LLaVA-Llama-3B model**
> |Method|Memory usage type|GPU memory usage (GB)|
> |-|-|-|
> |SFT|Single adapter train|14.579|
> |FedIT|Single adapter train|14.579|
> |TAKFL|Single adapter train|14.579|
> |FedMKT|Single adapter train|14.579|
> |DITTO|Dual adapter train|19.565|
> |FedSim|Dual adapter train|20.279|
> |FedDPA|Dual adapter train|19.623|
> |PerADA|Dual adapter train|19.565|
> |FedDAT|Triple adapter train|20.561|
> |FedMosaic (Ours) |Single adapter train w/ PQ-LoRA|18.130|
>
>
> > May I get more details on how distribution shifts are simulated? Do all clients change their distributions (or tasks) simultaneously?
>
> $\to$ Yes, all clients experience the distribution shift simultaneously (i.e., at the same round) in our experiments. Specifically, in both HFLB-Dynamic and DRAKE-Dynamic experiments, each client learns 4 tasks sequentially over 20 rounds, with a distribution shift occurring every 5 rounds.
>
> However, we note that it can naturally be extended to an “asynchronized distribution-shift” setup, where distribution shifts occur at different times across clients. To demonstrate that FedMosaic works robustly regardless of the timing of distribution shifts, we evaluate FedMosaic under the asynchronized distribution-shift setup. Specifically, instead of assigning every client a fixed 5 rounds per task, each client randomly adopts one of three task-round patterns: (5,5,5,5), (3,7,3,7), or (7,3,7,3), where each number denotes the number of rounds allocated to each task in order. As shown in Table K below, FedMosaic consistently outperforms other baselines even under this asynchronous distribution-shift scenario, demonstrating its stability to heterogeneity in the timing of distribution shifts.
>
> **Table K. Comparison in asynchronized distribution-shift heterogeneous PFL-Dynamic on DRAKE**
> |Method|Self $A_{last}$|Self $A_\text{AUC}$|Others $A_{last}$|Others $A_\text{AUC}$|
> |-|-|-|-|-|
> |SFT|66.23$\pm$0.26|58.35$\pm$0.28|46.94$\pm$0.34|46.36$\pm$0.28|
> |DITTO|60.11$\pm$0.80|54.60$\pm$0.15|47.27$\pm$0.28|46.43$\pm$0.10|
> |FedSIM|64.71$\pm$0.49|57.62$\pm$0.25|46.44$\pm$0.13|45.91$\pm$0.08|
> |FedIT|66.77$\pm$0.25|58.71$\pm$0.24|46.92$\pm$0.19|46.32$\pm$0.15|
> |PerADA|59.82$\pm$0.39|54.49$\pm$0.16|47.18$\pm$0.17|46.43$\pm$0.06|
> |TAKFL|65.47$\pm$0.95|58.07$\pm$0.52|47.14$\pm$0.23|46.46$\pm$0.02|
> |FedDPA|62.40$\pm$0.82|56.26$\pm$0.21|47.33$\pm$0.24|46.53$\pm$0.12|
> |FedDAT|59.06$\pm$0.35|54.79$\pm$0.21|48.90$\pm$0.09|47.83$\pm$0.06|
> |FedMKT|61.76$\pm$0.06|55.79$\pm$0.15|46.76$\pm$0.18|46.30$\pm$0.07|
> |FedMosaic (Ours)|**68.30**$\pm$**0.45**|**60.23**$\pm$**0.07**|**51.96**$\pm$**0.26**|**49.63**$\pm$**0.10**|

---

> ### Author Response · Authors · 2025-11-23
> **Follow-up Response to Reviewer UdQR (4/5)**
>
> > The claims of the work still feel conflated, because support for multiple modalities or multiple images is fundamentally a property of the underlying foundation models used in this work, not of the proposed method.
> >> In Table 1, the authors compare FedMosaic against several prior approaches, but it is unclear whether all of those methods use the same foundation model as this work does.
>
> $\to$ Thank you for bringing up this point! We respectfully clarify that **many of our baselines do not target small-scale models but already operate on large-scale models**, with several using billion-parameter foundation models (e.g., FedDPA, FedIT, FedMKT with Llama-7B) and others using million-parameter models (e.g., TAKFL, FedDAT). Moreover, in our uni-modal NLP experiments (Table 6), we use the same Llama models employed by several baselines (e.g., FedDPA, FedIT, FedMKT), and FedMosaic consistently outperforms them, demonstrating that the performance gain does not stem from the specific multi-modal models we used. Although we also include baselines (e.g., DITTO, FedSim, and PerAda), which originally did not use large-scale models, to ensure a broad and fair comparison, their formulations are not tied to specific architectures, **as evidenced by their use with large-scale models in other works** [15, 16]. For example, although DITTO was originally evaluated with simple CNN models for classification, [15] includes DITTO as a FL baseline using Llama-7B as the underlying model, and [16] evaluates DITTO with SAM as the base model.
>
> Finally, we would like to note that many newer foundation models have emerged since prior baselines were published, making it impractical to employ only the exact models used in earlier works, as this would limit the evaluation to older models only. Therefore, to demonstrate FedMosaic’s general applicability, **we provide experiments not only with recent multi-modal foundation models, but also with the same models used in prior baselines, such as Llama models in Table 6 (Sec. 6.2) and in Table 20 (Sec. A.15), and T5-based models in Table 36 (Sec. A.26)**.
>
>
> >> Even for the claim of "Multi-Data Sources," this capability largely stems from the underlying foundation multi-modal LLMs. This proposed work can leverage it because it is training such models, whereas prior methods may not demonstrate this property simply because their primary goal was not to train foundation-level multi-modal LLMs. In that sense, the mere act of training or fine-tuning a new set of models is not, by itself, a novel contribution.
>
> $\to$ We respectfully clarify that our major contribution is **addressing challenges in heterogeneous clients in collaborative learning**, not tackling federated learning in MLLMs. In other words, FedMosaic is not limited to, nor primarily focused on, federated learning of multi-modal foundation models. Instead, it is **broadly applicable to any FL scenario where clients are heterogeneous**, as demonstrated by our uni-modal experiments in Table 6 (Sec. 6.2) and our experiments with small-scale models (e.g., T5-based models) in Table 36 (Sec. A.26).
>
> Regarding the baselines’ target, as noted above, many baselines (e.g., FedDPA, FedIT, FedMKT, TAKFL, FedDAT) already tackle federated learning with million- or billion-scale models in their original papers. In addition, baselines such as DITTO are also used with large-scale models in other prior works, e.g., [15] includes DITTO as an FL baseline using Llama-7B as the underlying model, and [16] evaluates DITTO with SAM as the base model.
>
> In summary, we respectfully argue that (i) our contribution lies in addressing heterogeneous clients in realistic federated learning scenarios, and (ii) the baselines we employ are appropriately selected, consistent with their original papers and prior follow-up works.

---

### Official Review · Reviewer_pywM · 2025-10-30

**Soundness:** 3
**Presentation:** 2
**Contribution:** 3
**Rating:** 6
**Confidence:** 3

**Summary:**

This paper aims to address the challenges of personalized federated learning in scenarios where both data and models are heterogeneous. The authors proposed a framework named FedMosaic, which consists of two core components: the RELevance-guided Aggregation (RELA) and PQ-LoRA. RELA is a task-relevance-based model aggregation strategy that constructs customized global models for clients. PQ-LoRA is a module shareable across heterogeneous models, addressing differences in model depth and dimensions through "block-wise aggregation" and "weight alignment". Additionally, the authors propose DRAKE, a comprehensive multimodal federated learning benchmark that covers 40 different tasks and simulates real-world task diversity and temporal distribution shifts. Experiments on both multi-modal and text-only benchmarks demonstrate that FedMosaic outperforms PFL methods in both personalization and generalization.

**Strengths:**

Overall, this paper represents a meaningful problem in personalized federated learning. The authors find that existing personalized federated learning methods are still confined to simplified scenarios with highly homogeneous data and models across clients, while real-world scenarios are more complex. They proposed FedMosaic, which addresses the simultaneous heterogeneity of data and models through a task-correlation-aware model aggregation strategy and dimension-invariant modules. Additionally, they introduced DRAKE, a comprehensive multimodal federated learning benchmark.

**Weaknesses:**

**Major Weaknesses:**

Overall, this paper has some merits, but there are a few weaknesses that stop me from giving a higher rating. My major concerns are as follows.

(1) The paper mentions that the FedMosaic method does not require high computational costs, and the authors' experiments indeed include sections related to computational costs. However, the process of weight alignment in PQ-LoRA seems to be relatively complex, and the paper does not provide information about the computational costs of this part.

(2) Section 4.2.1 of the paper mentions using CKA to find relative depth alignment and demonstrates with Llama-1B and Llama-3B, but lacks sufficient explanation regarding the applicability of this method.

(3) In the weight alignment in the PQ-LoRA section, it mentions freezing the smaller model as a pivot and updating the larger model. The paper lacks an explanation for why this strategy was adopted.

(4) DRAKE is one of the contributions of the paper, but extensive details are provided in the appendix, with relatively limited space allocated in the main text.

**Minor Weaknesses:**

(1) Figure 2 provides an overview of FedMosaic, but the image is relatively dense and slightly lacking in readability, so it could be adjusted a bit.

**Questions:**

Please clarify my concerns in the weakness part.

---

> ### Author Response · Authors · 2025-11-19
> **Response to Reviewer pywM (1/2)**
>
> Dear Reviewer pywM,
>
> We sincerely thank you for your helpful feedback and insightful comments. We address your comments and questions below.
>
> > The paper mentions that the FedMosaic method does not require high computational costs, and the authors' experiments indeed include sections related to computational costs. However, the process of weight alignment in PQ-LoRA seems to be relatively complex, and the paper does not provide information about the computational costs of this part.
>
> $\to$ We thank the reviewer for raising this important question. We have added details about the computational cost of PQ-LoRA alignment in Sec. A.8, L1546 of the revision, showing that it incurs only a limited additional computational cost, with approximately 6.25% more than SFT (the most efficient baseline). The limited overhead can be achieved because : (i) PQ-LoRA alignment is performed only once before federated training to establish a shared initialization, and (ii) it is required only once for heterogeneous model-architecture pairs, not for all client model pairs (since some clients may share the same architecture), as noted in L303.
>
> Specifically, as described in Sec. A.17, we use 1,250 iterations (= 5,000 samples / batch size 4) for alignment, performed only once before federated learning, leading to a small additional cost. We note that this **cost can be further reduced with fewer aligning iterations while maintaining comparable performance**, as shown in Tab. 26, where using 156 iterations (= 625 samples / batch size 4) incurs only **0.78% additional cost**.
>
>
> > Section 4.2.1 of the paper mentions using CKA to find relative depth alignment and demonstrates with Llama-1B and Llama-3B, but lacks sufficient explanation regarding the applicability of this method.
>
> $\to$ (We understood “applicability of this method” as referring to whether the relative depth alignment found using CKA generalizes beyond Llama-1B and Llama-3B to other model sizes and architectures. Please correct us if you meant something else!) We indeed observed consistent layer-wise alignment across multiple architectures, including various same-family results (e.g., Llama-1B/3B/8B, Qwen-0.5B/1.5B/3B), and even cross-family pairs (e.g., Llama-1B vs. Qwen-1.5B). We realized that only the Llama-1B/3B example was shown in the main paper, and we had not explicitly pointed readers to the broader results. We have now added clear references to the relevant appendix sections (Sec. A.10) in the revision. Thank you for pointing this out!
>
> Furthermore, to strengthen the claim that relative depth alignment holds broadly, we additionally include InternLM vs. Llama results showing a similar trend in Fig. 13 (Sec. A.10) of the revision, further supporting our findings.
>
>
> > In the weight alignment in the PQ-LoRA section, it mentions freezing the smaller model as a pivot and updating the larger model. The paper lacks an explanation for why this strategy was adopted.
>
> $\to$ We appreciate the reviewer’s thoughtful question. We freeze the smaller model as a pivot because, **over time, new models that become available are typically larger and stronger, rather than smaller and weaker ones**. In this scenario, using the larger model as the pivot would cause the pivot to change whenever a new model is introduced, requiring re-alignment of all previously aligned models. For example, when aligning LLaVA-1B and LLaVA-3B, if we freeze the larger model (i.e., LLaVA-3B) as the pivot, the introduction of LLaVA-8B would make it the new pivot. Consequently, both LLaVA-1B and LLaVA-3B would need to be updated during alignment with LLaVA-8B, which would break the previously established alignment between LLaVA-1B and LLaVA-3B.
>
> Moreover, we note that **this strategy does not affect performance**. As shown in Table A, employing the larger model as the pivot yields comparable performance to our strategy. In summary, we adopt this approach not because it offers better performance, but because it provides a stable pivot for alignment when larger models are newly deployed. We have included these results in Sec. A.29, Tab. 38 of the revision.
>
> **Table A. Effect of pivot model selection for PQ-LoRA alignment in heterogeneous PFL-dynamic on DRAKE**
> |PQ-LoRA alignment scenario|Pivot model|Self $A_{last} \uparrow$|Self $A_\text{AUC} \uparrow$|Others $A_{last} \uparrow$|Others $A_\text{AUC} \uparrow$|
> |-|-|-|-|-|-|
> |LLaVA-Llama3.2-1B / LLaVA-Llama3.2-3B|small (1B)|**67.86**$\pm$**0.51**|**59.83**$\pm$**0.15**|51.16$\pm$0.04|**49.36**$\pm$**0.08**|
> ||large (3B)|67.71$\pm$0.27|59.54$\pm$0.05|**51.46**$\pm$**0.37**|49.29$\pm$0.19|
> |LLaVA-Qwen2.5-1.5B / LLaVA-Llama3.2-3B|small (1.5B)|70.62$\pm$0.29|63.33$\pm$0.26|**52.56**$\pm$**0.08**|50.76$\pm$0.01|
> ||large (3B)|**70.83**$\pm$**0.35**|**63.40**$\pm$**0.25**|52.55$\pm$0.21|**50.82**$\pm$**0.01**|

---

> ### Author Response · Authors · 2025-11-20
> **Response to Reviewer pywM (2/2)**
>
> > DRAKE is one of the contributions of the paper, but extensive details are provided in the appendix, with relatively limited space allocated in the main text.
>
> $\to$ We thank the reviewer for this helpful suggestion. In the revision, we moved details of the proposed DRAKE benchmark from the appendix to the main text. Specifically, the revised DRAKE section (Sec. 5) now includes: (i) detailed explanations of all sub-groups (VQA, visual relation, multi-modal reasoning, and unseen tasks), and (ii) illustrative data samples (Fig. 6). We believe these additions provide a clearer and more comprehensive understanding of DRAKE within the main paper. If there are further components you feel should be included, we would be happy to incorporate them.
>
> > Figure 2 provides an overview of FedMosaic, but the image is relatively dense and slightly lacking in readability, so it could be adjusted a bit.
>
> $\to$ We thank the reviewer again for this helpful suggestion. In the revision, we have updated Fig. 2 by removing unnecessary arrows and mathematical notations, making it less dense and easier to understand.

---

### Official Review · Reviewer_GXyk · 2025-11-11

**Soundness:** 3
**Presentation:** 3
**Contribution:** 4
**Rating:** 8
**Confidence:** 4

**Summary:**

The paper considers personalized federated learning (pFL) of multimodal large language models (MLLMs) under realistic scenarios that involve not only data heterogeneity, but also model architecture and model family heterogeneity, and task diversity. The paper designs a new method called FedMosaic that enables FL style collaboration across clients even in the simultaneous presence of all of the heterogeneities. FedMosaic has two important components - RELA (Relevance-guided Aggregation) and PQ-LoRA (Dimension-invariant Low-Rank Adaptation) that respectively address (data, task) and model heterogeneities. In an effort to make evaluation more realistic, the paper also introduces a new benchmark called DRAKE that incorporates all these heterogeneities and further includes aspects like dynamic distribution shifts and unseen tasks. Extensive experimental evaluation is provided in the paper for FedMosaic as well as several other state-of-the-art pFL baselines, which establish superior characteristics of FedMosaic.

**Strengths:**

(S1) The writing and presentation of the paper are very clear in terms of both algorithm design and experimental results. Adequate intuitions are provided throughout the paper and appendices. The problem is well motivated, related work is well cited, and the contributions are contextualized appropriately.

(S2) FedMosaic is an original and interesting solution to a very complex practical problem of multiple heterogeneities in pFL of MLLMs. This is a significant contribution to the field in terms of both ideas and solutions. RELA and PQ-LoRA would likely find use in other problems too.

(S3) The supporting experimental evidence provided in the paper and appendices is quite exhaustive and impressive. The paper undertakes a wide diversity of studies to establish the characteristics of FedMosaic from several angles and shows competitive or improved performance w.r.t. all compared baselines.

**Weaknesses:**

(W1) Introducing a new benchmark in an algorithms paper is counterproductive. The benchmark would be difficult to discover for any reader. To a reviewer, the benchmark's design is impossible to evaluate when only 10 lines can be allocated to it in the main body. While DRAKE looks extremely useful, there are several nuances which can only be understood by carefully reading multiple sections in the appendices. My opinion is that DRAKE should be submitted as a separate datasets & benchmarks style paper for it to be properly peer-reviewed as such.

(W2) There is no benchmark called HFLB in (Chen et al., 2024). The name/citation should be corrected.

**Questions:**

(Q1) Section 4.2.1 Figure 4, and Appendices A.12, A.17: Even though supporting empirical evidence is provided, I don't understand why layer correlations should exist across model families (Llama, Qwen, etc.). Is this exclusively caused by the common training data source from which $D_P$ is sampled? PQ-LoRA would only work if such correlation exists, right? How should one think about the system when common subset from pre-training/post-training data may be unknown/may not exist/may be inaccessible?

(Q2) Line 73, 210: Does the system require a separate model instance on the server for each client? If yes, is that scalable to large number of clients? If no, what do experiments suggest about observed number of model instances on the server per client, across datasets of interest?

---

> ### Author Response · Authors · 2025-11-19
> **Response to Reviewer GXyk (1/2)**
>
> Dear Reviewer GXyk,
>
> We sincerely thank you for your helpful feedback and insightful comments. We address your comments and questions below.
>
> > (W1) Introducing a new benchmark in an algorithm paper is counterproductive. The benchmark would be difficult to discover for any reader. To a reviewer, the benchmark's design is impossible to evaluate when only 10 lines can be allocated to it in the main body. While DRAKE looks extremely useful, there are several nuances which can only be understood by carefully reading multiple sections in the appendices. My opinion is that DRAKE should be submitted as a separate datasets & benchmarks style paper for it to be properly peer-reviewed as such.
>
> $\to$ We sincerely appreciate your thoughtful suggestion. We also thought about submitting it as a separate paper but we include it here as it will be shown to be useful used to evaluate methods in the realistic PFL scenario. For better readability, we moved details of the proposed DRAKE benchmark from the appendix to the main text of the revision. Specifically, the revised DRAKE section (Sec. 5) now includes: (i) detailed explanations of sub-groups (VQA, visual relation, multi-modal reasoning, and unseen tasks), and (ii) illustrative data samples (Fig. 6). We believe these additions provide a clearer and more comprehensive understanding of DRAKE within the main paper. If there are further components you feel should be included, we would be happy to incorporate them.
>
> > (W2) There is no benchmark called HFLB in (Chen et al., 2024). The name/citation should be corrected.
>
> $\to$ Thank you for pointing this out. Yes, there is no benchmark with the name, we refer to the benchmark introduced in (Chen et al., 2024) as HFLB (Heterogeneous Federated Learning Benchmark) to reflect its focus on heterogeneous client tasks.  We apologize for the confusion. We have clarified this naming  in the revision (Sec. 6.1, L398).
>
> > (Q1) Section 4.2.1 Figure 4, and Appendices A.12, A.17: Even though supporting empirical evidence is provided, I don't understand why layer correlations should exist across model families (Llama, Qwen, etc.).
>
> >> Is this exclusively caused by the common training data source from which $D_p$ is sampled? How should one think about the system when common subset from pre-training/post-training data may be unknown/may not exist/may be inaccessible?
>
> $\to$ No, the layer correlation is not caused by the common dataset $D_p$. Note that the correlation is measured between **off-the-shelf pre-trained models** (e.g., Llama, Qwen), not models aligned using $D_p$, using unseen samples for the off-the-shelf models (e.g., DRAKE’s subset). This indicates the observed correlation exists independently of any training through a shared dataset, making a **common pre-/post-training dataset unnecessary**.
>
> Instead, we believe the correlation primarily arises from the inherent tendency of transformer layers at similar relative depths to learn similar functions. Prior studies have shown that layers at different depths in LLMs serve distinct functions (e.g., early layers focus on lexical and syntactic processing, while later layers handle semantic reasoning and token prediction) [1, 2, 3], regardless of architecture. Such functional specialization can naturally lead to the observed cross-family correlations, consistent with recent work [4] that reported similar layer-wise correlations across different model families (e.g., Llama, Gemma, Mistral), further supporting our empirical findings.
>
> Moreover, to demonstrate that this correlation trend generally holds across different models, not just between Llama and Qwen, we have additionally included the layerwise correlation analysis between InternVL [5] and LLaVA-Llama in Fig. 13 (Sec. A.10) of the revision, which shows the same trend as our previous findings.
>
> >> PQ-LoRA would only work if such correlation exists, right?
>
> $\to$ Yes, PQ-LoRA operates under the presence of layer-wise correlation, but such correlations are generally observed across models regardless of their training data, as shown with various off-the-shelf models (e.g., Llama vs. Qwen vs. InternLM), i.e., not aligned models, above, demonstrated in Sec. A.17, and supported by prior work [4].

---

> ### Author Response · Authors · 2025-11-19
> **Response to Reviewer GXyk (2/2)**
>
> > (Q2) Line 73, 210: Does the system require a separate model instance on the server for each client? If yes, is that scalable to large number of clients? If no, what do experiments suggest about observed number of model instances on the server per client, across datasets of interest?
>
> $\to$ No, our proposed **FedMosaic does not store a separate model instance on the server for each client**. Instead, in each round, the server receives local LoRA adapters from local clients, aggregates them using client-specific weighting with RELA, and redistributes the aggregated model back to the clients. Thus, the server only aggregates the local models received in each round and redistributes the aggregated model without storing or further training any per-client server models, unlike prior methods that keep separate server-side models to train (e.g., PerAda, TAKFL) or extract logits (e.g., FedMKT) for global knowledge sharing.
>
>
> [1] Tenney et al., BERT rediscovers the classical NLP pipeline, ACL 2019 \
> [2] Durrani et al., . Analyzing individual neurons in pre-trained language models, EMNLP 2020 \
> [3] Kaplan et al., From tokens to words: On the inner lexicon of LLMs, ICLR 2025 \
> [4] Wolfram et al., Layers at Similar Depths Generate Similar Activations Across LLM Architectures, arXiv 2504 \
> [5] Chen et al., Internvl: Scaling up vision foundation models and aligning for generic visual-linguistic tasks, CVPR 2024

---

### Author Response · Authors · 2025-11-19
**General Response**

Dear reviewers and AC,

We sincerely thank the reviewers for their valuable feedback and encouraging comments including comprehensive experiments (Reviewer **GXyk, oYUk**), clear presentation (Reviewer **GXyk,  UdQR**), interesting and well-designed idea (Reviewer **GXyk, pywM, oYUk**), significant improvements (Reviewer **GXyk, UdQR, oYUk**), well-motivated problem (Reviewer **GXyk, oYUk**), well-organized related work (Reviewer **GXyk**), meaningful and realistic setup (Reviewer **GXyk, pywM, oYUk**), comprehensive and useful proposed benchmark (DRAKE) (Reviewer **pywM, oYUk**), general applicability of the proposed methods (Reviewer **GXyk**), comprehensive appendix (Reviewer **GXyk, UdQR**).

We appreciate your constructive comments on our manuscript. In response to the comments, we have carefully revised and enhanced the manuscript with the following additional discussions and experiments:
- Additional comprehensive comparison with existing PFL methods (Section 2)
- Clarification about PFL objective (Section 3)
- Detailed justification of the choice of the pivot model (Section 4.2.1, Section A.29)
- Additional details of the proposed DRAKE (Section 5)
- Clarification about HFLB benchmark (Section 6.1)
- Discussion about computational cost of PQ-LoRA alignment (Section A.8)
- Additional empirical analysis of block-wise aggregation (Section A.10)
- Additional hyperparameter analysis (Section A.18)
  - Effect of gradient computing interval $m$ in RELA
  - Effect of choice of gradient layer in RELA.
- Additional experimental results on heterogeneous PFL with heterogeneous LoRA ranks (Section A.24)
- Additional experimental results on conventional non-i.i.d. PFL setups (Section A.25)
- Additional experimental results on heterogeneous PFL with small-scale models (Section A.26)
- Additional comparison with privacy-preserving baselines with our proposed sanitized gradient (Section A.30)
- Visualization of task relevance of the proposed DRAKE benchmark (Section A.31)

We have uploaded the first revision of the manuscript (changes are highlighted by $\color{red}{\text{red}}$ color).

We hope our response and revision sincerely address all the reviewers’ concerns.

Thank you very much.

Best regards,

Authors.

---

### Comment · Area_Chair_jpuj · 2025-11-24

Dear Reviewers,

**We kindly encourage you to review and respond to the authors’ rebuttals**. Your timely feedback is important for ensuring a fair and thorough review process. Thank you for your contributions to ICLR 2026.

AC

---

### Author Response · Authors · 2025-11-25
**Second revision**

We have uploaded the second revision of the manuscript. The revision includes additional experiments under asynchronous distribution shifts, where distribution changes occur at different times across clients, to answer the reviewers’ question (UdQR).

Summary of the changes
- Additional experimental results under asynchronous distribution shift (Section A.32)

---

### Author Response · Authors · 2025-12-04
**Rebuttal Summary (2/2)**

**Rebuttal**

During the rebuttal period, we made every effort to thoroughly address the reviewers’ concerns through clarifications, additional experiments, and revisions, including but not limited to:
- Additional analyses (e.g., computational cost, hyperparameter, asynchronous distribution shift),
- Clarifications of design choices (e.g., pivot model selection, PQ-LoRA alignment dataset, CKA applicability) with supporting empirical results,
- Expanded details and justification for the DRAKE benchmark in Sec. 5.

We also actively discussed with the sole negative reviewer UdQR and responded to the main concerns as follows:

> **Limited model heterogeneity due to assuming the same LoRA rank across clients.**

$\to$ Although PQ-LoRA in FedMosaic primarily targets width (i.e., dimension) and depth (i.e., number of layers) heterogeneity that were not addressed in prior work, **we empirically show that our method is not restricted to a uniform LoRA rank setting.** We integrate FedMosaic with HETLoRA [1], a baseline that allows federated learning among clients with heterogeneous LoRA ranks on the same backbone model. Results in Table B **shows strong improvements in rank-heterogeneous PFL setups, demonstrating that FedMosaic remains effective beyond the same-rank assumption.**

> **Privacy robustness of the sanitized gradient $\tilde{g}$ in RELA.**

$\to$ We sanitize gradients used to measure cross-client task relevance via EMA, subsampling, and noise injection for privacy protection. To assess its privacy robustness, we run gradient inversion attacks and compare against existing defenses. As shown in Table C, **our defense mechanism effectively prevents reconstruction, confirming that the proposed mechanism is privacy-robust in addition to being efficient.**

> **Applicability of compared baselines to our PFL setup.**

$\to$ Our primary setting is heterogeneous PFL on large-scale multi-modal models with distinct tasks per client. We clarified that:
- **Model sizes**: Many baselines we compare against **already operate on large-scale models** (e.g., Llama-7B)
- **Data split**: We respectfully argue that **prior setups (e.g., non-IID splits of a single dataset) are relatively simplified and less reflective of the real-world complexity, as Reviewers pywM and oYUk also denoted**, whereas our multi-modal PFL setup is closely aligned with practical deployment scenarios, as also highlighted by Reviewers GXyk, oYUk, and pywM
- **Modality**: We have **already provided uni-modal (e.g., text-only) experiments** (Tab. 6, Sec. 6.2 and Tab. 20, Sec. A.15), demonstrating that our method is not confined to multi-modal settings

Furthermore, we also evaluate our method under the same simplified setups used by baselines: (i) non-IID splits of a single dataset (Table D), (ii) small-scale uni-modal models (Table E, F), and **FedMosaic still shows significant improvements, showing that it is not limited to any specific setup.**

> **Practicality of multi-modal personalized federated learning in the real-world.**

$\to$ We provide concrete real-world examples for multi-modal PFL, including MCP-based AI agent system, home-assistant robots, and hospitals examples, where each client AI conducting multi-modal tasks (e.g., ego-centric vision + text instruction, MRI+diagnosis) benefits from sharing knowledge among clients doing similar tasks. **We believe these abundant real-world examples are well-aligned with our experimental setup, which the Reviewers GXyk, pywM, oYUk found realistic and well-motivated.** We believe that both our FedMosaic and the proposed DRAKE benchmark offer a meaningful step toward enabling knowledge sharing even across heterogeneous clients, making collaborative learning more broadly applicable and more scalable to real-world scenarios.

Our revision is summarized in General Response. Overall, we conducted a substantial amount of experiments (Tables A-K, 7 new appendix sections, and 12 additional figures and tables in the revision) and provided detailed clarifications. We believe our additional experiments and clarifications (e.g., rank heterogeneous setup, privacy analysis) strengthen both the clarity and the technical scope of the work.
We deeply appreciate your dedication and careful consideration, and we hope our revision and responses will be fairly reflected in the final decision.

Sincerely,

Authors

[1] Cho et al., Heterogeneous LoRA for Federated Fine-tuning of On-Device Foundation Models, EMNLP 2024

---

### Author Response · Authors · 2025-12-04
**Rebuttal Summary (1/2)**

Dear ACs, SACs, and PCs

We sincerely appreciate your time and effort in serving the ICLR community in these unusual circumstances. Below is a concise summary of the strengths highlighted by the reviewers and the issues we have addressed during the rebuttal.

**Strength**
- **Tackling a well-motivated and realistic multi-modal heterogeneous setup** for personalized federated learning (PFL), capturing both data heterogeneity (e.g., VQA vs. visual reasoning) and model heterogeneity (e.g., Llama3.2-3B vs. Qwen2.5-1.5B) (Reviewers GXyk, pywM, oYUk). Reviewers pywM and oYUk further note that **existing PFL work often oversimplifies such heterogeneity**
- **Proposing an interesting and well-designed approach** (Reviewer GXyk, pywM, oYUk), FedMosaic, composed of RELA and PQ-LoRA, enabling effective knowledge sharing under the heterogeneous PFL setup. Reviewer GXyk further highlights it as **an original and a significant contribution to the field**
- **Demonstrating strong improvement over strong baselines and broad applicability** of the proposed method under comprehensive settings and detailed ablations (ALL Reviewers)
- Providing DRAKE, **a useful and comprehensive multi-modal PFL benchmark** (Reviewer pywM, oYUk)
- **Clear presentation with a comprehensive appendix** (Reviewer GXyk, UdQR), with Reviewer GXyk describing it as **exhaustive and impressive**

---

### Meta-Review · Area_Chair_7FFA · 2026-01-06

**Summary:**

This paper studies personalized federated learning (pFL) for multimodal large language models (MLLMs) under realistic settings that involve multiple forms of heterogeneity beyond data heterogeneity alone. To address these challenges, the authors propose a novel method, termed FedMosaic, which enables effective federated learning–style collaboration among clients even in the simultaneous presence of multiple heterogeneities. The paper presents extensive experimental evaluations comparing FedMosaic with several state-of-the-art pFL baselines, and the results consistently demonstrate the superior performance and robustness of the proposed method.

The paper was reviewed by four reviewers, almost all of whom were generally inclined toward acceptance. Reviewer UdQR engaged in detailed discussions with the authors on several specific aspects, including implementation details and the definitions of certain formulas, to which the authors provided thorough and satisfactory responses.

**Reviewer Concerns:**

During the rebuttal phase, the authors addressed the following concerns:
- Methodological details of the paper, including clarifications of previously unclear formulas and training procedures;
- Experimental details, with the authors providing additional information on computational costs and other implementation-related aspects.

**Reviewer Scores:**

Among the four reviewers, only Reviewer UdQR actively participated in the discussion. However, despite the authors’ thorough responses to the other three reviewers, it is unlikely that their scores will change, as they are already sufficiently high.

---

### Decision · Program_Chairs · 2026-01-26

Accept (Poster)